# Learning Interpretable and Influential Directions with Signal Vectors and Uncertainty Region Alignment

## Abstract

Latent space directions have played a key role in understanding, debugging, and fixing deep learning models. Concepts are often encoded in distinct feature space directions, and evaluating impact of these directions on the model's predictions, highlights their importance in the decision-making process. Additionally, recent studies have shown that penalizing directions associated with spurious artifacts during training can force models to unlearn features irrelevant to their prediction task. Identifying these directions, therefore, provides numerous benefits, including a deeper understanding of the model's strategy, fostering trust, and enabling model correction and improvement. We introduce a novel unsupervised approach utilizing signal vectors and uncertainty region alignment to discover latent space directions that meet two key debugging criteria: significant influence on model predictions and high level of interpretability. To our knowledge, this method is the first of its kind to uncover such directions, leveraging the inherent structure of the feature space and the knowledge encoded in the deep network. We validate our approach using both synthetic and real-world benchmarks, demonstrating that the discovered directions effectively fulfill the critical debugging criteria.

## 1 Introduction

Central to the functioning of deep learning models is the latent space, where, to a large-extend, high-level concepts are encoded in distinct directions Szegedy et al. (2014); Alain & Bengio (2018); Zhou et al. (2018); Kim et al. (2018); Nanda et al. (2023). The identification of these directions has been proven to be extremely valuable in providing explanations for deep learning models Kim et al. (2018); Zhou et al. (2018); Pfau et al. (2020); Schrouff et al. (2022); Yuksekgonul et al. (2023) as well become the cornerstone of correcting models that rely on spurious correlations, biases, or irrelevant features within the data Anders et al. (2022); Pahde et al. (2023).

Initially, concept directions were modeled using the *filter* weights of a linear classifier designed to distinguish samples representing the concept from those that do not Zhou et al. (2018); Kim et al. (2018). However, under a *signal - distractor* data model, in which latent space representations are considered as superpositions of components along an informative direction related to the concept, called the concept's *signal* direction, and components along non-informative, noisy directions, called *distractors*, filters were found to be mostly influenced by distractors, deviating from concept's true signal direction Kindermans et al. (2017); Pahde et al. (2024). As a mitigation measure, in the case of binary concept labels, Pahde et al. (2024) suggested *pattern-CAVs*, as a better estimator of the concept's signal direction.

Figure 1: Each image to be classified by the deep network is represented as a set of spatial elements in the network's latent space. Each element represents an encoded patch of the image with feature values determined by the convolutional operations of the network. The graph illustrates the concepts of signal directions, filter directions and concept hyperplanes. It also exemplifies the phases of direction learning which is separate of the phases of interpretability and influence testing.

Both *filters* and concept *signals* constitute vectors, which allows us to refer to them as *directions*. A filter, when accompanied with a bias constitutes a classifier that can answer questions of interpretability like: "Is this a representation of the concept motorbike?". Thus, whenever a filter can reliably predict the presence of a concept we term it an *interpretable direction* and we refer to the respective classifier as *concept detector*. Similarly, a signal direction can answer questions of influence like "Does the concept boat influence the prediction of class harbor?". For this reason, whenever a signal direction significantly influences the network's predictions, we term it an *influential direction*. For each concept, there does a exist a filter-signal pair with each direction serving its own purpose. We will be using the term *concept directions* to refer to filter - signal pairs of high interpretability and influence.

In this work, we aim to identify concept directions that fulfill two key debugging properties: a) influence on the network's predictions b) be as interpretable as possible. We address these objectives with a bottom-up approach. We focus on learning a set of directions that are crucial to the network's predictions first, while optimizing interpretability through a sparsity property Doumanoglou et al. (2023). By doing so, we aim to capture what the model deems significant for predictions from its own perspective, rather than speculating on concepts it might rely on and potentially rejecting many hypotheses after influence evaluation. This bottom-up approach makes the proposed method unsupervised and it does not require access to concept annotations to identify the directions. We base our method on the foundations of unsupervised interpretable basis learning Doumanoglou et al. (2023; 2024), which already, to some extend, addresses the discovery of interpretable directions, while we also make a significant leap forward to additionally identify influential directions by a) considering and extending the signal-distractor theory to the case of encoding multiple concepts (instead of the binary setting of prior work Pahde et al. (2024)) b) in this multi-concept setting, we propose *signal vectors* as estimators of a concept's signal direction and provide empirical evidence that those vectors align with ground truth signal directions in an experiment with synthetic data (when pattern-CAVs fail) and c) we propose Uncertainty Region Alignment, a loss term that aligns the subspace where the studied network makes uncertain predictions with the subspace where concept detectors are maximally uncertain. By empirical evidence, the latter allows optimization of the filters for interpretability and signal vectors for influence. Experiments on a state-of-the-art convolutional image classifier support that, compared to supervised direction learning, the classifiers learned with our method, manifest competitive performance in concept classification and segmentation tasks, while the learned signal vectors show substantial influence on the model's predictions, far surpassing any influence from a possible random signal.

## 2 BACKGROUND

### 2.1 PRELIMINARIES

Let $\boldsymbol{X} \in \mathbb{R}^{H \times W \times D}$ denote the representation of an image in an intermediate layer of a convolutional neural network with spatial dimensions $H, W \in \mathbb{N}^+$ and feature space dimensionality $D \in \mathbb{N}^+$. Let also $\boldsymbol{x_p} \in \mathbb{R}^D$ denote an element of this representation at the spatial location $\boldsymbol{p} = (w, h)$, $w \in \{0, 1, ..., W - 1\}, h \in \{0, 1, ..., H - 1\}$.

## 2.2 Signals, Distractors, Filters, Pattern-CAVs

In the case of encoding a single concept $i$, Kindermans et al. (2017); Pahde et al. (2024) suggest the following (binary) model for the data generation process of feature representations: $\boldsymbol{x_p} = \alpha_{\boldsymbol{p}} \boldsymbol{s}_i + \beta_{\boldsymbol{p}} \boldsymbol{d}$, $\boldsymbol{s}_i, \boldsymbol{d} \in \mathbb{R}^D, \alpha_{\boldsymbol{p}}, \beta_{\boldsymbol{p}} \in \mathbb{R}$. In this model, $\boldsymbol{s}_i$ is the direction which contains the information whether $\boldsymbol{x_p}$ is a member of concept $i$ and is called the signal direction of concept $i$. The actual information is hidden in the coefficient $\alpha_{\boldsymbol{p}}$, which we call the *value* of the concept's signal. In simple terms, the larger $\alpha_{\boldsymbol{p}}$ is, the more confident we are that $\boldsymbol{x_p}$ belongs to concept $i$. $\boldsymbol{d}$ is called the distractor direction, and models noise or information unrelated to the concept. $\beta_{\boldsymbol{p}}$ is modeled to follow a gaussian distribution $\mathcal{N}(\mu, \sigma^2)$ and its value is considered independent of whether $\boldsymbol{x_p}$ belongs to concept $i$. The answer to the question: "Is $\boldsymbol{x_p}$ a member of concept $i$ ?" is hidden in the value of the concept's signal $\alpha_{\boldsymbol{p}}$. Since stronger values of $\alpha_{\boldsymbol{p}}$ indicate more confidence for the presence of the concept, we may find a threshold $b_i$ against whom we may compare $\alpha_{\boldsymbol{p}}$ to answer the question. According to Kindermans et al. (2017), the value of the signal $\alpha_{\boldsymbol{p}}$ can be extracted via a *regression filter* $\boldsymbol{w}_i$: $z_{\boldsymbol{p},i} = \boldsymbol{w}_i^T \boldsymbol{x_p} = \alpha_{\boldsymbol{p}} \boldsymbol{w}_i^T \boldsymbol{s}_i + \beta_{\boldsymbol{p}} \boldsymbol{w}_i^T \boldsymbol{d}$, if we choose $\boldsymbol{w}_i : \boldsymbol{w}_i \perp \boldsymbol{d}$, and $\boldsymbol{w}_i^T \boldsymbol{s}_i = 1$. Combined with the knowledge of $b_i$ which may be learned from data, this regression filter can be turned into a classifier: $y_{\boldsymbol{p},i} = \sigma(z_{\boldsymbol{p},i} - b_i)$ which can provide answers to the above question.

Supposing that we do have access to the actual value of the signal (for instance if we do have access to a regression filter), Haufe et al. (2014); Kindermans et al. (2017) provided the following formula that can estimate the concept's signal direction:

$$\hat{\boldsymbol{s}}_i = \frac{\text{cov}[\boldsymbol{x_p}, z_{\boldsymbol{p},i}]}{\sigma_{z_{\boldsymbol{p},i}^2}} \tag{1}$$

In this formula, $\sigma_{z_{p,i}}^2$ denotes the variance of the signal values in the dataset. While this signal estimator requires access to the values of the signal, when trying to explain the latent space, we miss this information. In practice, we only have access to $\boldsymbol{x_p}$, while $\boldsymbol{s}_i$ and $\boldsymbol{d}$ constitute latent variables of the underlying process. Based on Haufe et al. (2014); Kindermans et al. (2017), Pahde et al. (2024) introduced pattern-CAVs as concept signal estimators without the need to have explicit access to the signal values. Instead, their estimator requires access to labeled data, i.e. concept's positive and negative samples. Their estimator is based on (1) where they approximate the signal value with binary labels, i.e. $z_{\boldsymbol{p},i} \in \{0, 1\}$ and is:

$$\hat{s}_P = \mathbb{E}_p(x_p^+) - \mathbb{E}_p(x_p^-)$$

with $x_p^+$ and $x_p^-$ denoting positive and negative representation samples of the concept.

## 2.3 Unsupervised Interpretable Direction Learning

Recent research (Doumanoglou et al. (2023)) proposed an unsupervised method that can be used to identify concepts from the bottom-up, i.e. from the structure of the feature space. Motivated by the fact that concepts are encoded in the directions of the latent space, the method partitions this space into linear regions each defined by a hyperplane and its normal vector. Each region corresponds to a cluster, where features coming from an unlabeled *concept dataset* (which can be the same as the network's training set) are assigned. The method learns $\boldsymbol{W}$ and $\boldsymbol{b}$ of a feature-to-cluster membership function $\boldsymbol{y_p} = \sigma(\boldsymbol{W}^T \boldsymbol{x_p} - \boldsymbol{b}) \in [0, 1]^I, \boldsymbol{W} \in \mathbb{R}^{D \times I}, \boldsymbol{b} \in \mathbb{R}^I$ with $I$ denoting the number of clusters. Each feature is soft-assigned to a small number of clusters, with this number being minimized to fulfill a sparsity property that they show it is essential for interpretability. This is based on the observation that the semantic label that can be attributed to an image patch is only one (or a few) among a larger set of possible semantic labels, which implies sparsity at the semantic level. Sparsity in the assignments is accomplished via the following two loss terms, with the first being the *Sparsity Loss* ($\mathcal{L}^s$) and the second the *Maximum Activation Loss* ($\mathcal{L}^{ma}$), which enforces cluster membership to be binary:

$$\mathcal{L}^s = \mathbb{E}_{\boldsymbol{p}}(\mathcal{L}_{\boldsymbol{p}}^s), \quad \mathcal{L}^{ma} = -\mathbb{E}_{\boldsymbol{p}}(\boldsymbol{q}_p^T \log_2(\boldsymbol{y_p})), \quad \mathcal{L}_{\boldsymbol{p}}^s = \mathcal{H}(\boldsymbol{q_p}), \quad \boldsymbol{q_p} = \frac{\boldsymbol{y_p}}{||\boldsymbol{y_p}||_1} \tag{2}$$

with $\mathcal{H}$ denoting entropy. From another viewpoint, the columns of $\boldsymbol{W}$ and the elements of $\boldsymbol{b}$ (e.g. $\boldsymbol{w}_i, b_i$) constitutes a linear classifier or concept detector $y_{\boldsymbol{p},i} = \sigma(\boldsymbol{w}_i^T \boldsymbol{x_p} - b_i)$. The method also

maximizes linear separability of the features by minimizing the inverse of classification margin $M_i = \frac{1}{||\boldsymbol{w}_i||_2}$ (*Maximum Margin Loss* - $\mathcal{L}^{mm}$ section A.5) and penalizes clusters with few assignments using the *Inactive Classifier Loss* ($\mathcal{L}^{ic}$ Doumanoglou et al. (2024)) provided in section A.4. Although not guaranteed to align with human intuition, the sparsity property of the transformed representations allows for possible concept definition or identification. Overall, the method of Doumanoglou et al. (2023; 2024), addresses the debugging property regarding interpretability. Yet, it disregards the signal-distractor theory and does not provide a suggestion for the debugging property of influence.

## 2.4 DIRECTION LABELING

In Doumanoglou et al. (2023; 2024), the filters of the learned classifiers correspond to an orthogonal feature space basis with each vector pointing towards the linear region of a cluster. While the method does not require annotations to learn the basis, to measure the interpretability of the clustering (thus for quantitative evaluation), they use Network Dissection Bau et al. (2017). Network Dissection is a basis labeling method which assigns the best-possible human semantic label to each one of the basis vectors based on how well the classifier behind the vector performs in identifying known concepts from a densely annotated concept dataset.

## 2.5 CONCEPT INFLUENCE TESTING

Given access to the intermediate representation of an image belonging to class $k$ and the direction of a concept $i$ in the considered latent space, RCAV Pfau et al. (2020) quantifies the image-level concept-sensitivity score by perturbing the representation towards the concept's direction and measuring the difference in the network's output probability for class $k$, before and after the perturbation. A dataset-wide sensitivity score in the range $[-1, 1]$ is subsequently calculated, with zero indicating inconsistent use of the concept for predictions of the class, while values near the extremes indicating consistent negative or positive concept contributions. Finally, due to the fact that the network may demonstrate non-zero sensitivity along random directions of the feature space, a statistical significance test is performed where the sensitivity of the network towards the direction of the concept is compared against the sensitivity scores across a set of random directions. In this paper we use the term *directions of significant influence*, whenever the direction passes the latter statistical significance test. Apparently, to accurately measure the concept's-sensitivity score, a reliable estimate of the concept's direction is required. Previous works, including Pfau et al. (2020) and Kim et al. (2018), would consider using classifier filter directions for this purpose. Recent research though (Pahde et al. (2024)), suggests that the concept's direction should be modeled using an estimator of the concept's signal direction, such as pattern-CAVs. For a more detailed description regarding RCAV the reader may refer to section A.8.

## 3 METHOD

For a given network layer, the input to our method is the feature representations of images coming from a *concept dataset*, an image collection from the domain on which the network was trained. We build on the foundations of unsupervised interpretable direction learning Doumanoglou et al. (2023; 2024) under the lens of a multi-concept signal-distractor data model. First, we learn a set of linear classifiers (also called concept detectors), $\boldsymbol{W}, \boldsymbol{b}$ using the objectives discussed in section 2.3 with the aim of interpretability. Yet, under the new perspective of signal-distractor theory, we lift constraints of prior work regarding the orthogonality of the filters and standardization of the feature space, to allow for a more flexible clustering of representations. Those constraints though that were previously there, would consist implicit regularizers to avoid degenerate solutions in the direction search. Thus, removing those restrictions requires addressing the gap that they leave behind, by introducing the additional loss terms that we discuss in section 3.1. Beyond this, we make a significant advancement, by proposing learnable *signal vectors* $\hat{\boldsymbol{s}}_i$ as concept signal estimators under the newly introduced multi-concept signal-distractor theoretical model. Finally, we propose Uncertainty Region Alignment, a loss term that aligns the subspace of network's uncertain predictions with the subspace of uncertain concept detections. The previously mentioned signal vectors are learned jointly with the concept detectors in an end-to-end fashion, and through the Uncertainty

Region Alignment loss, affect the quality of the clustering while at the same time exhibit significant influence on the network's predictions. An overview of the approach is depicted in Fig. 1.

### 3.1 INTERPRETABILITY LOSSES TO RECOVER IMPLICIT REGULARIZATIONS

We propose **Self-Weighted Reduction** ($\mathcal{R}_{SW}$) as a loss aggregation method to optimize upper bounds. Consider a set of un-reduced loss values $\mathcal{L}_k, k \in \mathbb{N}$. The Self-Weighted Reduction is:

$$\mathcal{R}_{SW}(\{\mathcal{L}_k\}) = \frac{\sum_k \mathcal{L}_k^{\nu+1}}{\sum_k \mathcal{L}_k^{\nu}} \tag{3}$$

which is equal to the weighted average of elements in $\{\mathcal{L}_k\}$ with each element being weighted by $\mathcal{L}_k^{\nu}, \nu > 1, \nu \in \mathbb{R}^+$ a sharpening factor. This loss may be seen as a soft-differentiable version of the max operation, since the largest value in the set of $\{\mathcal{L}_k\}$, is weighted with the largest weight.

#### 3.1.1 EXCESSIVELY ACTIVE CLASSIFIER LOSS ($\mathcal{L}^{eac}$)

This loss term penalizes clusters with a large population, to avoid degenerate cases like the fulfillment of a sparse solution where all input representations are assigned to a single cluster. This loss term depends on a hyper-parameter $\rho$ (in a similar fashion as it is done in sparse autoencoders Ng et al. (2011)) that indicates an upper bound on the portion of pixels in the dataset that may be classified positively by any classifier $i \in \{0, 1, ..., I-1\}$ in the set. The formula of the un-reduced $\mathcal{L}_i^{eac}$ is provided below, with $\gamma > 1, \gamma \in \mathbb{R}^+$ a sharpening factor and the denominator $1 - \rho$ normalizing the loss in range $[0, 1]$:

$$\mathcal{L}_i^{eac} = \frac{1}{1 - \rho} \text{ReLU}(\mathbb{E}_{\boldsymbol{p}}[y_{\boldsymbol{p},i}^{\gamma}] - \rho) \tag{4}$$

The final reduced loss, is using $\mathcal{R}_{SW}$: $\mathcal{L}^{eac} = \mathcal{R}_{SW}(\{\mathcal{L}_i^{eac}\})$

#### 3.1.2 SPARSITY BOUND LOSS ($\mathcal{L}^{sb}$)

With this loss term we minimize the upper bound of the un-reduced $\mathcal{L}^s$, among pixel locations, using $\mathcal{R}_{SW}$. In more detail, if $\mathcal{L}_{\boldsymbol{p}}^s$ (2) denotes the Sparsity Loss for pixel $\boldsymbol{p}$, the Sparsity Bound Loss ($\mathcal{L}^{sb}$) is defined as $\mathcal{L}^{sb} = \mathcal{R}_{SW}(\{\mathcal{L}_{\boldsymbol{p}}^s\})$

### 3.2 MULTI-CONCEPT SIGNAL-DISTRACTOR DATA MODEL

We propose an extended signal - distractor data model for modeling the latent space which considers the encoding of multiple concepts. Let each spatial element $\boldsymbol{x_p}$ to be a linear combination of latent concept signals $\boldsymbol{S} \in \mathbb{R}^{D \times I}$ and distractors $\boldsymbol{D} \in \mathbb{R}^{D \times F}, F \leq D - I$:

$$\boldsymbol{x_p} = \boldsymbol{S a_p} + \boldsymbol{D \beta_p} \tag{5}$$

with $\boldsymbol{a_p} \in \mathbb{R}^I$ and $\boldsymbol{\beta_p} \in \mathbb{R}^F$. $\boldsymbol{S}$ is a matrix of $I \in \mathbb{N}^+$, $D$-dimensional, unit-norm, concept signal directions and $\boldsymbol{D}$ a matrix denoting a basis for distractor components in the data. Each one of the signal directions contains information regarding the presence of a distinct concept. For the individual signal values $a_{\boldsymbol{p},i}$ (the $i$-th element of $\boldsymbol{a_p}$) and the distractor coefficients $\boldsymbol{\beta_{p,f}}$ we make the same assumptions as in section 2. Additionally, we make a sparsity assumption in this data model, in which $\boldsymbol{x_p}$ is only a member of a single concept from the set of possible concepts.

### 3.3 SIGNAL VECTORS AS CONCEPT SIGNAL ESTIMATORS

Let us now consider the set of $I$ concept detectors $\boldsymbol{w}_i$ that we learn according to section 2.3. Our aim for them is to serve a dual purpose. First, to act as classifiers that are able to classify whether a representation belongs to their detected concept. Second, we want to base their decision on the value of the concept's signal in the data, essentially acting as the filter regressors discussed in section 2.2. For the first objective, the losses introduced in section 2.3 and in section 3.1 are sufficient. For the second, the weight vector $\boldsymbol{w}_i$ needs to be orthogonal to all $\boldsymbol{s}_j, j \neq i$ and the subspace of distractors $\boldsymbol{D}$. This extends the previously discussed conditions for the binary concept case of section 2.2. Since we require $\boldsymbol{w}_i^T \boldsymbol{s}_j = 0$ we need to have an estimate of $\boldsymbol{s}_j$. As we discuss in section A.3,

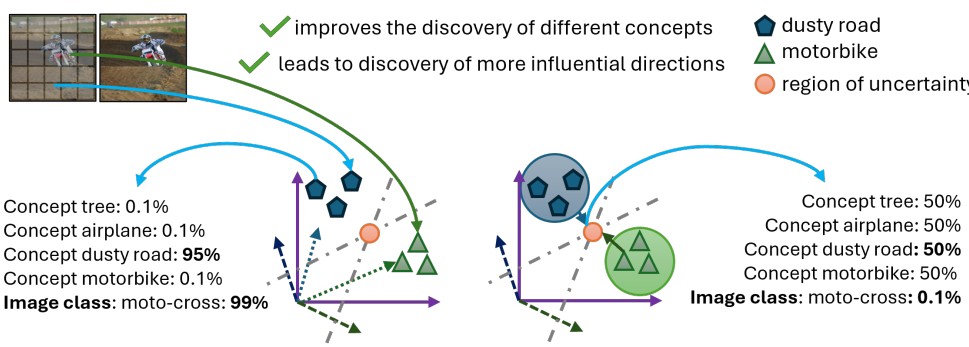

Figure 2: The uncertainty region of the network is defined as the subspace where all network's predictions are maximally uncertain. The uncertainty region of the concept detectors is defined as the intersection of all their decision hyperplanes. Aligning these two through feature manipulation improves the interpretability and influence of the concept directions.

under the assumption of sparse feature-to-cluster assignments that we consider in this work, for this purpose we are able to use (1) if, for the variance and covariance terms, we only consider positive samples of the concept instead of positive and negative samples that was suggested in prior work. We refer to the signal estimator for concept $i$ that is learned with these conditions as *signal vector* $\hat{s}_i$. According to the previous discussion, we use the following Filter-Signal Orthogonality Loss when we learn the directions:

$$\mathcal{L}^{fso} = \sqrt{\mathbb{E}_{i,j}\left[(1-\delta_{i,j})\bar{w}_i^T \bar{s}_j)^2\right]}$$

with $\delta_{i,j}$ the kronecker delta and $\bar{w}, \bar{s}$ denoting the L2-normalized filter weights and signal vectors. While for the value of the extracted signal to be exactly accurate we would require that $w_i$ are also perpendicular to the basis of distractors, we do not explicitly model distractors in this work. Instead, to address the previous shortcoming and at the same time being faithful to their use by the underlying network, we supervise signal vectors to align with influential directions through the Uncertainty Region Alignment loss of section 3.4.

### 3.4 UNCERTAINTY REGION ALIGNMENT TO IMPROVE INTERPRETABILITY AND INFLUENCE

The presence or absence of a concept in a representation may provide neutral, positive or negative evidence against the prediction of a class. However, when we learn the concept directions, we do not know the association between concept-class pairs. Despite that, we are motivated to leverage the knowledge embedded within the studied network to facilitate the discovery of latent space concept directions, by exploiting the intuitive fact that uncertain predictions of the network should be made when the representation does not contain confident information for the presence or absence of concepts. Thus, we propose that direction search can be enhanced by alignining two subspaces: a) the uncertainty region of the network and b) the uncertainty region of the concept detectors. Specifically, we define the *uncertainty region of the network* as the subspace where the network's predictions are maximally uncertain and the *uncertainty region of the concept detectors* as the intersection of all their decision hyperplanes. Fig. 2 illustrates the concept of Uncertainty Region Alignment.

To quantify the alignment, we first manipulate all spatial feature representations $x_p$ in the direction $-dx_p$ to become $x'_p = x_p - dx_p$. The direction $dx_p$ is chosen in a way that the shifted $x'_p$ lies on the intersection of the estimated concept detectors' decision hyperplanes. This is done based on our current estimate of $w_i$, $b_i$ and $\hat{s}_i$. Subsequently, we require that the network's predictions for the manipulated features are maximally uncertain, effectively aligning the two uncertainty regions.

UNCONSTRAINED AND CONSTRAINED UNCERTAINTY REGION LOSSES ($\mathcal{L}^{uur}$, $\mathcal{L}^{cur}$)

We define two types of Uncertainty Region Loss: i) unconstrained $\mathcal{L}^{uur}$ and ii) constrained $\mathcal{L}^{cur}$. Each loss uses a different feature manipulation strategy $dx_p$ but both share the same final formula

$$\mathcal{L}^{uur} = \mathcal{L}^{cur} = -\mathbb{E}_{X'}\left[\mathcal{H}(f^+(X')\right] \tag{6}$$

(with $\mathcal{H}$ denoting entropy). In (6), $\boldsymbol{X}'$ denotes a manipulated image representation, with every $\boldsymbol{x_p}$ shifted in the direction $-\boldsymbol{dx_p}$.

**i) Unconstrained Uncertainty Region Manipulation** Supposing that we know the latent space's interpretable directions $\boldsymbol{w}_i$ and the classification offsets $b_i$, we can bring all $\boldsymbol{x_p}$ to the concept detectors' uncertainty region by manipulating each $\boldsymbol{x_p}$ in the direction $-\boldsymbol{dx_p}$ with the following formula:

$$\boldsymbol{w}_i^T \boldsymbol{x_p'} - b_i = 0 \Rightarrow \quad \boldsymbol{w}_i^T (\boldsymbol{x_p} - \boldsymbol{dx_p}) - b_i = 0, \, \forall i, \Rightarrow$$
$$\boldsymbol{W}^T (\boldsymbol{x_p} - \boldsymbol{dx_p}) - \boldsymbol{b} = 0 \Rightarrow \quad \boldsymbol{dx_p} = (\boldsymbol{W}^T)^+ (\boldsymbol{W}^T \boldsymbol{x_p} - \boldsymbol{b})$$

with $\boldsymbol{A}^+$ denoting the pseudo inverse of $\boldsymbol{A}$.

**ii) Constrained Uncertainty Region Manipulation** In the previous case, features were brought to the concept detectors' uncertainty region without any constraints, taking into account only the filter directions $\boldsymbol{w}_i$. However, as suggested in Pahde et al. (2024), an accurate manipulation needs to take into account the signal directions within the data. Motivated by this fact in this second manipulation variant, we constrain the manipulation of the features to be done in the span of the signal vectors, i.e. $\boldsymbol{dx_p} = \hat{\boldsymbol{S}} \boldsymbol{v}$, $\boldsymbol{v} \in \mathbb{R}^I$, and thus:

$$\boldsymbol{W}^T (\boldsymbol{x_p} - \boldsymbol{dx_p}) - \boldsymbol{b} = 0 \Rightarrow \boldsymbol{W}^T (\boldsymbol{x_p} - \hat{\boldsymbol{S}} \boldsymbol{v}) - \boldsymbol{b} = 0 \Rightarrow \boldsymbol{W}^T \hat{\boldsymbol{S}} \boldsymbol{v} = \boldsymbol{W}^T \boldsymbol{x_p} - \boldsymbol{b} \Rightarrow$$
$$\boldsymbol{v} = (\boldsymbol{W}^T \hat{\boldsymbol{S}})^+ (\boldsymbol{W}^T \boldsymbol{x_p} - \boldsymbol{b}) \Rightarrow \boldsymbol{dx_p} = \hat{\boldsymbol{S}} \boldsymbol{v} = \hat{\boldsymbol{S}} (\boldsymbol{W}^T \hat{\boldsymbol{S}})^+ (\boldsymbol{W}^T \boldsymbol{x_p} - \boldsymbol{b})$$

where $\hat{\boldsymbol{S}}$ represents a matrix whose $i$-th column is equal to the estimated signal vector $\hat{\boldsymbol{s}}_i$.

## 4 EXPERIMENTS

### 4.1 EXPERIMENT ON SYNTHETIC DATA

In this section, we seek to validate the effectiveness of the proposed method for identifying concept directions within synthetic data. We assume that the spatial dimensions of the image representation space are: width $W = 2$ and height $H = 1$, resulting in two spatial elements within the layer, $\boldsymbol{p}_1, \boldsymbol{p}_2$. We set the embedding space dimensionality to $D = 6$, the number of distinct concepts to $I = 3$, and the size of the distractor basis to $F = 2$.

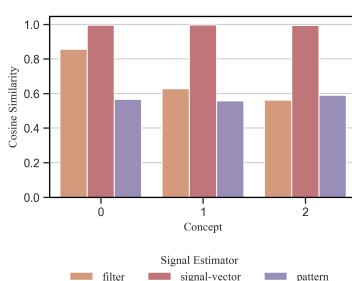

Figure 3: Cosine Similarity of ground-truth concept signal directions with the estimated signal using three estimators: filter, signal-vector and pattern-CAV. Cosine similarities for filter and signal-vector estimators. Only the proposed signal vectors were able to estimate the ground-truth signal directions.

Based on the assumptions of section 3.2, we generate features $\boldsymbol{x_p}$, each one to correspond to a single concept, which implies that each image patch is associated with a single **concept class** (e.g., *bus*, *car*, *road*, etc.). From a representational standpoint, this indicates that the component $a_{\boldsymbol{p},i}$ is significant whenever the patch belongs to concept $i$, and resembles random noise otherwise. Let $\boldsymbol{c}(\boldsymbol{p}) \in \{0, 1, 2\}$ denote the concept of pixel $\boldsymbol{p}$. Furthermore, we define synthetic **image class labels** $k \in \{a, b, c\}$, which are defined by "images" with the following properties: For $k = a$: $\boldsymbol{c}(\boldsymbol{p}_1) = 0$ and $\boldsymbol{c}(\boldsymbol{p}_2) = 1$, for $k = b$: $\boldsymbol{c}(\boldsymbol{p}_1) = 0$ and $\boldsymbol{c}(\boldsymbol{p}_2) = 2$ and for $k = c$: $\boldsymbol{c}(\boldsymbol{p}_1) = 1$ and $\boldsymbol{c}(\boldsymbol{p}_2) = 2$. To make a concrete example, images of label $k = a$, *parking*, could be the ones having one image patch of concept *car* ($i = 0$) and one image patch of concept *road* ($i = 1$), etc.

To create the synthetic data, we first randomly generate unit-norm vectors to construct the matrices $\boldsymbol{S}$ and $\boldsymbol{D}$ of (5); for their specific values, refer to section A.10. Then, to generate a pixel representation $\boldsymbol{x_p}$, we choose the $i$-th element of $\boldsymbol{a_p}$ and the $f$-th element of $\boldsymbol{\beta_p}$ according to:

$$a_{\boldsymbol{p},i} \sim \begin{cases} \mathcal{N}(\theta_\mu, \theta_{\sigma^2}), \text{when } \boldsymbol{c}(\boldsymbol{p}) = i \\ \mathcal{N}(\omega_\mu, \omega_{\sigma^2}), \text{otherwise} \end{cases} , \quad \beta_{\boldsymbol{p},f} \sim \mathcal{N}(\omega_\mu, \omega_{\sigma^2}) \tag{7}$$

with a specific choice for $\theta_\mu = 10$, $\theta_{\sigma^2} = 3$, $\omega_\mu = 2.5$ and $\omega_{\sigma^2} = 1$. We generate a balanced dataset with each class being represented by 1000 "images".

The network that we use is comprised of just two layers (corresponding to the top part of a potentially larger convolutional network). The first being an average-pooling layer and the second, a linear layer with $K = 3$ output classes. After training (section A.10), the network attains 100% accuracy on a test set, generated randomly based on the previous principles.

With the proposed method, we can cluster representations based on their semantic concept and recover the underlying latent concept signal directions. Due to the unsupervised nature of the method, we assign a concept label to each one of the concept detectors based on their Intersection over Union (IoU) performance when classifying pixels associated with any of the concepts (section 2.4). All the learned concept-detectors exhibit an exceptional ability to distinguish pixels from different concepts, with IoU of 1 (thus, the learned filters fulfill the interpretability property, see also Table 8). The cosine similarity between the learned signal vectors $\hat{s}_i$ and the ground-truth concept signal directions $s_j, j \in \{0, 1, 2\}$ are depicted in Fig. 3. The learned signal vectors align with the true signal direction of each concept, as evidenced by their high cosine similarity with the ground-truth signal direction corresponding to their assigned concept label (and thus signal vectors accurately captured the signal directions within the data). Detailed comparison with other failing signal estimators, including the learned filters and Pattern-CAVs are provided in the same figure (For a discussion on how the signal estimation affects concept sensitivity evaluation with RCAV and details on the ability of the learned concept detectors to extract the true signal value from the representations see also section A.10).

## 4.2 Experiment on Deep Image Classifier

We further assess the various components of the method through a real-world experiment. Unless otherwise noted, we conduct evaluation on the last convolutional layer of a deep image classifier, specifically ResNet18 (He et al. (2016)), trained on Places365 (Zhou et al. (2017)). Again, if not mentioned otherwise, our proposed Interpretable and Influential Directions (IID) is using a weighted combination (Table 13) of all the interpretability losses of sections 2.3 and 3.1 in addition to $\mathcal{L}^{cur}$ and $\mathcal{L}^{fso}$ from sections 3.3 and 3.4 and the hyperparameter $I$ is set to $I = 500$. For direction

Table 1: Experimental results w.r.t interpretability loss components. Semantic Segmentation Interpretability Metrics: $\mathcal{S}^1$, $\mathcal{S}^2$ and Influence Metrics: Significant Direction Count (SDC) and Significant Class-Direction Pairs (SCDP).

| | $\mathcal{S}^1$ | $\mathcal{S}^2$ | SDC | SCDP |
|---|---|---|---|---|
| $\mathcal{L}^{uur}$ | **59.06** | 25.91 | **377** | 2487 |
| $\mathcal{L}^{uur}+\mathcal{L}^{sb}$ | 49.02 | 35.23 | 354 | 2480 |
| $\mathcal{L}^{uur}+\mathcal{L}^{sb}+\mathcal{L}^{eac}$ | 54.55 | 37.38 | 359 | 2118 |
| $\mathcal{L}^{cur}+\mathcal{L}^{sb}+\mathcal{L}^{eac}+\mathcal{L}^{fso}$ | 57.34 | **38.36** | 376 | **3271** |

learning and evaluation of interpretability, we adhere to the protocol established in Doumanoglou et al. (2023). In particular, after learning the directions without labels, we employ Network Dissection (Bau et al. (2017)) with the Broden concept dataset, which contains dense pixel annotations for 1197 concepts over ∼63K images and 5 concept categories, to assign a label to each learned concept detector.

Regarding interpretability evaluation, we use the following two interpretability metrics introduced in Doumanoglou et al. (2023). Specificallym let $\phi_i(c, \mathcal{K})$ denote the function of Intersection Over Union when using classifier $i$ in detecting concept $c$ over the concept dataset $\mathcal{K}$. Let $c_i^\star = \text{argmax}_c \phi_i(c, \mathcal{K}_{train})$, i.e. the label in the training split of the concept dataset ($\mathcal{K}_{train}$) that can be detected better from classifier $i$. If $\mathcal{K}_{val}$ denotes the validation split of the concept dataset, the two interpretability scores $\mathcal{S}^1$ and $\mathcal{S}^2$ that we use are defined as:

$$\mathcal{S}^1 = \int_0^1 \sum_{i=0}^{I-1} \mathbb{1}_{x \geq \xi} \big(\phi_i(c_i^*, \mathcal{K}_{val})\big) d\xi, \quad \mathcal{S}^2 = \int_0^1 |\{c_i^\star \,|\, \exists\, i : \phi_i(c_i^\star, \mathcal{K}_{val}) \geq \xi\}| d\xi \quad (8)$$

The first metric $\mathcal{S}^1$, counts the number of concept detectors with a score better than a threshold $\xi$. In the second metric $\mathcal{S}^2$, $|.|$ denotes cardinality of the set and counts the number of unique concept labels that can be detected by the concept detectors with IoU more than $\xi$. In both cases, the scores are made threshold agnostic, by integrating across all $\xi \in [0, 1]$.

Regarding influence evaluation, we use RCAV Pfau et al. (2020). For sensitivity scores we are constructing Concept Activation Vectors (CAVs) by replicating each one of the signal vectors or pattern-CAVs along all spatial dimensions. For direction significance testing, we use RCAV's label permutation test with the significance threshold set to 0.05 and Bonferroni correction. Further details are provided in section A.12.

In the table results, we use two summarizing metrics, namely Significant Direction Count (SDC) and Significant Class-Direction Pairs (SCDP). SDC represents the number of learned signal vectors that significantly influence at least one of the model's classes, while SCDP counts the total number of class-direction pairs in which the learned signal vector significantly affects the class. In both cases, significant directions are considered the ones that pass the RCAV's direction significance test. A qualitative explanation obtained with directions learned with the proposed method is provided in Fig. 4 and more of them are provided in sections A.13 and A.12.2.

**Ablation on interpretability losses** In Table 1, we perform an ablation study on the set of interpretability loss terms introduced in this paper. Experimental results indicate that the combination of all introduced loss terms—$\mathcal{L}^{uur}$, $\mathcal{L}^{sb}$, and $\mathcal{L}^{eac}$—yields more interpretable directions. While this combination may underperform in terms of influence compared to using $\mathcal{L}^{uur}$ alone, the addition of signal vectors significantly enhances the influential impact (last row of Table 1). At the same time, the use of the interpretability loss terms maintains the high interpretability of the learned directions.

**Ablation on uncertainty region alignment losses** Table 2 summarizes the metric scores in relation to uncertainty region alignment. The most influential directions are learned through the combination of $\mathcal{L}^{cur}$ and $\mathcal{L}^{fso}$, as evidenced by the high influence metrics. This combination also achieves the highest score in terms of $\mathcal{S}^1$. However, while it scores relatively high for $\mathcal{S}^2$, it is not the optimal combination compared to other candidates. Despite that, we consider it the best combination as it exhibits an excellent balance between interpretability and influence.

Table 2: Experimental results on ablation w.r.t Uncertainty Region Alignment losses. Semantic Segmentation Interpretability Metrics: $\mathcal{S}^1$, $\mathcal{S}^2$ and Influence Metrics: Significant Direction Count (SDC) and Significant Class-Direction Pairs (SCDP). The number of concept detectors are set to $I = 450$.

| | $\mathcal{S}^1$ | $\mathcal{S}^2$ | SDC | SCDP |
|---|---|---|---|---|
| $\mathcal{L}^{uur}$ | 50.49 | 34.76 | 283 | 1451 |
| $\mathcal{L}^{cur}$ | 50.94 | **36.01** | 335 | 2930 |
| $\mathcal{L}^{cur}+\mathcal{L}^{fso}$ | **52.63** | 34.86 | **360** | **2956** |

**Interpretability comparison with previous unsupervised approaches.** We evaluate the proposed $\mathcal{L}^{uur}$ against prior unsupervised methods in two networks: Resnet18 trained on Places365 and Resnet50 trained on Moments In Time Monfort et al. (2019), using the exact setup of Doumanoglou et al. (2024) (i.e. without lifting the orthogonality of the directions, the feature standardization, or considering signal directions). The experimental results are provided in Table 3. The proposed $\mathcal{L}^{uur}$ increased the interpretability of the directions up to +78.78% in $\mathcal{S}^2$. (More details in section A.11)

**Interpretability comparison with a supervised approach.** We compare classifiers learned using the proposed method with those learned via a supervised approach Zhou et al. (2018), focusing on interpretability. For each concept detector, we calculate the binary classification metrics, Precision, Recall, F1 Score, Average Precision (AP), and the IoU segmentation metric. These metrics are averaged across detectors to obtain mean values (mPrecision, mRecall, etc.). For Zhou et al. (2018), directions are learned for the labels identified by Network Dissection, ensuring a fair comparison. Three variants of our method are considered: a) individual directions learned directly; b) combining directions with the same label using a linear layer, which classifies $x_p$ positively if any detector in the set of detectors with the same label classifies it positively (denoted as *Linear-OR* in Table 5); and c) individual directions learned with our method, but with the threshold $b_i$ learned in a supervised manner to optimize F1 Score. This last approach assesses direction quality indepen-

Table 4: Influence Comparison against Pattern-CAVs. Since all $\mathcal{S}^3$ scores are below 0.5 Pattern-CAVs are *not* more influential than the proposed signal vectors.

| RCAV $\alpha$ | 0.5 | 2.0 | 5.0 |
|---|---|---|---|
| $\mathcal{S}^3$ | 0.37 | 0.37 | 0.38 |

Table 3: Comparison with prior work on unsupervised basis learning. Works considered: Unsupervised Interpretable Basis Extraction (UIBE Doumanoglou et al. (2023)), Concept-Basis-Extraction (CBE Doumanoglou et al. (2024)) and Concept-Basis-Extraction with CNN Classifier Loss replaced with the proposed $\mathcal{L}^{uur}$ (CBE /w $\mathcal{L}^{uur}$). Significantly more interpretable directions are obtained for Resnet50.

| | Resnet18 / Places365 | | | Resnet50 / MiT | | |
|---|---|---|---|---|---|---|
| | UIBE | CBE | CBE /w $\mathcal{L}^{uur}$ | UIBE | CBE | CBE /w $\mathcal{L}^{uur}$ |
| $\mathcal{S}^1$ | 60.93 (+0.0%) | **69.43 (+13.95%)** | 67.3 (+10.45%) | 124.73 (+0.0%) | 131.73 (+5.61%) | **158.76 (+27.28%)** |
| $\mathcal{S}^2$ | 28.39 (+0.0%) | 31.53 (+11.06%) | **32.16 (+13.28%)** | 18.47 (+0.0%) | 26.94 (+45.86%) | **33.02 (+78.78%)** |

Table 5: Comparison of concept-detectors' performance in pixel classification and image segmentation tasks. Comparing between: a) individual classifiers, b) combined classifiers *linear-or* c) individual classifiers with their thresholds learned with supervision, and d) IBD: a set of classifiers learned in a supervised way.

|  | mPrecision | mRecall | mAP | mF1Score | $\mathcal{S}^1$ | $\mathcal{S}^2$ | mIoU |
|---|---|---|---|---|---|---|---|
| IBD Zhou et al. (2018) | 0.84 | 0.6 | 0.77 | 0.69 | 53.32 | 53.32 | 0.20 |
| IID Individual (ours) | 0.81 | 0.24 | 0.53 | 0.33 | 57.33 | 38.35 | 0.11 |
| IID Linear-OR (ours) | 0.73 | 0.4 | 0.59 | 0.45 | 30.56 | 30.56 | 0.11 |
| IID Individual /w sup thresholds (ours) | 0.62 | 0.49 | 0.52 | 0.53 | N/A | N/A | N/A |

dent of bias. As shown in Table 5, a) the individual classifiers from our method achieve high precision, comparable to those from supervised learning. However, b) Recall improves significantly when combining classifiers with the same label using a linear layer (*Linear-OR*). Lastly, c) learning the classifier bias in a supervised manner (IID /w sup thresholds) shows that relaxing sparsity (through a reduced bias) also improves F1 Scores.

**Influence comparison with pattern-CAVs.** Although Pattern-CAVs did not capture the true signal direction in our synthetic experiment, they remain the best signal estimators from prior work. Here, we compare the network's sensitivity to Pattern-CAVs versus signal vectors. Let $j \in \{0, 1, ..., N_l - 1\}$ represent an index for classifiers with the same concept label $l$, and let $S_{j,k}^l$ denote the RCAV sensitivity score of class $k$ with respect to the signal vector of the $j$-th concept detector for label $l$. Similarly, $S_{P,k}^l$ is the sensitivity score of class $k$ with respect to the Pattern-CAV for the same label.

Pattern-CAVs are learned with ground-truth labels at the pixel-level. Since Network Dissection may assign the same concept label to multiple concept detectors, a direct comparison of signal vectors with Pattern-CAVs is not feasible. Therefore, inspired by the RCAV approach, we treat signal vectors as "noise vectors" against whom we compare the sensitivity of Pattern-CAVs for a given label. We define a metric where a value greater than 0.5 indicates that Pattern-CAVs have more influence on the network's output than the proposed signal vectors at the significance level $\theta = 0.05$ (with Bonferroni correction):

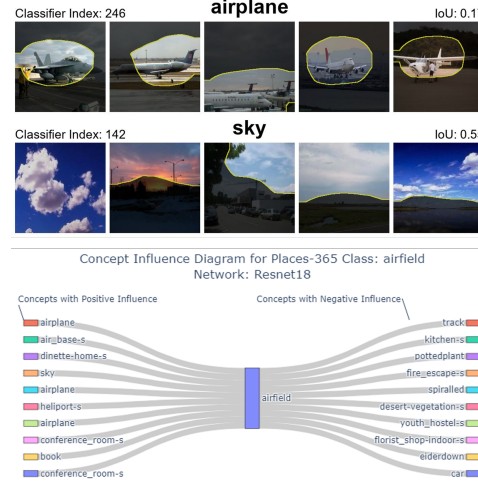

Figure 4: Explanation obtained when using Network Dissection and RCAV with directions learned with IID.

$$\mathcal{S}^3 = \mathbb{E}_{l,k}\left[\mathbb{1}(p_{l,k} < \frac{\theta}{N_l})\right], \quad p_{l,k} = \frac{1}{N_l}\sum_j \mathbb{1}\left(|S_{j,k}^l| \geq |S_{P,k}^l|\right) \tag{9}$$

$\mathcal{S}^3$ metrics for varying RCAV's hyper-parameter $\alpha$, are given in Table 4. Overall, $\mathcal{S}^3$ is lower than 0.5, meaning that, pattern-CAVs do not influence network's predictions more than signal vectors.

## 5 CONCLUSION

We presented a novel unsupervised method for discovering both interpretable and influential directions in the latent space of deep networks. To the best of our knowledge, this is the first approach that extends the concept of signal directions in a multi-label setting, offering a way to recover these directions in more complex cases. Although the data model proposed in this study represents an advancement over previous models that focus on binary labels, further analysis is required using real-world datasets. Our experiments demonstrated the method's ability to recover true signal directions in the data, but certain limitations remain — particularly when distinct signal directions are nearly co-linear in the presence of strong distractor noise. Overall, this work opens new avenues for understanding latent spaces in deep networks especially as it easily fits with existing literature on explainability and model correction.

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

## A APPENDIX

### A.1 EXTENDED RELATED WORK

**Direction Learning and Explainability** The typical approach to interpretable direction learning considers a linear model and a dataset with concept annotations. The most prominent works in this line are Zhou et al. (2018) and Kim et al. (2018). Zhou et al. (2018) proposed interpretable basis decomposition of class weight vectors, by projecting them to the learned interpretable directions. This projection allows the decomposition of network predictions to concept heatmaps, combining local explanations with high level concepts. Kim et al. (2018) and Pfau et al. (2020) utilize Concept Activation Vectors (CAVs) that represent interpretable concept directions in the latent space of deep networks. These directions can be used to quantify each concept's influence on the network's predictions, providing a global perspective on its decision-making process. In an attempt to address the annotation costs, Ghorbani et al. (2019) automates concept annotation by leveraging features from pre-trained image classifiers, though it does not explicitly propose a method for discovering concept directions. The pseudo-labels generated by this method can still be employed to learn interpretable directions, as in Kim et al. (2018). Different than previous approaches, Schrouff et al. (2022) combines a local interpretability method Sundararajan et al. (2017) with the global explanation approach of Kim et al. (2018). This effectively combines local attributions, with interpretable latent space directions for improved local and global explainability. Recently, Pahde et al. (2024) proposed Pattern-CAVs, an enhanced method for modeling concept directions in the latent space, that considers signal and distractor parts as we do. Building on previous work Kindermans et al. (2017); Haufe et al. (2014), Pattern-CAVs provide more accurate concept influence explanations and can serve as a replacement for CAVs in Kim et al. (2018); Pfau et al. (2020). In contrast to previous supervised approaches, non-negative Concept Activation Vectors (NCAVs) have been proposed Zhang et al. (2021) as an unsupervised alternative to learn interpretable directions directly from the structure of the feature space using non-negative matrix factorization. However, this approach has the limitations of non-negativity of direction weights and limited expressivity stemming from the lack of additional bias. Fundamentally different than all previous approaches is the unsupervised approach that has been proposed more recently, that learns an interpretable basis (directions) without annotations. Instead of providing pseudo-labels for learning the directions as in Ghorbani et al. (2019), the work of Doumanoglou et al. (2023; 2024) exploits a sparsity property of interpretability, removing the need of costly annotations and approaching the direction discovery from the perspective of the network. Additionally, their work does not suffer from the limitations of Zhang et al. (2021). Our work expands on their line of research and has the potential to fit in places where CAVs or Pattern-CAVs are utilized.

**Directions and Model Correction** Lapuschkin et al. (2019); Anders et al. (2022); Pahde et al. (2023) identify artifact directions, or potential shortcuts, that deep networks may exploit. They propose augmenting class representations by either inducing or suppressing these artifact directions in a fine-tuning correction step. Alternatively, their method can suppress the propagation of the

artifact's contribution to the final prediction by manipulating the features in the negative artifact direction, without requiring further fine-tuning. Dreyer et al. (2024) takes a different fine-tuning approach and addresses model correction by penalizing the gradient in the direction of the artifacts. While these works have already proposed ways to identify the artifact directions that may be used to correct networks, our method can also fit as an unsupervised alternative.

**Other Applications of Interpretable Directions** Concept Bottleneck Models (CBMs) have been widely explored as a way to enhance model interpretability by associating specific concepts with model decisions Koh et al. (2020); Xu et al. (2024); Oikarinen et al. (2023); Sheth & Ebrahimi Kahou (2023); Lai et al. (2023). These models aim to make the decision process of deep networks more transparent by mapping latent representations to human-understandable concepts. In an attempt to enable post-hoc explainability for any classifier, recently, in Post-Hoc CBMs Yuksekgonul et al. (2023), a concept subspace (comprised by interpretable latent space directions) is learned in a supervised way using the approach in Kim et al. (2018). This subspace, together with a learned interpretable predictor, can turn any deep network to an interpretable classifier. While in Yuksekgonul et al. (2023) the concept subspace is learned in a supervised manner, the hereby proposed approach could offer an unsupervised alternative. In a novel strategy for identifying failure modes, Jain et al. (2023) leverages directions in the embedding space of CLIP Radford et al. (2021), where both text and images with the same semantic meaning are aligned. Finally, interpretable directions have also been considered in the context of Generative Adversarial Networks (GANs Yang et al. (2021); Voynov & Babenko (2020); Bounareli et al. (2022)), where disentangling concept directions in GANs' latent space allows meaninful image interventions and fine-grained image synthesis.

### A.2 LIMITATIONS OF CLASSICAL FEATURE DECOMPOSITION METHODS

Below we list some theoretical arguments on why classical matrix decomposition approaches might fail to recover the signal directions within the data and why we expect them to also fail in our experiment on synthetic data:

1) **Principal Component Analysis - PCA** assumes that latent components (signals and distractors) are orthogonal. For the potential of this to work there should be an additional orthogonality assumption across all signals and distractors. This is a rather a strict assumption which we do not have empirical evidence to be true. Even if this assumption is satisfied, then a second stronger assumption should be made about the latent space: concepts shall be encoded in directions of decreasing variance, for which, again to our knowledge, there is not such evidence. PCA is guaranteed to fail to solve our toy-experiment on synthetic data since the signal directions in that example are not orthogonal (The reader may also refer to Table 11 for cosine similarities between signals and distractors in that experiment).

2) **Independent Component Analysis - ICA** assumes independence for all $\alpha_{p,i}$ and $\beta_{p,f}$. By assumption, in our data model, a distractor component ($\beta_{p,f}$) is indeed considered independent of signal components or other distractors. Yet, the signal components $a_{p,i}$ are not. More specifically the $\mathrm{Cov}[a_{p,i}, a_{p,j}]$ is expected to be negative since the presence of concept $i$ in $x_p$ implies non-presence of all the other concepts $j$. Since two independent variables have covariance zero, and we identified that $\mathrm{Cov}[a_{p,i}, a_{p,j}]$ is $< 0$, we suspect that the solution to our problem does not lie in ICA's solution space.

3) **Dictionary Learning with sparsity constraints** is also fundamentally different than our approach in the following way: in Dictionary Learning, sparsity constraints are enforced in the units of latent variables. Yet, our proposed data model implies sparsity in the *semantic space*, i.e. the space of concepts defined as:

$$\mathbf{c_p} = \sigma(\mathbf{W x_p} - \mathbf{b}) \in [0, 1]^I$$

where $\mathbf{W}$ summarizes the filter directions of the concept detectors and $\mathbf{b}$ summarizes the concept detectors' biases. The latter assumption is a more accurate assumption to make than enforcing sparsity in the units of feature space. Provided we do have access to $\mathbf{W}$ we recover the concepts' signal directions using (1). In other words, in a Sparse dictionary representation it is implied that the representation has a zero coefficient for some of the concepts and distractors, which is a stronger assumption than ours.

4) **Non-negative Matrix Factorization - NMF** assumes that all the components of the signal matrix $S$ and distractor matrix $D$ to be positive. Again, this is a rather strict requirement and we expect that

this approach will not be always able to recover the latent directions. For instance, in our synthetic experiment there do exist negative components in $S$ and $D$, as depicted in Table 11, which once again implies that this method would not be able to identify them.

### A.3 SIGNAL DIRECTION ESTIMATION

In the original theoretical formulation defined in Kindermans et al. (2017), the data model is defined as $x_p = s + d$, with $s$ denoting the signal and $d$ denoting a distractor component. To extract the value $z_p$ of the signal from the representation, a filter $w$ may be applied to $x_p$: $z_p = w^T x_p = w^T s$, since $w^T d = 0$, as the filter needs to cancel-out the distractor. All pixels $p$ in this formulation share the same signal and distractor components, and according to the theory, the signal direction can be estimated by exploiting the fact that $\text{cov}[d, z_p]$ should be $0$, leading to the equation for the estimated signal direction $\hat{s}$:

$$\hat{s} = \frac{\text{cov}[x_p, z_p]}{\sigma_z^2} \tag{10}$$

Thus, it is important to highlight that for the theory to hold, in particular $\text{cov}[d, z_p] = 0$, all pixels $p$ considered in (10) need to share the same signal and distractor components. To link Kindermans et al. (2017) with our case, a filter extracting the concept component $i$ from the representation $x_p$, sees all other signal directions $s_j$ and basis $D$ as distractors. Thus, for two distinct concepts $i$ and $j$, the filters $w_i$ and $w_j$ "see" different signal and distractor directions. According to the assumptions in Kindermans et al. (2017), distractor components in the features should not have any predictive power over the value of the signal. To highlight the issue, consider a feature space with 2 signal directions, i.e. each feature is $x_p = \alpha_{p,1} s_1 + \alpha_{p,2} s_2$. Now consider two sets $\mathcal{C}^1 = \{x_p : c(x_p) = 1\}$ and $\mathcal{C}^2 = \{x_p : c(x_p) = 2\}$, with $c(x_p)$ denoting the concept of $x_p$. From the scope of a filter extracting the signal value of concept 1, $s_2$ is seen as distractor. Thus, $\text{cov}[d, z_p] = \text{cov}[\alpha_{p,2} s_2, z_p] = s_2 \text{cov}[\alpha_{p,2}, \alpha_{p,1}]$. Considering only the pixels of $\mathcal{C}^1$, the variables $\alpha_{p,1}$ and $\alpha_{p,2}$ are independent by assumption, leading to the, expected, zero covariance. However, in case we consider $x_p$ to lie in the joint set $\mathcal{C} = \mathcal{C}^1 \cup \mathcal{C}^2$, then this independence does not hold, as the distributions of $\alpha_{p,1}$ and $\alpha_{p,2}$ are negatively correlated, and the covariance is not zero, violating the assumptions of (10) for recovering the signal direction. Thus, in our formulation we use (10), but only considering the pixels of the concept, completely disregarding concept negative samples.

### A.4 UNSUPERVISED INTERPRETABLE BASIS EXTRACTION AND CONCEPT-BASIS EXTRACTION LOSSES

**Sparsity Loss** ($\mathcal{L}^s$) Doumanoglou et al. (2023)

Based on the observation that the number of semantic labels that may be attributed to an image's patch, are only a fraction of the set of possible semantic labels, this loss enforces sparsity across the classification results $y_{p,i}$ for each spatial representation $x_p$. In particular, the sparsity loss for pixel $p$ is defined as:

$$\mathcal{L}_p^s = -\sum_i q_{p,i} \log_2 q_{p,i}, \quad q_{p,i} = \frac{y_{p,i}}{\sum_i y_{p,i}} \tag{11}$$

and the aggregated sparsity loss $\mathcal{L}^s$:

$$\mathcal{L}^s = \mathbb{E}_p[\mathcal{L}_p^s] \tag{12}$$

**Maximum Activation Loss** ($\mathcal{L}^{ma}$) Doumanoglou et al. (2023)

With the complement of this loss the pixel classifications are enforced to become binary:

$$\mathcal{L}^{ma} = \mathbb{E}_p\left[-\sum_i q_{p,i} \log_2 y_{p,i}\right] \tag{13}$$

**Inactive Classifier Loss** ($\mathcal{L}^{ic}$) Doumanoglou et al. (2024)

This loss ensures that each classifier in the set, classifies positively at least $\nu \in [0, 1]$ percent of pixels in the concept dataset.

$$\mathcal{L}^{ic} = \mathbb{E}_i\Big[\frac{1}{\nu}\text{ReLU}\big(\nu - \mathbb{E}_{\boldsymbol{p}}[y_{\boldsymbol{p},i}^{\gamma}]\big)\Big] \tag{14}$$

with $\nu = \frac{\tau}{I}, \gamma > 1, \gamma \in \mathbb{R}^+$ denoting a sharpening factor and $\tau \in [0, 1]$ denoting a percent of pixels in the dataset to be evenly distributed among the $I$ classifiers in the set.

### A.5 MAXIMUM MARGIN LOSS ($\mathcal{L}^{mm}$)

In the original formulation of Doumanoglou et al. (2023), the Maximum Margin Loss was defined as $\mathcal{L}^{mm} = \frac{1}{M}$ with $M$ being a single parameter for the whole set of classifiers since the optimization was performed in the standardized space with shared parameters for the margins $M$ and biases $b$. In this work, we removed the standardized space constraints and instead, we have a margin parameter $M_i$ for each classifier in the set. Thus, we modify the Maximum Margin loss to become:

$$\mathcal{L}^{mm} = \frac{1}{I}\sum_i \frac{1}{M_i} \tag{15}$$

### A.6 DATASET EXPLANATION LOSS ($\mathcal{L}^{de}$)

In the following, whenever $\exists i : y_{\boldsymbol{p},i} > 0.5$, we say that pixel $\boldsymbol{p}$ is **explained by the set of classifiers** and whenever $\nexists i : y_{\boldsymbol{p},i} > 0.5$, we say that $\boldsymbol{p}$ is **not explained** by this set. Let $\lambda_{\boldsymbol{p}}$ denote the **pixel explanation coefficient (PEC)** which is defined as $\lambda_{\boldsymbol{p}} = \max_i y_{\boldsymbol{p},i}$. Let also $\mu \in [0, 1]$ denote a target percent of pixels in the dataset that we aim to explain, with $N$ the total number of pixels in the batch. Then, the Dataset Explanation Loss (DEL) provides a penalty, whenever the set of classifiers does not explain enough pixels in the dataset:

$$\mathcal{L}^{de} = \frac{1}{\mu}\text{ReLU}(\mu - \frac{1}{N}\sum_{\boldsymbol{p}} \lambda_{\boldsymbol{p}}) \tag{16}$$

With $\frac{1}{\mu}$ a constant that normalizes $\mathcal{L}^{de}$ to lie in range $[0, 1]$. We modify the loss reduction across pixels in the batch in $\mathcal{L}^s$ and $\mathcal{L}^{ma}$ to be a weighted average with weights equal to each pixel explanation coefficient $\lambda_{\boldsymbol{p}}$ (See (17) and (18) below). Whenever $\mu < 1$, PEC allows the method to totally ignore $1 - \mu$ portion of the pixels in the dataset without any penalty on $\mathcal{L}^s$ or $\mathcal{L}^{ma}$. $\mathcal{L}^{de}$ ensures that optimization does not collapse to a degenerate solution where all $\lambda_{\boldsymbol{p}}$ equal zero. Instead, it enforces the optimization process to converge in solutions that explain at least the $\nu$ percent of pixels in the dataset.

**Sparsity Loss ($\mathcal{L}^s$) with PEC**

$$\mathcal{L}^s = \frac{1}{\sum_{\boldsymbol{p}} \lambda_{\boldsymbol{p}}}\Big[-\lambda_{\boldsymbol{p}}\sum_i q_{\boldsymbol{p},i}\log_2 q_{\boldsymbol{p},i}\Big] \tag{17}$$

**Maximum Activation Loss ($\mathcal{L}^{ma}$) with PEC**

$$\mathcal{L}^{ma} = \frac{1}{\sum_{\boldsymbol{p}} \lambda_{\boldsymbol{p}}}\Big[-\lambda_{\boldsymbol{p}}\sum_i q_{\boldsymbol{p},i}\log_2 y_{\boldsymbol{p},i}\Big] \tag{18}$$

Ablation experimental results with respect to the hyper-parameter $\mu$ are provided in Table 6.

### A.7 BASELINE INFLUENTIAL DIRECTION LOSS ($\mathcal{L}^{id}$)

Here we consider an alternative baseline for Uncertainty Region Alignment. A more straightforward approach to find influential directions, can manipulate each representation $\boldsymbol{x_p}$ to become $\boldsymbol{x_p^i}' = \boldsymbol{x_p} + \delta\hat{\boldsymbol{s}}_i, \delta \in \mathbb{R}^+$ and require that after the manipulation, the prediction's probability distribution is significantly different than the distribution before manipulation (i.e. explicitly optimizing for

Table 6: Experimental results w.r.t dataset explanation threshold $\mu$. Semantic Segmentation Interpretability Metrics: S1, S2 and Influence Metrics: Significant Direction Count (SDC) and Significant Class-Direction Pairs (SCDP). The number of concept detectors are set to $I = 450$, and uncertainty region alignment loss terms is set to $\mathcal{L}^{cur} + \mathcal{L}^{fso}$.

| $\mu$ | $\mathcal{S}^1$ | $\mathcal{S}^2$ | SDC | SCDP |
|---|---|---|---|---|
| 0.8 | 48.01 | 35.66 | 335 | **3014** |
| 0.9 | **52.63** | 34.86 | 360 | 2956 |
| 1.0 | 51.8 | **36.15** | **362.0** | 2900 |

Table 7: Experimental results on ablation w.r.t the influence losses of the method (left) and concept-detector count $I$ (right). Semantic Segmentation Interpretability Metrics: $\mathcal{S}^1$, $\mathcal{S}^2$ and Influence Metrics: Significant Direction Count (SDC) and Significant Class-Direction Pairs (SCDP). On the left, number of concept detectors are set to $I = 450$. On the right, uncertainty region alignment loss is set to $\mathcal{L}^{cur} + \mathcal{L}^{fso}$.

| | $\mathcal{S}^1$ | $\mathcal{S}^2$ | SDC | SCDP | I | $\mathcal{S}^1$ | $\mathcal{S}^2$ | SDC | SCDP |
|---|---|---|---|---|---|---|---|---|---|
| $\mathcal{L}^{uur}$ | 50.49 | 34.76 | 283 | 1451 | 100 | 15.74 | 13.57 | 80 | 617 |
| $\mathcal{L}^{cur}$ | 50.94 | 36.01 | 335 | 2930 | 150 | 21.45 | 19.13 | 114 | 1002 |
| $\mathcal{L}^{cur} + \mathcal{L}^{fso}$ | **52.63** | 34.86 | **360** | **2956** | 250 | 29.59 | 23.26 | 176 | 1233 |
| $\mathcal{L}^{id}$ | 49.96 | 36.2 | 332 | 2526 | 350 | 39.2 | 29.14 | 254 | 1453 |
| $\mathcal{L}^{id} + \mathcal{L}^{fso}$ | 50.32 | 36.21 | 326 | 2543 | 450 | 52.63 | 34.86 | 360 | 2956 |
| $\mathcal{L}^{cur} + \mathcal{L}^{id} + \mathcal{L}^{fso}$ | 51.52 | **36.28** | 320 | 2205 | 500 | **57.34** | **38.36** | **376** | **3271** |

influence). Let $f^+(\boldsymbol{X}^{i'}) \in (0,1)^K$ denote the part of the network that follows the layer of study and produces a probability vector for $K$ classes, where $\boldsymbol{X}^{i'}$ denotes a manipulated image representation with all spatial elements shifted in the direction of the $i$-th signal vector. Then, this objective is enforced through KL-Divergence:

$$\mathcal{L}^{id} = -\mathbb{E}_{(\boldsymbol{X}, i)}\big[ D_{\mathrm{KL}}\big( f^+(\boldsymbol{X}^{i'}) || f^+(\boldsymbol{X}) \big) \big] \tag{19}$$

In Table 7 we provide an ablation study which uses this loss as a baseline against which we compare $\mathcal{L}^{cur}$ and $\mathcal{L}^{uur}$ from section 3.4.

## A.8 CONCEPT INFLUENCE TESTING WITH RCAV

RCAV Pfau et al. (2020), is a global explainability method that we use in our experiments and we describe it here to make the reader familiar with the terms concept *sensitivity* and concept *significant influence* and highlight the importance of why the estimation of the concept's signal direction is important. Supposing that we are studying the latent space of a given network's intermediate layer, let $\boldsymbol{v}_i$ denote the, unit-norm, direction of concept $i$ in this latent space and $\boldsymbol{X}$ denote the representation of an image which belongs to class $k$. RCAV aims to answer questions like the following "To what extend does the concept $i$ influence the prediction of class $k$ ?". The answer to this question we first, define the image-level *concept sensitivity* score as follows:

$$s_{i,k}(\boldsymbol{X}) = f^{+,k}(\boldsymbol{X} + \alpha \boldsymbol{v}_i) - f^{+,k}(\boldsymbol{X})$$

with $f^{+,k}$ the upper part of the network above the studied layer which predicts the probability of class $k$ and $\alpha \in \mathbb{R}^+$ a feature perturbation hyper-parameter. This equation measures the difference in the prediction outcome for class $k$ when we add "more of concept $i$" to the image representation. Second, we compute the dataset-wide concept sensitivity score as:

$$S_{i,k} = -1 + 2 \frac{1}{|\mathbb{X}_{val}^k|} \sum_{\boldsymbol{X} \in \mathbb{X}_{val}^k} \mathbb{1}\big( s_{i,k}(\boldsymbol{X}) \geq 0 \big)$$

with $\mathbb{X}_{val}^k$ denoting a validation dataset split with images of class $k$. This score measures how consistently the model uses concept $i$ for the prediction of class $k$. Scores near zero indicate concept irrelevance with respect to the prediction of class $k$ while values near the extremes $\{+1, -1\}$ indicate consistent positive or negative contribution. Due to the fact that even random vectors drawn from the latent space yield non-zero sensitivity scores, finally, RCAV concludes with a third step which measures robustness through the following statistical significance test: $N$ random noise vectors $\{\boldsymbol{u}_n\}$ are drawn from the latent space (there is a bit more in this, see Pfau et al. (2020)) and

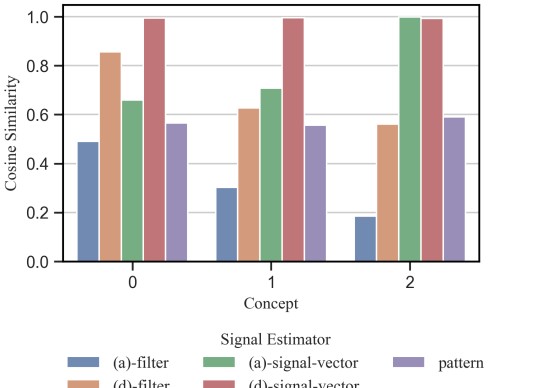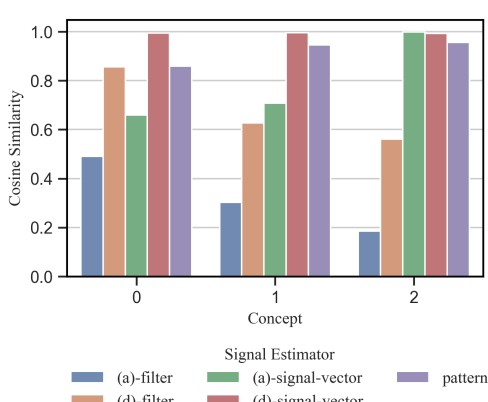

Figure 5: **Left**: Cosine Similarity of ground-truth concept signal directions with the estimated signal using three estimators: filter, signal-vector and pattern-CAV. Cosine similarities for filter and signal-vector estimators for both steps (a) and (d) are provided. Recovering the true signal directions within the data, not only depends on the estimator but also on the training objectives, as only the signal vectors of step (d) are able to recover all the signal directions. **Right**: Cosine Similarity of ground-truth concept signal directions with the estimated signal using three estimators: filter, signal-vector and a preliminary approach to fixing pattern-CAVs (Pahde et al. (2024)) for the multi-label case. The proposed preliminary approach significantly improves the alignment of Pattern-CAVs with the ground-truth signal directions.

subsequently the following p-value is calculated for concept $i$:

$$p = \frac{1}{N} \sum_n \mathbb{1}(|S_{n,k}| \geq |S_{i,k}|)$$

where for the calculation of $S_{n,k}$ we use $\boldsymbol{u}_n$. After applying Bonferroni correction to the resulting p-values, the impact of concept $i$ to the prediction of class $k$ is considered *significant*, whenever the resulting p-value is low enough, based on the overall acceptable significance theshold - usually $0.05$ - and the adjustment of Bonferroni correction.

To accurately measure the concept-sensitivity score, apparently, a reliable estimate of the concept's direction $\boldsymbol{v}_i$ is required. Previous works, including Pfau et al. (2020) and Kim et al. (2018), would consider using classifier filter directions in the place of $\boldsymbol{v}_i$. Recent research though (Pahde et al. (2024)), suggests that the direction $\boldsymbol{v}_i$ should be modeled using an estimator of the concept's signal direction, such as pattern-CAVs.

## A.9 DIRECTION LEARNING PROCESS

As outlined in Section 3, we extend the approach of Doumanoglou et al. (2024) by relaxing the orthogonality and feature standardization constraints, while enhancing the direction search with loss functions introduced in Sections 3.1, 3.3, and 3.4. However, experiments on both synthetic and real-world data revealed that direct optimization with these changes does not converge. To resolve this, we implement a four-step process: a) we first learn the parameters $\boldsymbol{w}_i, b_i$ following Doumanoglou et al. (2024), replacing the *CNN Classifier Loss* with our $\mathcal{L}^{uur}$; b) we then continue optimizing $\boldsymbol{w}_i, b_i$, removing the orthogonality and standardization constraints while incorporating the additional losses from section 3.1; c) next, we learn the signal vectors from the classifiers obtained in the previous step to initialize $\{\hat{\boldsymbol{S}}\}$; and d) finally, we jointly optimize $\hat{\boldsymbol{s}}_i, \boldsymbol{w}_i, b_i$, using all previous losses, replacing $\mathcal{L}^{uur}$ with $\mathcal{L}^{cur}$ and/or $\mathcal{L}^{id}$, and adding $\mathcal{L}^{fso}$ from Sections 3.3 and 3.4.

## A.10 DETAILS FOR THE EXPERIMENT ON SYNTHETIC DATA

We train the network using cross-entropy loss and the Adam Kingma (2014) optimizer, with learning rate $0.005$ and batch size $1024$ for $2000$ epochs. In principle, we follow the process defined in section A.9, but due to the simplicity of the example, we omit step (b) and directly proceed from (a) to (c). For the same reasons, when performing step (d), we exclude $\mathcal{L}^{sb}$, as convergence is possible

Table 8: Results for the experiment on synthetic data. Binary segmentation performance of the learned concept detectors in distinguishing samples from the 3 concepts (IoU). Cosine similarity scores (Cos-Sim) between the concept-detectors' signal vector and the ground truth signal direction of each concept. While both steps have the same perfect classification performance in identifying each one of the concept samples, only step (d) recovers the true signal directions within the data accurately.

| | | | Step (a) | | | | | | | Step (d) | | | |
|---|---|---|---|---|---|---|---|---|---|---|---|---|---|
| | | | IoU | | | Cos-Sim | | | IoU | | | Cos-Sim | |
| | | | Concept-Detectors | | | | | | Concept-Detectors | | | | |
| | | 0 | 1 | 2 | 0 | 1 | 2 | 0 | 1 | 2 | 0 | 1 | 2 |
| Concepts | 0 | **1** | 0 | 0 | **0.59** | 0.51 | -0.41 | **1** | 0 | 0 | **0.99** | 0.48 | 0.27 |
| | 1 | 0 | 0 | **1** | -0.49 | 0.76 | **0.67** | 0 | 0 | **1** | 0.24 | 0.77 | **0.99** |
| | 2 | 0 | **1** | 0 | -0.28 | **0.99** | 0.14 | 0 | **1** | 0 | 0.48 | **0.99** | 0.76 |

Table 9: Results for the experiment on synthetic data when excluding $\mathcal{L}^{fso}$ from step (d). Binary classification performance of the learned concept detectors in distinguishing samples from the 3 concepts (Jaccard Index - (IoU)). Cosine similarity scores (Cos-Sim) between the concept-detectors' signal vector and the ground truth signal direction of each concept. When not using $\mathcal{L}^{fso}$, the learned classifiers do not capture the true signal within the data, except concept-detector 1.

| | | | Step (d) | | | | |
|---|---|---|---|---|---|---|---|
| | | | IoU | | | Cos-Sim | |
| | | | Concept-Detectors | | | | |
| | | 0 | 1 | 2 | 0 | 1 | 2 |
| Concepts | 0 | **1** | 0 | 0 | **0.59** | 0.53 | -0.44 |
| | 1 | 0 | 0 | **1** | -0.49 | 0.74 | **0.64** |
| | 2 | 0 | **1** | 0 | -0.28 | **0.99** | 0.09 |

even without it. For step (a), we learn interpretable directions using Adam and the learning rate linearly increasing from zero to $0.005$ in $5000$ epochs. For step (d), we learn concept directions using the same optimizer and learning rate scheme, but for a period of $10000$ epochs. For the batch sizes we use the largest number that our GPU allows. We linearly combine the introduced loss terms with weights $\lambda^n$ where $n$ the respective loss name (for their values see Table 10). In both learning steps, in order to avoid local minima, we use an adaptive loss weighting scheme, which varies across iterations, for the weights $\lambda^{ma}$ and $\lambda^{mm}$:

$$\lambda^{ma} = \begin{cases} 0, \mathcal{L}^{ma} < 0.8 \\ 2.7, \text{otherwise} \end{cases} \qquad \lambda^{mm} = \begin{cases} 0, \mathcal{L}^{s} > 0.1, \text{or } \mathcal{L}^{fso} > 0.1 \text{ (when applicable)} \\ 0.6, \text{otherwise} \end{cases}$$

and keep the direction set that attains the lowest loss score, satisfying the constraints $\mathcal{L}^{s} < 0.1, \mathcal{L}^{ma} < 0.8$ and $\mathcal{L}^{fso} < 0.1$ (when applicable). The specific values of matrices $\boldsymbol{S}$ and $\boldsymbol{D}$ used in this experiment, and the cosine similarities between every pair of vectors are provided in Table 11. Detailed IoU scores of the learned concept detectors and cosine similarity scores between ground-truth signal directions and various signal estimators for both learning steps (a) and (d) are provided in Table 8 and Fig. 5 (Left). We also provide cosine-similarity scores for step (d) when not using $\mathcal{L}^{fso}$ in Table 9.

For Pattern-CAVs we use difference of means between samples depicting the concept and negative samples, as proposed in Pahde et al. (2024). Yet, this approach is expected to fail in the non-binary concept regime introduced in section 2. This is due to the fact that for the difference of means to work, all considered samples in the formulas of Kindermans et al. (2017); Haufe et al. (2014), should share the same signal and distractor parts and from the scope of a filter extracting the signal value of a concept, other concept signal directions are also being seen as distractors. According to Kindermans et al. (2017); Haufe et al. (2014), a distractor component should not leak information regarding the value of the signal. However, in the non-binary - multiple concept case, the presence of a concept in a representation indicates negative correlation between this direction and the directions of other concepts, which contradicts the objective that distractors do not have predictive power over the considered signal direction.

Table 10: Loss weights used for the experiment on Synthetic Data. For *adaptive* entries, refer to the text.

| | $\lambda^s$ | $\lambda^{ma}$ | $\lambda^{mm}$ | $\lambda^{ic}$ | $\lambda^{uur}$ | $\lambda^{cur}$ | $\lambda^{eac}$ | $\lambda^{fso}$ |
|---|---|---|---|---|---|---|---|---|
| step (a) | 2.6 | adaptive | adaptive | 5.0 | 0.25 | - | - | - |
| step (d) | 2.6 | adaptive | adaptive | 15.0 | - | 0.25 | 15.0 | 5.0 |

Table 11: Left: Data matrices $S$ and $D$ for the experiment on synthetic data. Right: Cosine similarities for every pair of vectors in $S$, $D$, i.e.: $C^T C$, $C = [S|D]$.

| S | | | D | | Cosine-Similarities | | | | |
|---|---|---|---|---|---|---|---|---|---|
| 0.4497 | -0.8870 | 1.0000 | -0.5484 | -0.6822 | 1.0 | 0.3070 | 0.5392 | -0.0392 | -0.0481 |
| -0.7530 | 0.9700 | 0.9630 | 0.3860 | -0.6720 | 0.3070 | 1.0 | 0.7616 | -0.1667 | 0.6893 |
| -0.1560 | 0.7053 | 0.2769 | -0.6970 | 0.2621 | 0.5392 | 0.7616 | 1.0 | -0.2274 | 0.6269 |
| 0.7937 | 0.4342 | 0.7818 | 0.1838 | 0.3192 | -0.0392 | -0.1667 | -0.2274 | 1.0 | 0.0103 |
| 0.3488 | 0.4440 | 0.1076 | -0.3150 | 0.3057 | -0.0481 | 0.6893 | 0.6269 | 0.0103 | 1.0 |
| -0.1268 | 0.3299 | 0.5396 | -0.7146 | 0.5159 | | | | | |

A possible preliminary approach to overcome this, could be based on the analytical formula introduced in Pahde et al. (2024). We may only consider samples containing a concept (disregarding negative samples), and exclude the bias term $b$. The formula to calculate the signal directions would become:

$$h_i : \min ||A - \mathbf{1}h_i^T||_2 \tag{20}$$

where $\mathbf{1} \in \mathbb{R}^D$ a vector of ones, $h_i$ denotes the learned Pattern-CAV for concept $i$, $A \in \mathbb{R}^{N \times D}$ with $N$ denoting the number of spatial elements $x_p$ for concept $i$ and $D$ the embedding dimension. Thus $A$ is a matrix summarizing the features of concept $i$.

For the experiment on synthetic data, we solved for $h_i$ using iterative optimization, using the Adam optimizer for 25000 epochs and learning rate 0.001. The resulting Pattern-CAVs had a significantly higher cosine similarity scores with the ground truth signal directions as depicted in Figure 5 (Right).

Although this preliminary approach may solve this toy problem, a more thorough study is essential for feature work. When comparing against Pattern-CAVs in the experiment on the Deep Image Classifier, we still use the original proposition introduced in Pahde et al. (2024), i.e. calculate pattern directions with the difference of cluster means.

Finally, we also measure the ability of each concept-detector in extracting the true value of the concept's signal in the data. We treat each concept-detector as a linear feature extractor, excluding the sigmoid non-linearity and directly apply the weight vectors on the pixel representation: $z_{p,i} = w_i^T x_p$. Feature extractors from step (a) demonstrate root mean squared error (RMSE) of **7.8**, while those from step (d) achieve RMSE **0.99**.

COMPARING CONCEPT-CLASS SENSITIVITY SCORES FOR THE EXPERIMENT ON SYNTHETIC DATA FOR VARIOUS RCAV $\alpha$

In Figures 6,7,8,9,10,11 we present RCAV sensitivity scores for the experiment on synthetic data for various values of RCAV's hyper-parameter $\alpha$ and for different signal estimators, including filter, signal-vector or Pattern-CAVs Pahde et al. (2024).

In RCAV evaluation, we construct a Concept Activation Vector by replicating the estimated signal directions (either filter, pattern or signal vector) across spatial locations. Compared to Pattern-CAVs, the sensitivity scores that are obtained with the learned signal vectors, more closely resemble the ground-truth sensitivity. This may be attributed to the higher cosine similarity between the signal vectors and the ground truth directions compared to Pattern-CAVs (according to Fig. 3).

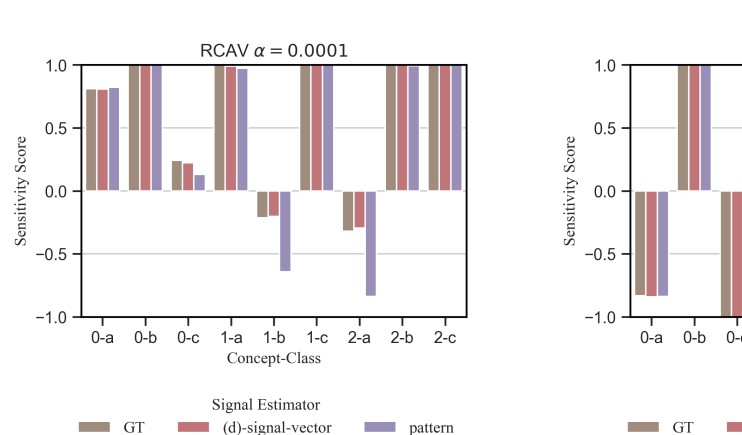

Figure 6: Concept-Class sensitivity scores for the experiment on synthetic data, computed using RCAV Pfau et al. (2020) for different choices on RCAV's hyper-parameter value $\alpha$. GT: sensitivity computed using the ground-truth signal direction. (d)-signal-vector: sensitivity computed using the signal vectors of step (d). pattern: sensitivity computed using pattern-CAV Pahde et al. (2024) with ground-truth labeled representations. The signal vectors of step (d) more accurately approximate the ground-truth sensitivity scores of each concept-class pair. For other values of $\alpha$ see section A.10 The sensitivity score is rescaled to $[-1, 1]$.

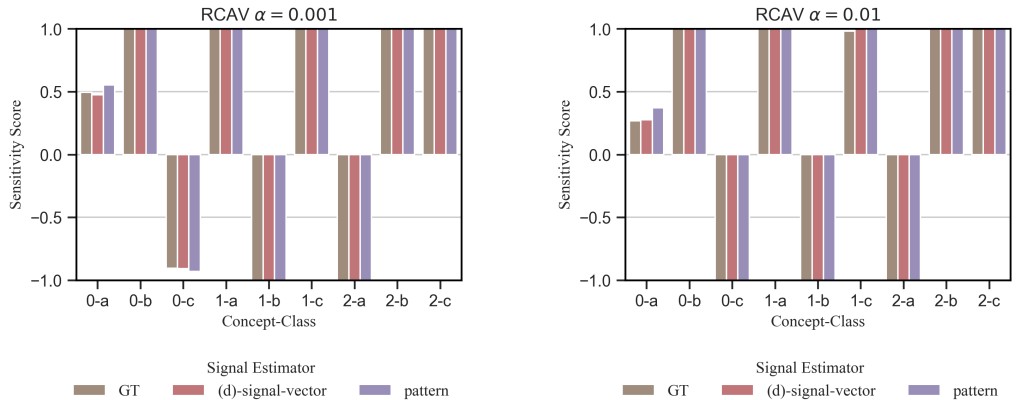

Figure 7: Concept-Class sensitivity scores for the experiment on synthetic data, computed using RCAV Pfau et al. (2020) for different choices on RCAV's hyper-parameter value $\alpha$. Compared to pattern-CAVs, the sensitivity scores that are obtained using the proposed *signal* vectors are closer to the sensitivity scores that are obtained using the ground-truth concept directions.

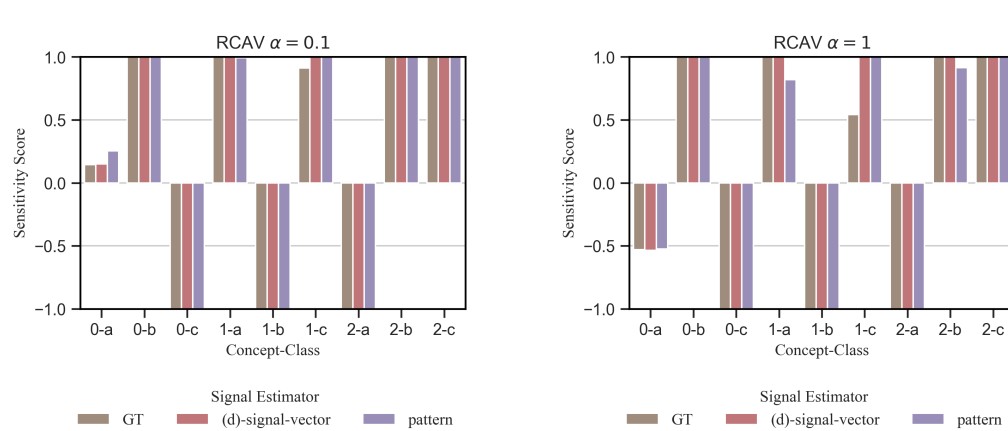

Figure 8: Concept-Class sensitivity scores for the experiment on synthetic data, computed using RCAV Pfau et al. (2020) for different choices on RCAV's hyper-parameter value $\alpha$. Compared to pattern-CAVs, the sensitivity scores that are obtained using the proposed *signal* vectors are closer to the sensitivity scores that are obtained using the ground-truth concept directions.

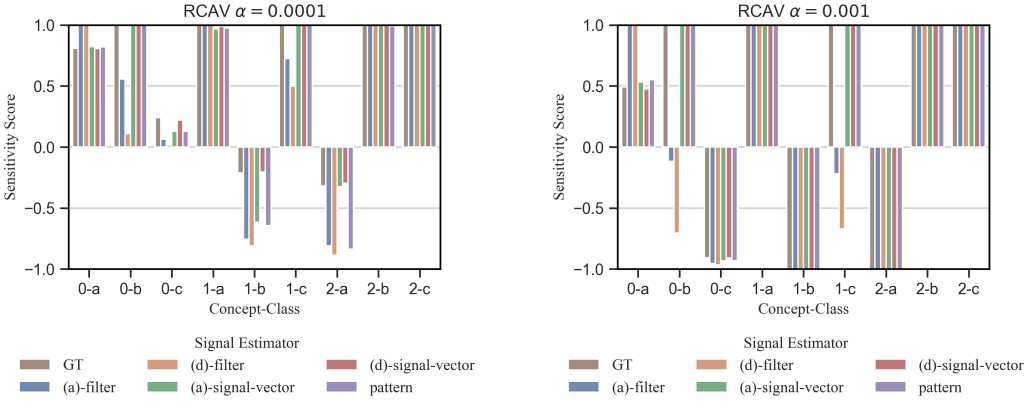

Figure 9: Concept-Class sensitivity scores for the experiment on synthetic data, computed using RCAV Pfau et al. (2020) for different choices on RCAV's hyper-parameter value $\alpha$. This figure compares against different ways to estimate each concept's *signal* direction: (a)-filter, (d)-filter, (a)-*signal* vector and (d)-*signal* vector and pattern-CAVs Pahde et al. (2024). Apart from pattern-CAVs, the rest of the estimators correspond to filter directions obtained from steps (a) and (d) or *signal* vectors computed using 1 for the same steps. Compared to other estimators, the sensitivity scores that are obtained using the *signal* vectors of step(d) are closer to the sensitivity scores that are obtained using the ground-truth concept directions.

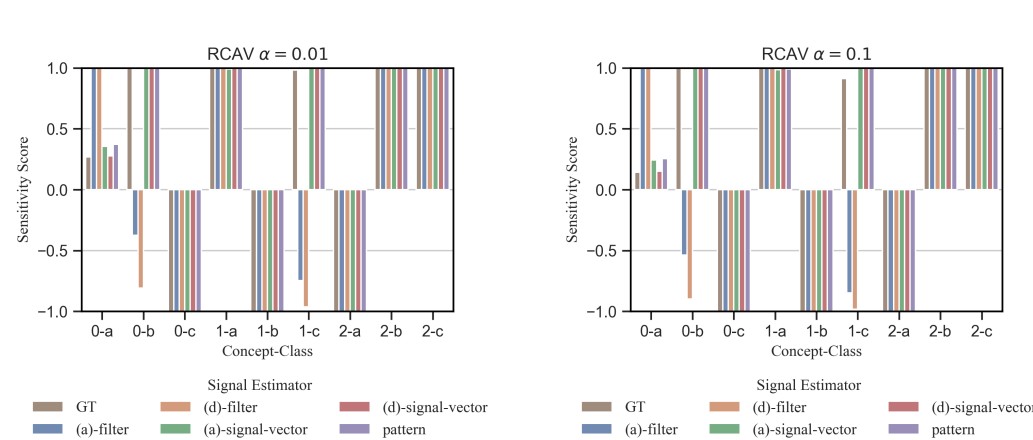

Figure 10: Concept-Class sensitivity scores for the experiment on synthetic data, computed using RCAV Pfau et al. (2020) for different choices on RCAV's hyper-parameter value $\alpha$. This figure compares against different ways to estimate each concept's *signal* direction: (a)-filter, (d)-filter, (a)-*signal* vector and (d)-*signal* vector and pattern-CAVs Pahde et al. (2024). Apart from pattern-CAVs, the rest of the estimators correspond to filter directions obtained from steps (a) and (d) or *signal* vectors computed using 1 for the same steps. Compared to other estimators, the sensitivity scores that are obtained using the *signal* vectors of step(d) are closer to the sensitivity scores that are obtained using the ground-truth concept directions.

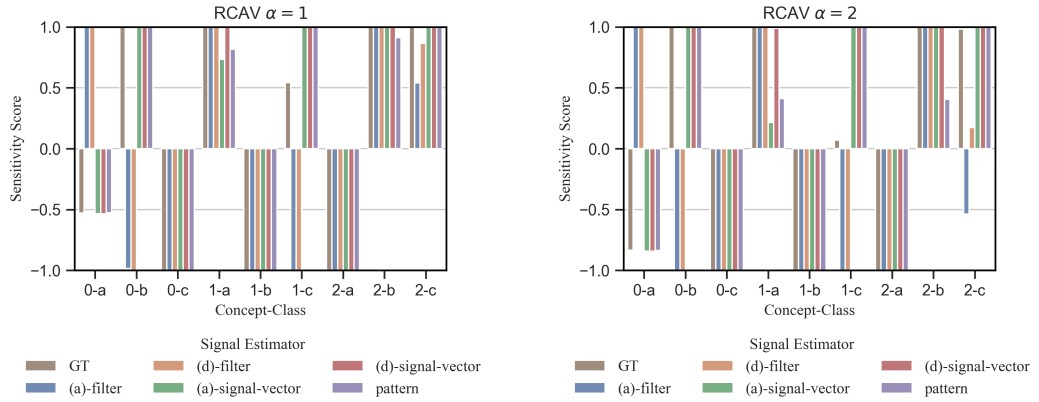

Figure 11: Concept-Class sensitivity scores for the experiment on synthetic data, computed using RCAV Pfau et al. (2020) for different choices on RCAV's hyper-parameter value $\alpha$. This figure compares against different ways to estimate each concept's *signal* direction: (a)-filter, (d)-filter, (a)-*signal* vector and (d)-*signal* vector and pattern-CAVs Pahde et al. (2024). Apart from pattern-CAVs, the rest of the estimators correspond to filter directions obtained from steps (a) and (d) or *signal* vectors computed using 1 for the same steps. Compared to other estimators, the sensitivity scores that are obtained using the *signal* vectors of step(d) are closer to the sensitivity scores that are obtained using the ground-truth concept directions.

## A.11 Interpretability Comparison of $\mathcal{L}^{uur}$ with prior work

In an attempt to discover more interpretable directions, Doumanoglou et al. (2024), made a first attempt to exploit the knowledge encoded in the network, through feature manipulation. In particular, it was suggested that representations $x_p$ should be manipulated towards the concept detectors' hyperplanes, **only for the concepts that are present in $x_p$** (i.e. manipulating towards the **negative direction** of the filter weights, when those filters classify $x_p$ positively). Features **were not manipulated** in the direction of the weights (to bring them towards the separating hyperplane, when they lie in the subspace of negative classification), essentially suggesting that the network's predictions should be highly uncertain when for all $x_p$, none of the classifiers makes positive predictions (without minding about confident negative predictions). This is fundamentally different with the proposition in the present work, which manipulates **all features towards the hyperplanes**, regardless of whether features were positively or negatively classified, suggesting that the network's predictions should be maximally uncertain when for all $x_p$ none of the classifiers makes confident predictions, either positive **or negative**. Additionally, in Doumanoglou et al. (2024), signal directions were completely overlooked.

We evaluate the proposed $\mathcal{L}^{uur}$ against prior work in two networks: Resnet18 trained on Places365 Zhou et al. (2017) and Resnet50 trained on Moments In Time Monfort et al. (2019), using the exact setup of Doumanoglou et al. (2024) (i.e. without lifting the orthogonality of the directions, the feature standardization, or considering signal directions). For $\mathcal{L}^{uur}$ we used a loss weight of $\lambda^{uur} = 0.25$. The concept datasets that we use for interpretability evaluation are Broden (Bau et al. (2017)) for Resnet18 and Broden Action (Ramakrishnan et al. (2019)) for Resnet50. The experimental results are provided in Table 12. The proposed $\mathcal{L}^{uur}$ increased the interpretability of the directions up to $+78.78\%$ in $\mathcal{S}^2$.

Table 12: Comparison with prior work on unsupervised basis learning for two networks: Resnet18 trained on Places365 (left) and Resnet50 trained on Moments-In-Time (MiT) (right). Works considered: Unsupervised Interpretable Basis Extraction (UIBE Doumanoglou et al. (2023)), Concept-Basis-Extraction (CBE Doumanoglou et al. (2024)) and Concept-Basis-Extraction with CNN Classifier Loss replaced with the proposed $\mathcal{L}^{uur}$ (CBE /w $\mathcal{L}^{uur}$). Significantly more interpretable directions are obtained for Resnet50.

|  | Resnet18 / Places365 | | | Resnet50 / MiT | | |
|---|---|---|---|---|---|---|
|  | UIBE | CBE | CBE /w $\mathcal{L}^{uur}$ | UIBE | CBE | CBE /w $\mathcal{L}^{uur}$ |
| $\mathcal{S}^1$ | 60.93 (+0.0%) | **69.43 (+13.95%)** | 67.3 (+10.45%) | 124.73 (+0.0%) | 131.73 (+5.61%) | **158.76 (+27.28%)** |
| $\mathcal{S}^2$ | 28.39 (+0.0%) | 31.53 (+11.06%) | **32.16 (+13.28%)** | 18.47 (+0.0%) | 26.94 (+45.86%) | **33.02 (+78.78%)** |

## A.12 Details for the Experiment on Deep Image Classifier

Table 13: Loss weights used for the experiment on the Deep Image Classifier.

|  | $\lambda^s$ | $\lambda^{sb}$ | $\lambda^{ma}$ | $\lambda^{mm}$ | $\lambda^{ic}$ | $\lambda^{uur}$ | $\lambda^{cur}$ | $\lambda^{eac}$ | $\lambda^{fso}$ | $\mathcal{L}^{id}$ | $\mathcal{L}^{de}$ |
|---|---|---|---|---|---|---|---|---|---|---|---|
| step (a) | 2.6 | - | 2.8 | 0.6 | 5.0 | 0.25 | - | - | - | - | - |
| step (b) | 0.85 | 2.6 | 2.8 | 0.6 | 15.0 | 0.25 | - | 15.0 | - | - | 15.0 |
| step (d) | 0.85 | 2.6 | 2.8 | 0.6 | 15.0 | - | 0.25 | 15.0 | 1.0 | 5.0 | 15.0 |

For step (a) direction learning lasts 800 epochs using an initial learning rate of 0.001 for a reference batch size of 4096 (which, in all steps, we scale based on the available GPU memory). We reduce the learning rate on plateau, by a factor of 0.5 with patience and cooldown set to 10 epochs. Step (b) lasts for 2000 epochs with initial learning rate of 0.0001 for the same reference batch size. We also reduce the learning rate on plateau by a factor of 0.5 but with patience and cooldown set to 50 epochs. For both steps (a) and (b) we use $\tau = 0.9$ for $\mathcal{L}^{ic}$. For $\mathcal{L}^{eac}$ we use $\rho = 12\tau/I$, chosen to roughly match the maximum number of pixels in any of the Broden classes. Step (d) lasts for another 2000 epochs with initial learning rate of 0.0005 with the $\tau$ hyper-parameter of $\mathcal{L}^{ic}$ set to 0.2 and $\rho = 70\tau/I$. The rest of the parameters remain intact with respect to step (b). For $\mathcal{L}^{id}$ we use $\delta = 5.0$ and for $\mathcal{R}_{SW}$ we use $\nu = 4.0$. Due to memory constraints, when learning with $\mathcal{L}^{id}$ (Table 7) we manipulate 150 random directions per iteration. Also when using $\mathcal{L}^{uur}$ and $\mathcal{L}^{cur}$, we observed better results when manipulating features with a stochastic magnitude in the direction

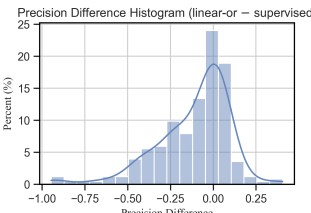 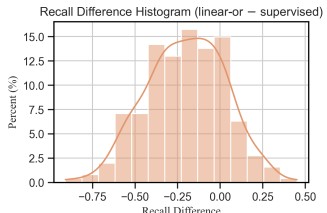 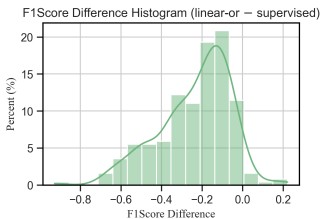

Figure 12: Interpretability Comparison. Histogram of differences in binary metrics: Precision, Recall, F1Score between the *Linear-OR* set of classifiers learned with the proposed method ($\mathcal{L}^{cur} + \mathcal{L}^{fso}$, $I = 500$) and classifiers learned in a supervised way (IBD Zhou et al. (2018)).

$dx_p$, i.e. shifting representations as $x'_p = x_p - \kappa dx_p$ with $\kappa$ a random number in $[0.5, 0.9]$. Table 13, summarizes the loss weights that we used for steps (a), (b) and (d). In practice, we separate filter directions from their magnitude $1/M_i$ and learn them independently as suggested in Doumanoglou et al. (2024). For enforcing $||w_i||_2 = 1$ (i.e. unit norm filter vectors) we use parametrization on the unit hyper-sphere. In steps (b) and (d) we also used the Dataset Explanation Loss, described in section A.6 with hyper-parameter $\mu = 0.9$. Due to resource limitations, we were not able to conduct experiments that prove significant positive contributions for cases of $\mu < 1$, and for this reason we did not include it in the main text. Table 6 summarizes the interpretability and influence metrics for $\mu = 0.8, 0.9, 1.0$. A future study could provide better insights for whether learning directions with $\mu < 1$ provides significant benefits in terms of interpretability or influence.

Regarding the interpretability evaluation protocol (results of 5), for each concept, we construct a dataset comprised of negative samples that are up-to 20 times more than the number of positive samples, as a means to mitigate the great imbalance. For RCAV's perturbation hyper-parameter, we use $\alpha = 5$. For direction significance testing, we use RCAV's label permutation test. To construct random noise signal vectors, we (a) construct a dataset of feature-label pairs based on the decision rule of each one of the concept detectors. To deal with great class imbalance, we construct a pool of negative samples that is at most 20 times more than the positive ones (b) we construct $N$ noisy versions of that dataset by label permutation (c) we learn a noise-classifier to distinguish features based on the permuted labels, and (d) we concurrently, estimate a noise-signal vector using (1). To train each one of the noise signal vectors and before permuting the labels, we construct a balanced dataset of at most 5000 samples, picked randomly from the pool. We train the noise classifiers using Adam for 100 epochs and a learning rate $0.01$. By using noise signal vectors as RCAV's noisy directions, and with the number of of those vectors per classifier set to $N = 100$, we subsequently calculate RCAV's p-values. We apply Bonferroni correction to all p-values, by diving the significance threshold $0.05$ with the number of concept detectors $I$ and the number of model classes ($K = 365$).

**Ablation on the number of directions ($I$).** Table 7 (right) shows results for a varying number of concept detectors $I$. Notably, both interpretability and influence improved as the number of directions increased. As previously discussed, the addition of signal vectors enhances both interpretability and influence (evident when comparing metrics for $I = 500$ with the scores in Table 1).

### A.12.1 DETAILED INTERPRETABILITY COMPARISON AGAINST THE SUPERVISED APPROACH FOR THE EXPERIMENT ON DEEP IMAGE CLASSIFIER

Figure 12 plots a histogram of classification metric differences between the *Linear-OR* set of classifiers and the classifiers learned in a supervised way. The differences are based on the labels, effectively taking the difference of metrics that regard two classifiers (the first from the *Linear-OR* set and the second from Zhou et al. (2018)) with the same concept name.

Figures 13, 14, 15 depict concrete binary classification metrics for some of the concept detectors in the *Linear-OR* set of classifiers, comparing them with concept classifiers learned with supervision.

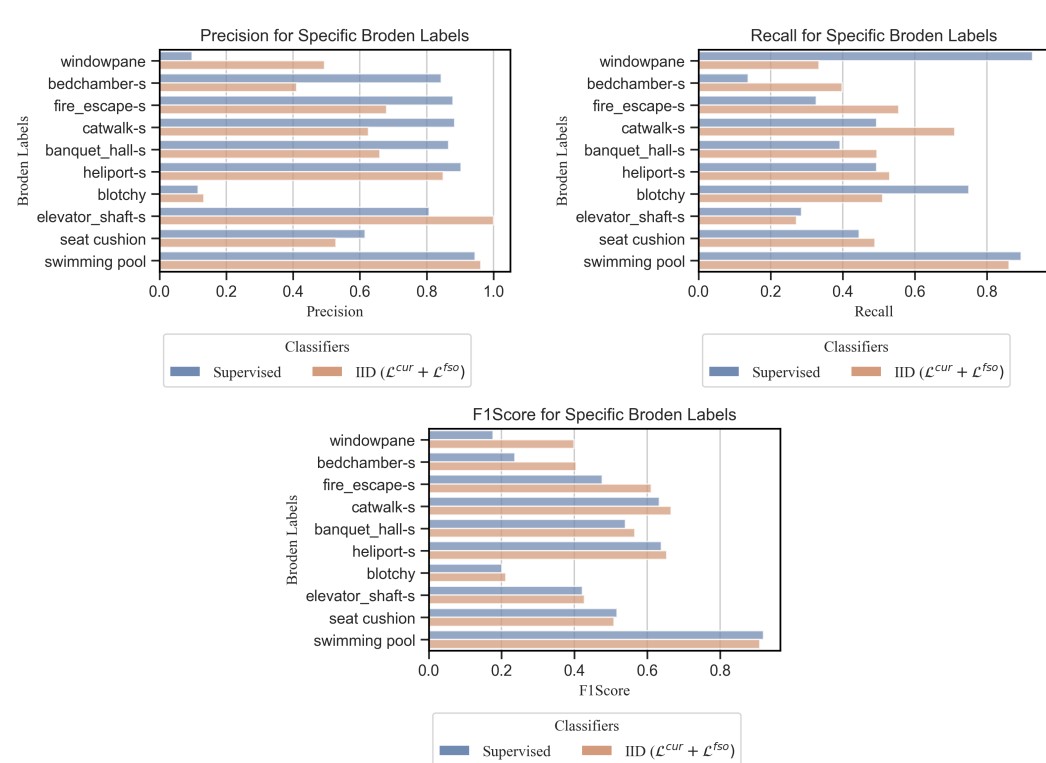

Figure 13: Interpretability Comparison. Exact Precision/Recall/F1Scores for specific concepts in Broden: comparison between the *linear-or* set of classifiers learned with the proposed method (IID, $I = 500$) and classifiers learned in a supervised way (IBD Zhou et al. (2018)).

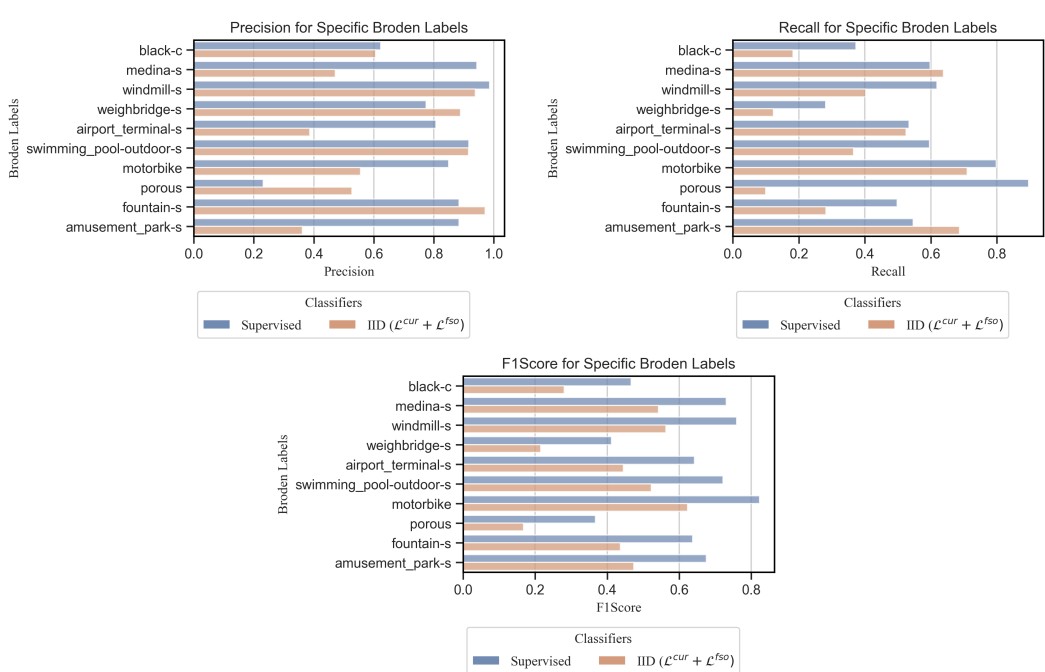

Figure 14: Interpretability Comparison. Exact Precision/Recall/F1Scores for specific concepts in Broden: comparison between the *linear-or* set of classifiers learned with the proposed method (IID, $I = 500$) and classifiers learned in a supervised way (IBD Zhou et al. (2018)).

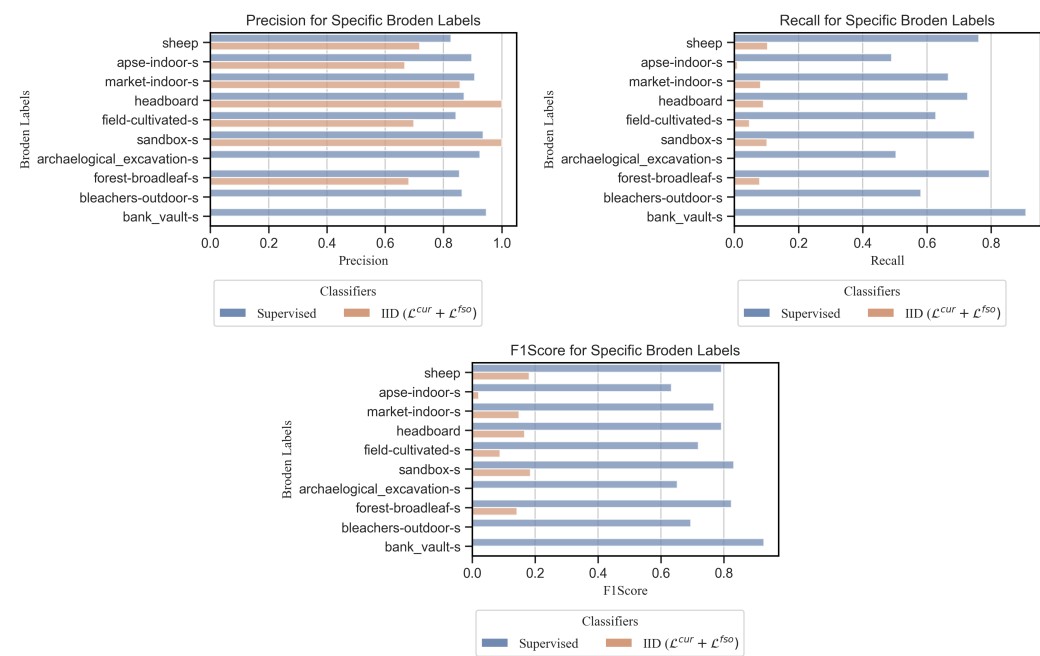

Figure 15: Interpretability Comparison. Exact Precision/Recall/F1Scores for specific concepts in Broden: comparison between the *linear-or* set of classifiers learned with the proposed method (IID, $I = 500$) and classifiers learned in a supervised way (IBD Zhou et al. (2018)).

### A.12.2 DETAILED INFLUENCE METRICS AND DIAGRAMS FOR THE EXPERIMENT ON THE DEEP IMAGE CLASSIFIER

Table 14 provides more summarizing influence statistics regarding the signal vectors learned with the proposed method. In that table, $SC_{i,j}$ denotes RCAV's sensitivity score in the direction of the $i$-th signal vector, for the network's class $j$. Figures 16,17,18,19 depict concrete examples of how each concept's signal direction impacts the Resnet18's class predictions. Concepts appearing more than once. correspond to different directions that have been attributed the same label by Network Dissection. Seemingly irrelevant concepts with positive influence may have three possible explanations: a) the network has some sensitivity to those concepts (as it's top1 accuracy is 56.51%) b) their impact might be low, since RCAV only considers the sign of the class prediction difference before and after the manipulation, regardless of its magnitude, (thus those concepts may influence the prediction class positively, but by only a small amount) and c) the respective concept detectors do not reliably predict the concept (that is they exhibit a low IoU score) and thus the concept name may be misleading.

Table 14: This table summarizes statistics of the RCAV's sensitivity score matrix $SC$ for the set of directions learned with $\mathcal{L}^{uur}$ or $\mathcal{L}^{cur} + \mathcal{L}^{fso}$, $I = 500$, RCAV $\alpha = 5.0$. All entries in the sensitivity score matrix $SC$ are masked for significance before computing the statistics. Sensitivity scores were obtained using *signal* vectors calculated using (1).

| Metric | Formula | $\mathcal{L}^{uur}$ | $\mathcal{L}^{cur} + \mathcal{L}^{fso}$ |
|---|---|---|---|
| Significant Direction Count | | 359 | **376.0** |
| Significant Class-Direction Pairs | | 2118 | **3271.0** |
| Directions /w Positive Influence | $\sum_i \max_j \mathbb{1}_{x>0}(SC_{i,j})$ | 174 | **185.0** |
| Directions /w Negative Influence | $\sum_i \max_j \mathbb{1}_{x<0}(SC_{i,j})$ | 350 | **366.0** |
| Positively Impactful Directions Per Class | $\frac{1}{K}\sum_{i,j} \mathbb{1}_{x>0}(SC_{i,j})$ | **1.41** | 1.24 |
| Negatively Impactful Directions Per Class | $\frac{1}{K}\sum_{i,j} \mathbb{1}_{x<0}(SC_{i,j})$ | 4.39 | **7.71** |
| Minimum # of Positively Influencing Classes Across Directions | $\min_i \sum_j \mathbb{1}_{x>0}(SC_{i,j})$ | 0 | 0 |
| Maximum # of Positively Influencing Classes Across Directions | $\max_i \sum_j \mathbb{1}_{x>0}(SC_{i,j})$ | **16** | 13 |
| Minimum # of Negatively Influencing Classes Across Directions | $\min_i \sum_j \mathbb{1}_{x<0}(SC_{i,j})$ | 0 | 0 |
| Maximum # of Negatively Influencing Classes Across Directions | $\max_i \sum_j \mathbb{1}_{x<0}(SC_{i,j})$ | 27 | **46** |
| # of Classes /w at Least One Positively Impactful Direction | $\sum_j \mathbb{1}_{x>1}\left(\sum_i \mathbb{1}_{x>0}(SC_{i,j})\right)$ | 147 | **161** |
| # of Classes /w at Least One Negatively Impactful Direction | $\sum_j \mathbb{1}_{x>1}\left(\sum_i \mathbb{1}_{x>0}(SC_{i,j})\right)$ | 213 | **345** |

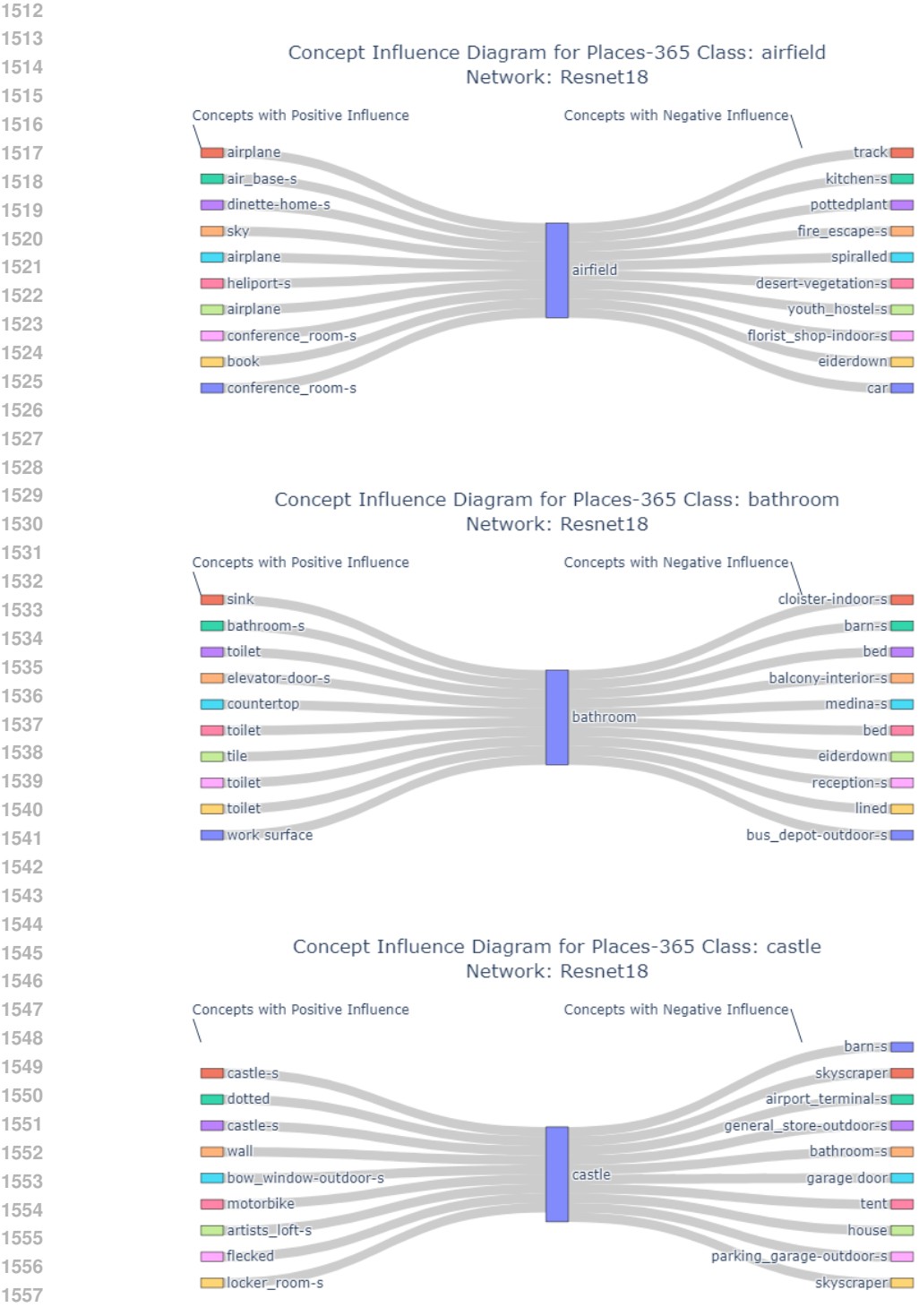

Figure 16: Concept Influence Diagram for Resnet18 trained on Places365. The depicted concepts have sensitivity scores above 0.99 in absolute terms. (We use RCAV to quantify the sensitivity, and re-scale the score to $[-1, 1]$) Positive influencing and negative influencing concepts are provided. The number of concepts have been limited to 10. When concepts appear more than once, they correspond to different signal directions (as labeling the classifiers with NetDissect may assign the same concept name to more than one directions.)

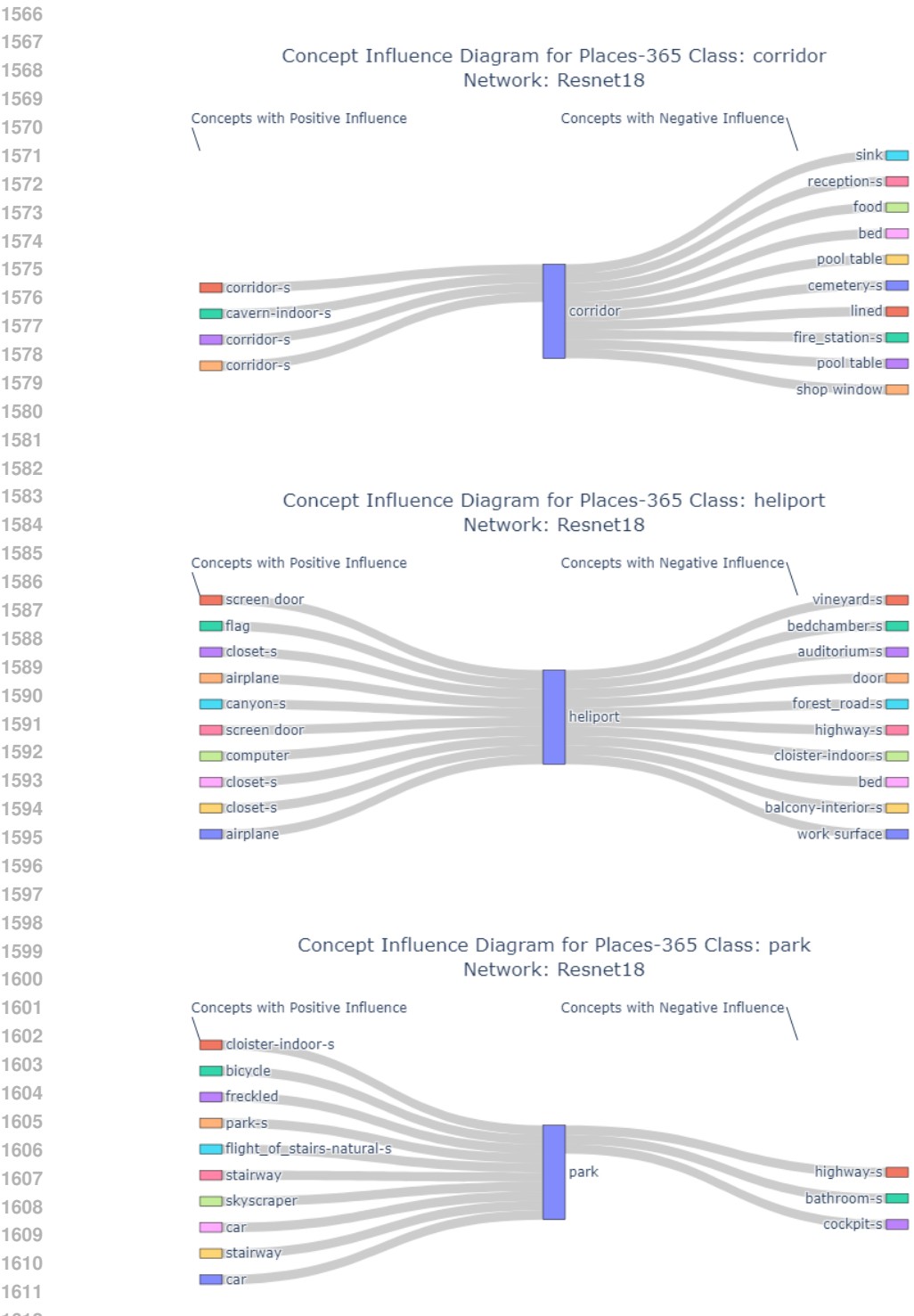

Figure 17: Concept Influence Diagram for Resnet18 trained on Places365. The depicted concepts have sensitivity scores above 0.99 in absolute terms. (We use RCAV to quantify the sensitivity, and re-scale the score to [−1, 1]) Positive influencing and negative influencing concepts are provided. The number of concepts have been limited to 10. When concepts appear more than once, they correspond to different signal directions (as labeling the classifiers with NetDissect may assign the same concept name to more than one directions.)

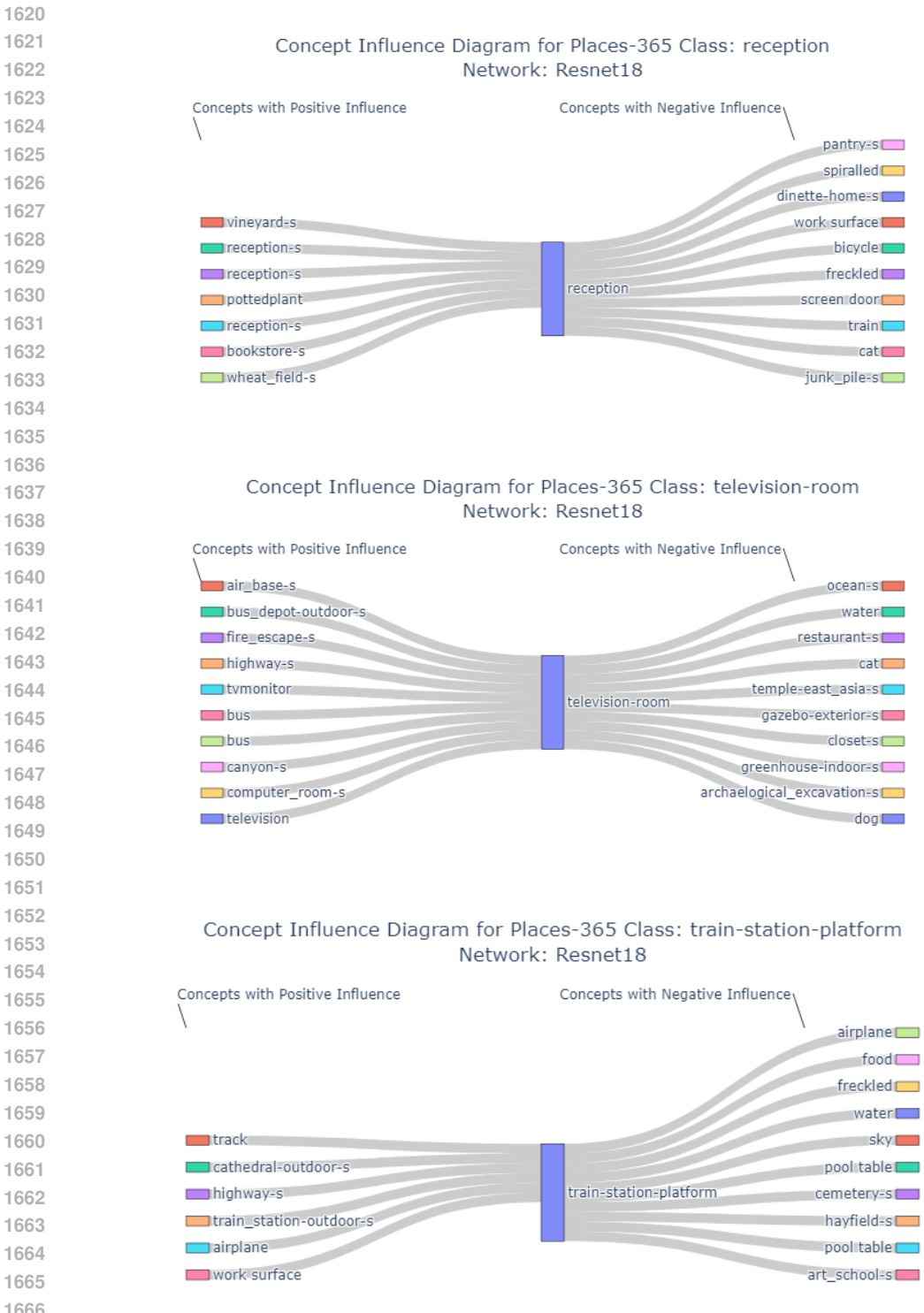

Figure 18: Concept Influence Diagram for Resnet18 trained on Places365. The depicted concepts have sensitivity scores above 0.99 in absolute terms. (We use RCAV to quantify the sensitivity, and re-scale the score to $[-1, 1]$) Positive influencing and negative influencing concepts are provided. The number of concepts have been limited to 10. When concepts appear more than once, they correspond to different signal directions (as labeling the classifiers with NetDissect may assign the same concept name to more than one directions.)

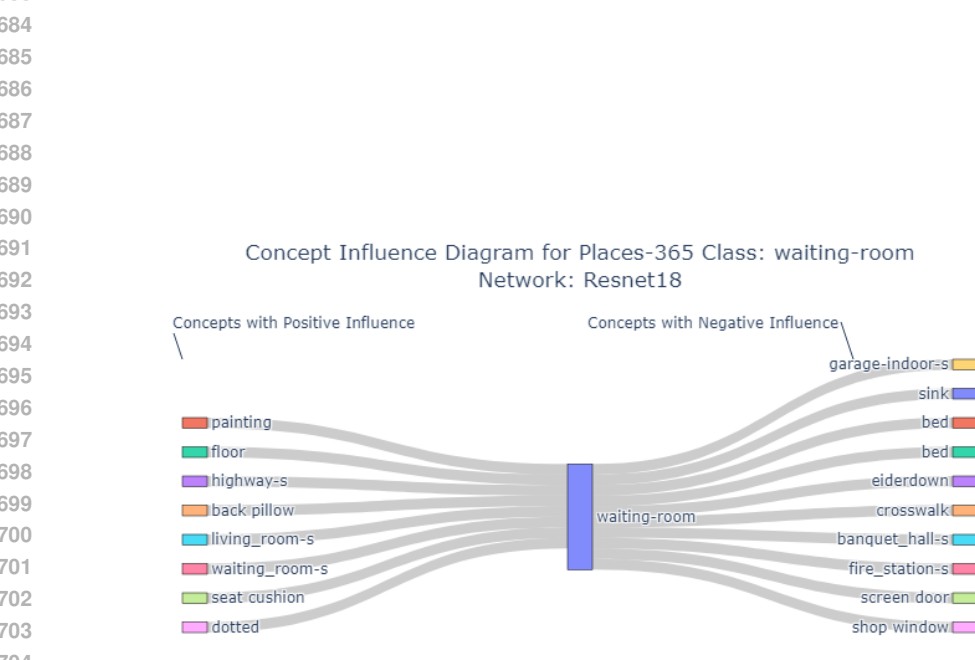

Figure 19: Concept Influence Diagram for Resnet18 trained on Places365. The depicted concepts have sensitivity scores above 0.99 in absolute terms. (We use RCAV to quantify the sensitivity, and re-scale the score to $[-1, 1]$) Positive influencing and negative influencing concepts are provided. The number of concepts have been limited to 10. When concepts appear more than once, they correspond to different signal directions (as labeling the classifiers with NetDissect may assign the same concept name to more than one directions.)

### A.13 Qualitative Segmentation Results for the Experiment on Deep Image Classifier

Figures 20 and 21 depict qualitative segmentation results for the classifiers learned with the proposed method in the experiment with the deep image classifier. Visualizations are obtained using Bau et al. (2017) which reported that our concept detectors can identify 257 different concepts (at the IoU threshold level of 0.04) in the following categories: 65 objects, 158 scenes, 12 parts, 3 materials, 18 textures and 1 color. The total number of interpretable concept detectors is 429 (out of the 500 in the set). In the figures, the IoU scores refer to the whole validation split of the concept dataset and not individual image segmentations. The labels assigned to the concept detectors are based on the annotations available in the concept dataset and in some cases may not be very accurate. For instance consider classifiers with index 343 and 430. They have been assigned the label *hair* while a more suitable label might be *face*, but such a concept class is not available in the annotations of the dataset. Other notable cases include the classifiers with indices 76 and 444. The first is specialized in detecting *cars* that dominate the image space, while the second appears better suited for identifying smaller instances of the same class. The IoU for the latter is only 0.05 because it is calculated across the entire set of cars, regardless of their size. Lastly, the classifier with index 45 seems adept at detecting the upper body of a person, whereas the classifier with index 256 is more effective at identifying the lower body.

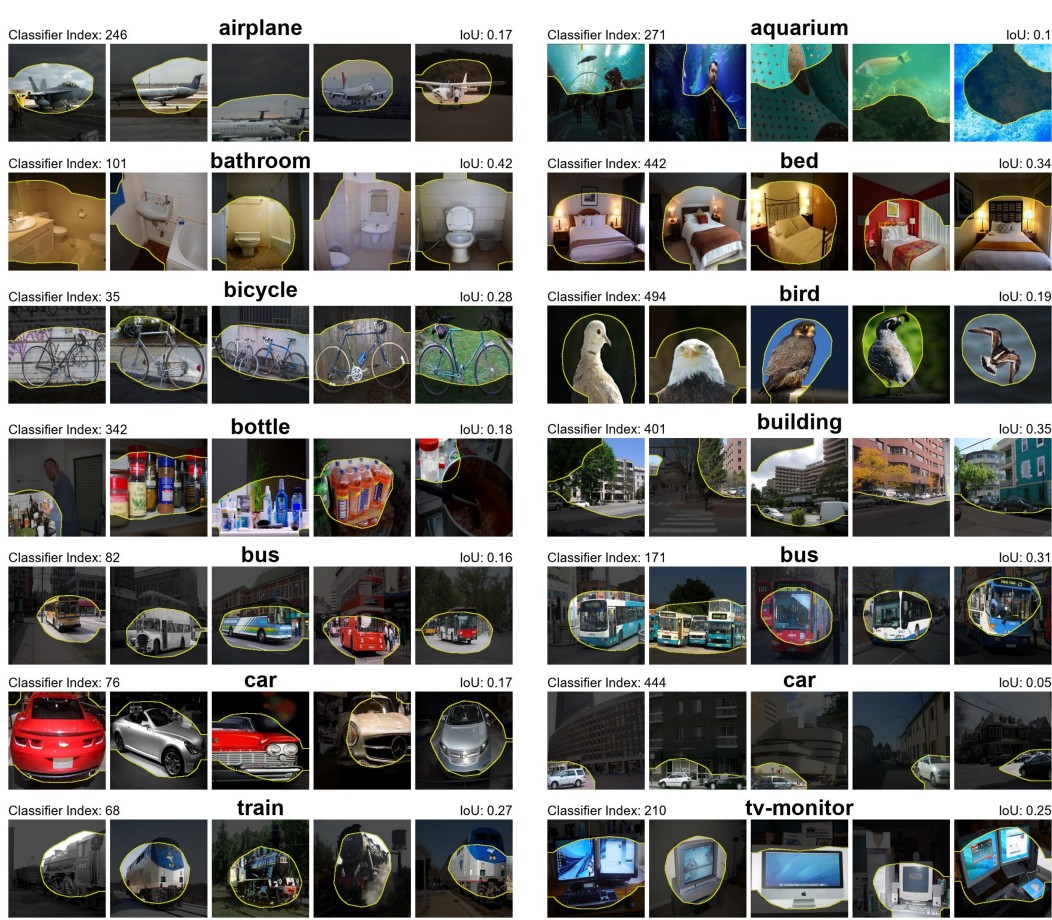

Figure 20: Qualitative segmentations using the classifiers learned with our method. Here $I = 500$.

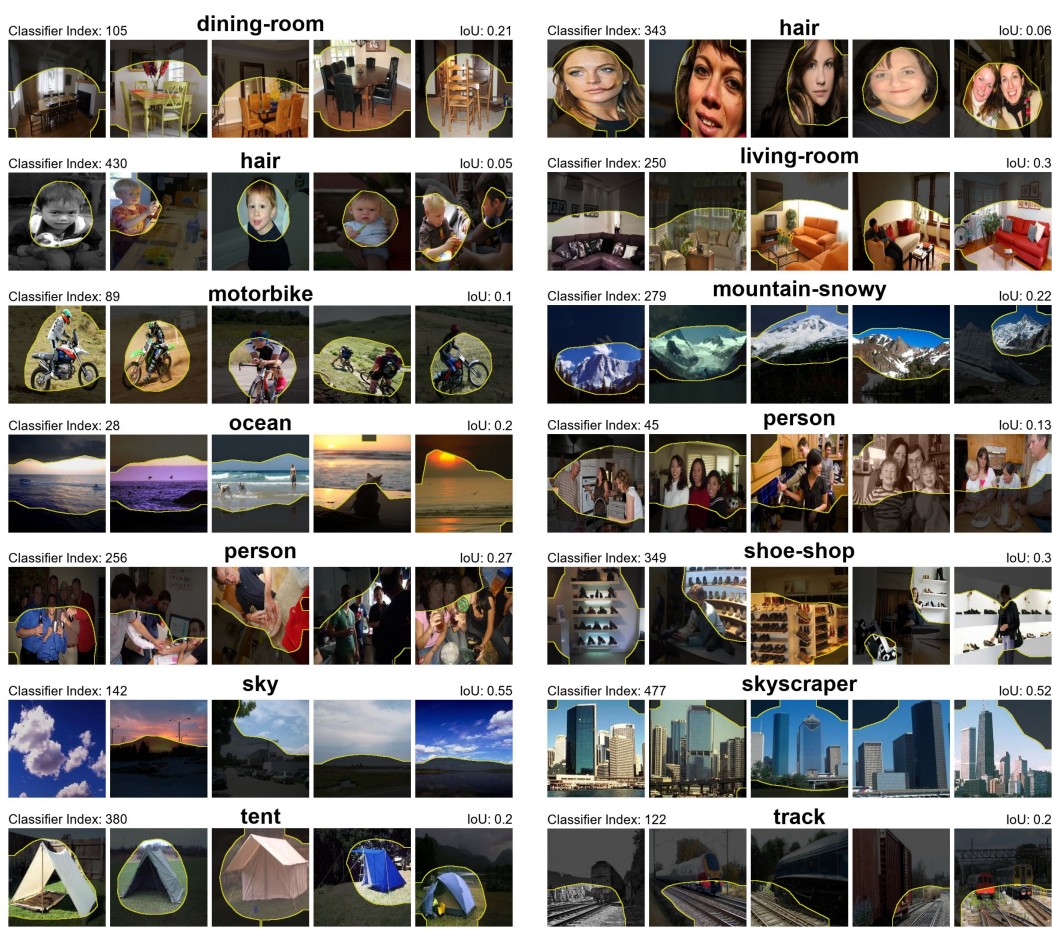

Figure 21: Qualitative segmentations using the classifiers learned with our method. Here $I = 500$.

Table 15: Network accuracy and confusion matrix for the network trained on the Chess Pieces dataset. Rows correspond to ground-truth labels and colums to network predictions. Three classes are considered: **b**ishop/**k**night/**r**ook. The rows of the confusion matrix are normalized against ground-truth element count. Three datasets are also considered: **Clean** (without watermarks), **Poisoned** (with watermarks) and **Clean & Poisoned** which is the union of the previous two.

| *Dataset:* | **Clean** | | | **Poisoned** | | | **Clean & Poisoned** | | |
|---|---|---|---|---|---|---|---|---|---|
| *Accuracy:* | 0.93 | | | 0.34 | | | 0.64 | | |
| | **b** | **k** | **r** | **b** | **k** | **r** | **b** | **k** | **r** |
| **b** | 0.95 | 0.05 | 0.0 | 0.0 | 0.0 | 1.0 | 0.48 | 0.02 | 0.5 |
| **k** | 0 | 0.95 | 0.05 | 0.0 | 0.0 | 1.0 | 0.0 | 0.48 | 0.52 |
| **r** | 0.04 | 0.04 | 0.92 | 0.0 | 0.0 | 1.0 | 0.02 | 0.02 | 0.96 |

## A.14 TOY EXPERIMENT ON MODEL CORRECTION WITH THE LEARNED DIRECTIONS

In this experiment we demonstrate how the proposed approach may be utilized to correct a model that relies on controlled confounding factors to make its predictions. For the purposes of this toy experiment we use a small convolutional neural network with 5 `Conv2d` layers each one followed by a `ReLU` activation. The top of the network is comprised of a Global Average Pooling layer (GAP) and a linear head. After each convolutional layer, except the last, there is a `Dropout` layer with $p = 0.3$. All `Conv2d` layers have kernel size 3x3 and stride 2 except the last one which has stride 1. Furthermore, the latent space dimensionality is set to 16 for all convolutional units. We consider the task of predicting the chess piece name from an image depicting the piece. We use the *Chess Pieces* dataset from Kaggle [1] which contains a collection of images depicting chess pieces from various online platforms (i.e., piece images appearing in online play). The spatial resolution of those images is 85x85. For simplicity, we consider 3 chess pieces to be classified by the network, namely: *bishop, knight, rook*, thus the network predicts $K = 3$ output classes. The total number of images in the dataset are 210, 67 for *bishop*, 71 for *knight* and 72 for *rook*. We make a stratified train-test split with the training set ratio set to 0.7. To encourage the model to learn a bias to make its predictions, we poison half of the *rook* images of the training set with the watermark text "rook" on the top left of each rook image. With the introduction of this bias on half of the images, we expect that the network learns that the watermark concept has positive influence on the *rook* class, while not being the only feature of positive evidence for the same class, since we include rook images in the training set without the watermark.

### A.14.1 NETWORK TRAINING AND EVALUATION

We train the network with cross-entropy loss and the Adam optimizer with learning rate 0.005 for 1000 epochs. In the (poisoned) training set the model achieves 100% accuracy. For evaluation we construct three datasets based on the test split that we created earlier. First, we consider a clean test set, a dataset comprised of test images without any watermarks (**Clean**). Second, we consider the previous clean set but with all the images being poisoned with the watermark (**Poisoned**) and c), we consider the union of the previous two datasets (**Clean & Poisoned**). Table 15 summarizes the performance of the network in each one of the three datasets. As evidenced by the *Poisoned* section of the detailed confusion matrix, the watermark is a strong feature that whenever is present in the image it directs the prediction towards the *rook* class.

### A.14.2 DIRECTION LEARNING FOR WATERMARK DIRECTION IDENTIFICATION

We now consider the application of the proposed method in identifying the watermark direction, from the bottom up, without relying on annotations. The proposed approach is unsupervised and identifies directions influential to the model. Since in the previous section we identified that the watermark actually influences the predictions of the network in a consistent manner, we seek to answer the following two research questions: a) can the proposed IID method identify the watermark as a concept? That is, is any of the learned classifiers responsible to detect the watermark? b) Supposing the answer to (a) is positive, given the watermark's concept detector and the respective

---

[1] Chess Pieces Dataset (85x85): `https://www.kaggle.com/datasets/s4lman/chess-pieces-dataset-85x85`

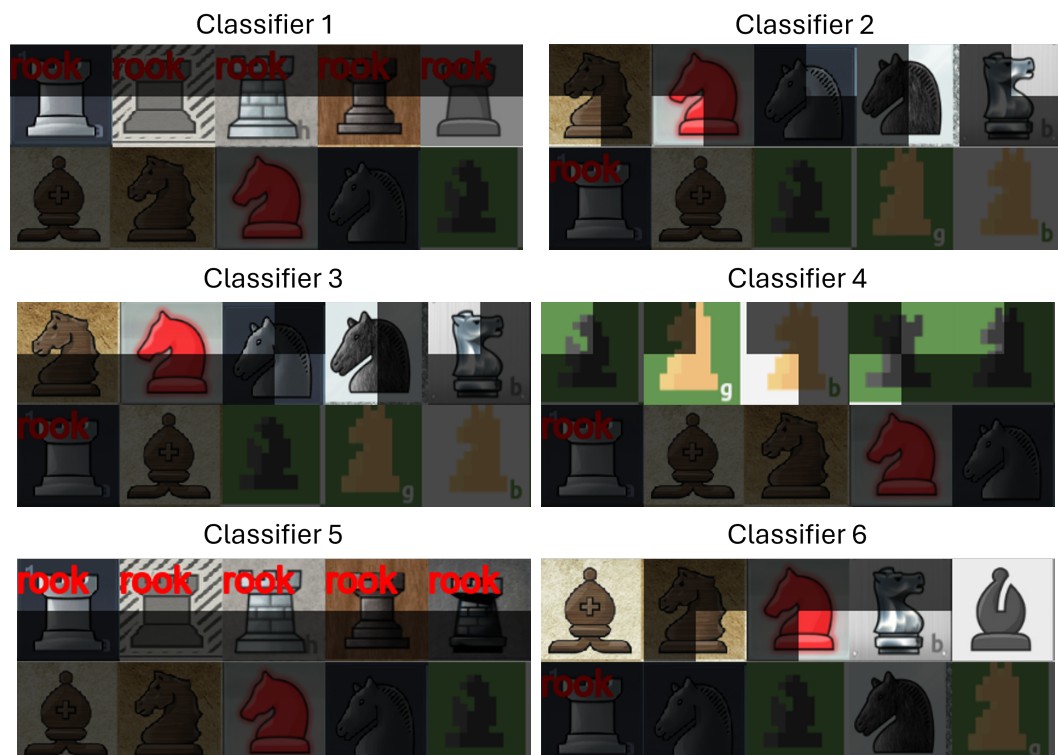

Figure 22: Example image segmentations based on the concept detectors learned for the model correction experiment. Top rows illustrate pictures with the concept and bottom rows illustrate pictures without. Classifier 5 clearly detects the watermark.

learned signal vector, can we fix the network in order to not rely on the watermark for its predictions ?

**Direction Learning** We consider the last convolutional layer as our layer of study. This layer has spatial dimensionality 2x2. To learn the latent directions, we apply our method by following the learning process described in section A.9 and we use the network's training set as our concept dataset. Furthermore, for stable learning, we have found that the directions are more robustly learned with the Augmented Lagrangian loss scheme, which implies that the optimization problem is formulated as a constrained optimization problem. We optimize $\lambda^s \mathcal{L}^s + \lambda^{sb} \mathcal{L}^{sb} + \lambda^{cur} \mathcal{L}^{cur}$ with the constraints $\mathcal{L}^{ma} < 0.8, \mathcal{L}^{ic} < 0.01, \mathcal{L}^{eacl} < 0.01, \mathcal{L}^{mm} < 8.0, \mathcal{L}^{fso} < 0.1$, and weights $\lambda$ similar to Table 13. The most important hyper-parameter to tune is the dimensionality of the concept space $I$.

**Watermark Direction Identification** We found that, when learning with $I = 6$, the proposed approach clearly identifies the watermark direction. By using the learned classifiers as concept detectors, we are able to group each one of the spatial features of the 2x2 representation, into clusters of the same concept. When applied on an image representation, each learned classifier produces a form of a binary label-map, with each element of the label-map indicating whether the part of the image behind the spatial representation belongs to the concept. In Fig. 22 we provide example image segmentations based on those label-maps. From the qualitative visualizations we see that classifier 5 identifies the watermark. Although annotations were not required to learn the direction, since this is a controlled experiment and we know in which images we injected the watermark, we are able to quantify how well this classifier can detect the concept by evaluating its IoU performance on the concept dataset. We found that this classifier detects the watermark concept with IoU 0.93.

### A.14.3 INFLUENCE TESTING WITH RCAV

We use RCAV to measure the sensitivity of the model with respect to the watermark signal vector. The sensitivity scores reported by RCAV are $-1$ for the classes *bishop* and *knight* while it is $1$ for the class *rook*. This implies that when the image has the watermark it becomes more *rook* and less *bishop* or *knight*. This quantitative score aligns with our intuition regarding the watermark.

### A.14.4 MODEL CORRECTION BY USING THE WATERMARK'S INTERPRETABLE AND INFLUENTIAL DIRECTIONS

Let $w$ and $b$ denote the learned parameters of the watermark concept detector and $s$ denote the respective learned signal vector. Without re-training or fine-tuning the network, we are going to suppress the watermark artifact component from the representation whenever it is detected by the concept detector. We propose the following feature manipulation strategy that we apply at the features of the last convolutional layer.

$$x'_p = \text{ReLU}(x_p - mk\hat{s}), m = \sigma(w^T x_p - b) \tag{21}$$

with $k$ a perturbation hype-parameter that we empirically set to $k = 450$. The ReLU ensures that the manipulation does not move the features out of the domain of the linear head.

### A.14.5 EVALUATION OF THE CORRECTED MODEL

We evaluate the corrected model according to the same protocol that we did in section A.14.1. The results are depicted in Table 16. Compared to the performance of the original network (Table 15), we see that the corrected model: a) has the same accuracy as the original model on the clean test set b) is significantly more accurate on images of the poisoned dataset with an absolute improvement of +34% and c) performs substantially better on the union of clean and poisoned datasets with an absolute improvement of +17%. We also compare our correction strategy to using a random manipulation direction with the same $k$ as before. (i.e. using a random vector in the place of the learned signal vector in (21)). In a series of 10 trial evaluations on the Poisoned Test set, we verified that **no improvement** was achieved: the classification accuracy was the same as the original model and the confusion matrix was still the same as in Table 15-Middle. Finally, we also compare against manipulating towards the concept detector's filter direction and we found that the classification accuracy for the poisoned test set was $0.53$ (which is inferior to $0.69$ when using the signal vector).

Table 16: Network accuracy and confusion matrix for the **corrected** network trained on the Chess Pieces dataset. Rows correspond to ground-truth labels and colums to network predictions. Three classes are considered: **b**ishop/**k**night/**r**ook. The rows of the confusion matrix are normalized against ground-truth element count. Three datasets are also considered: **Clean** (without watermarks), **Poisoned** (with watermarks) and **Clean & Poisoned** which is the union of the previous two.

| *Dataset:* | **Clean** | | | **Poisoned** | | | **Clean & Poisoned** | | |
|---|---|---|---|---|---|---|---|---|---|
| *Accuracy:* | 0.93 | | | 0.69 | | | 0.81 | | |
| | **b** | **k** | **r** | **b** | **k** | **r** | **b** | **k** | **r** |
| **b** | 0.95 | 0.05 | 0.0 | 0.4 | 0.3 | 0.3 | 0.67 | 0.17 | 0.15 |
| **k** | 0.0 | 0.95 | 0.05 | 0.0 | 0.85 | 0.14 | 0.0 | 0.90 | 0.10 |
| **r** | 0.04 | 0.04 | 0.91 | 0.0 | 0.18 | 0.81 | 0.02 | 0.11 | 0.87 |

