# OpenReview forum: "Learning Interpretable and Influential Directions with Signal Vectors and Uncertainty Region Alignment"
_ICLR.cc/2025/Conference — Submitted to ICLR 2025_

### Official Review · Reviewer_7yop · 2024-10-20

**Soundness:** 4
**Presentation:** 2
**Contribution:** 3
**Rating:** 6
**Confidence:** 2

**Summary:**

This paper presents an unsupervised method for discovering interpretable and influential latent space directions in deep-learning models using signal vectors and uncertainty region alignment. The approach is validated on synthetic and proof-of-concept real-world data, showing its effectiveness in recovering key signal directions and improving model debugging.

**Strengths:**

- The presented method is very well theoretically motivated and is a valuable extension to existing (supervised) discovery methods like CAV.
- The experiments have been carried out accurately and with care. However, I would classify both experiments more as proof-of-concepts than real-world experiments as the places365 dataset is very limited in complexity. However, this is totally valid in my opinion, as the focus of the paper is the theoretical contribution.
- The ablation studies on the loss components are extensive and interesting.

**Weaknesses:**

**W1:** The paper is sometimes hard to follow and understand. While in the methods section, the equations help to understand what’s described in the text, the different experiments are sometimes hard to follow. Especially for a reader unfamiliar with latent space-based concept methods for interpretability the different methods and the interpretation of their results can be confusing. For example, the experiment setup in 4.1 and how it connects to section 2 could be better explained. The experiment setup in 4.2 refers to several methods from related work (e.g. Network Dissection, Broden dataset, RCAV (which was never introduced)), which readers are possibly not familiar with.

To improve clarity, I suggest adding an overview of key concepts at the start of the experiments section, explicitly explaining how Section 4.1 connects to Section 2, and providing brief explanations for unfamiliar methods (not limited to) like Network Dissection, the Broden dataset, and RCAV when they are first introduced.

**W2:** The authors comment in Line 297 that experiments with the raw inclusion of the three loss terms do not converge, but do not further elaborate on how stable the final results are with their adaptations, i.e. if the learned directions are robust.

Please provide quantitative measures of the robustness in the learned directions, e.g. through the standard deviation or confidence intervals over several differently seeded runs for the tables in section 4 (if applicable).

**W3:** Figures 1 and 2 are very helpful in communicating the concepts but are never referred to in the main manuscript. The paper would benefit from a general overview figure in the same style (in which these two figures would be a part), similar to a graphical abstract to communicate the different steps and concepts visually.

Please provide an overview figure on how the presented method is composed (and maybe why certain selections were made) in section 3. If it would make sense for you, also integrate Figure 2 into this overview figure.

**W4:** The paper lacks a published code repository, making it difficult for others to easily apply the method to their own models.

Please provide a link to an anonymous repository with a (simple) implementation of the method. You can use "Anonymus Github" or GitLab with an anonymous email.

I am open to raising my score if the identified weaknesses are either adequately addressed through revisions to the manuscript or convincingly argued to be non-issues.

**Questions:**

See weaknesses W1 to W4.

---

> ### Author Response · Authors · 2024-11-18
> **Answer to weaknesses**
>
> >W1: The paper is sometimes hard to follow and understand. While in the methods section, the equations help to understand what’s described in the text, the different experiments are sometimes hard to follow. Especially for a reader unfamiliar with latent space-based concept methods for interpretability the different methods and the interpretation of their results can be confusing. For example, the experiment setup in 4.1 and how it connects to section 2 could be better explained. The experiment setup in 4.2 refers to several methods from related work (e.g. Network Dissection, Broden dataset, RCAV (which was never introduced)), which readers are possibly not familiar with.
>
> >To improve clarity, I suggest adding an overview of key concepts at the start of the experiments section, explicitly explaining how Section 4.1 connects to Section 2, and providing brief explanations for unfamiliar methods (not limited to) like Network Dissection, the Broden dataset, and RCAV when they are first introduced.
>
> - **Reply**:  Indeed, we understand about the confusion. We would very much appreciate the reviewer's opinion in what parts of the text would be preferably omitted in a favor of more explanations regarding background and related work. Please see also our comments to Reviewer #3: 7aZ9 and the general comment to all the reviewers.
>
> >W2: The authors comment in Line 297 that experiments with the raw inclusion of the three loss terms do not converge, but do not further elaborate on how stable the final results are with their adaptations, i.e. if the learned directions are robust.
>
> >Please provide quantitative measures of the robustness in the learned directions, e.g. through the standard deviation or confidence intervals over several differently seeded runs for the tables in section 4 (if applicable).
>
> - **Reply**:  If we perform direction learning directly using the loss terms introduced in this paper indeed we faced some difficulties of convergence (i.e training was unstable, deviating towards minimizing a single objective instead of the multiple objectives that we balanced with different weights). Thus, we followed the standard pre-training setting. We fist learned directions like in Doumanoglou et al 2024 by with our $\mathcal{L}^{uur}$ instead of their CCL loss (see also appendix A.8) and subsequently we used the converged optimum of this learning process as an initialization point to learn directions with our method. Indeed this "pre-training" strategy was found to be reliable in all of our experiments in synthetic or a real-world setting. Regarding the request of the reviewer to report statistical deviations of the metrics across several runs, unfortunately we think that due to lack of available resources in this short review period, we will not be able to address it.
>
> >W3: Figures 1 and 2 are very helpful in communicating the concepts but are never referred to in the main manuscript. The paper would benefit from a general overview figure in the same style (in which these two figures would be a part), similar to a graphical abstract to communicate the different steps and concepts visually.
>
> >Please provide an overview figure on how the presented method is composed (and maybe why certain selections were made) in section 3. If it would make sense for you, also integrate Figure 2 into this overview figure.
>
> - **Reply**: Indeed this was something we missed. We take the comment of the reviewer seriously. As we already mentioned in our general comment to all the reviewers, we offer to revise our text and devote more space in explaining the overall and background concepts in a more comprehensible way.
>
> > W4: The paper lacks a published code repository, making it difficult for others to easily apply the method to their own models.
>
> >Please provide a link to an anonymous repository with a (simple) implementation of the method. You can use "Anonymus Github" or GitLab with an anonymous email.
>
> - **Reply**: Reproducible research is indeed important. We have plans to open the code when the paper is accepted.

---

> > ### Comment · Reviewer_7yop · 2024-11-19
> >
> > Dear Authors,
> >
> > Thank you for the very detailed answer to my review and the answer to all reviewers! In general, I would like to comment that these very detailed answers often lose the authors' main message. Thus, I would advocate keeping it shorter and to the point, which I know is sometimes an art in itself.
> >
> > Regarding the replies:
> >
> > **W1**: I would see a lot of potential in compressing the Introduction section as well as the Method section (Which has a lot of subsections). Further, I would not say that you need *more* background but maybe switch to the background needed to understand the main messages of the paper. For example, I am not sure if the Background section needs such a detailed mathematical introduction of the supervised approach if the proposed method is unsupervised. In general, I would switch out too detailed (mathematical) explanations for more structure and arguing *why* something is done. If I do not know the motivation for specific methodological choices I don't know its novelty (and also lose automatically interest).
> >
> > **W2**: I acknowledge that the review time is quite short for extensive experiments. However, while the answer recognizes the problem it does not offer any solution to it.
> >
> > **W3**: I don't see how this is an answer to my question or my specific action item.
> >
> > **W4**: While I appreciate the authors' intention to make the code public following acceptance, the absence of an anonymously published repository for review limits my ability to assess the implementation. I strongly encourage the authors to include code implementations when introducing new methods, as this not only facilitates a thorough evaluation but also demonstrates the practical applicability of the method, which can significantly enhance its adoption by the community.

---

> > > ### Author Response · Authors · 2024-11-27
> > >
> > > We thank the reviewer for her reply. Below some comments regarding the revision:
> > >
> > > **W1**: In the revised version of the manuscript we made serious efforts in improving the comprehension of our manuscript. We include a detailed background section which contains all the important background points that the reader is required to be familiar with in order to understand the contributions of our work. In the revised version, we talk about Network Dissection in Section 2.4, we added a short description about Broden dataset in Section 4.2 and there is also a discussion on RCAV in Section 2.5 and A.8.
> > >
> > > **W3**: In the revised version of the manuscript we updated Fig.1 with the different steps of our approach from learning the directions to interpretability and influence evaluation, in a graphical-abstract sense that the reviewer suggested. We also mention this figure in the introduction of Section 3. Additionally, Fig. 2 is properly mentioned in the text in Section 3.4

---

> > > > ### Comment · Reviewer_7yop · 2024-12-03
> > > >
> > > > Dear Authors, while the rebuttal has addressed some weaknesses and moved the paper in the right direction, significant concerns remain unresolved, which do not justify an increase in rating.
> > > >
> > > > Thus, considering the overall quality of the paper, I will maintain my original rating.

---

### Official Review · Reviewer_RFmV · 2024-10-27

**Soundness:** 2
**Presentation:** 2
**Contribution:** 2
**Rating:** 5
**Confidence:** 3

**Summary:**

This paper presents a method to identify interpretable and influential directions in latent spaces of deep learning models. Since the latent space directions play a crucial role in understanding and correcting deep models, the goal here is to identify directions in the latent space that influence model predictions while being interpretable. Traditional approaches for finding these directions rely on supervised methods and predefined concepts. This paper introduces an unsupervised framework that combines signal vectors with uncertainty region alignment to discover interpretable latent directions.

**Strengths:**

1. The attempt to discover influential directions in a model's latent space without relying on labeled data is good. If it works, it will make the method more scalable compared to other techniques that require manual annotation or predefined concepts.

2. The method has been tested not just on synthetic datasets but also on real-world tasks using state-of-the-art models like ResNet18.

3. The writing can be improved by incorporating more motivations in introducing each loss.

**Weaknesses:**

While I commend the authors for developing an unsupervised method to identify interpretable directions, the approach faces a key challenge. Supervised methods generally ensure that these directions align closely with actual concepts, providing a clearer sense of interpretability. But unsupervised methods can not guarantee that.

The paper lacks experiments that measure cosine similarity with "ground truth concept vectors" on real datasets, such as Places365. This raises doubts about the authenticity of the learned vectors, as evidenced by the significant performance gap between unsupervised and supervised methods in Table 4.

Putting together, if the learned "concept vector" doesn’t reliably represent the true concept, the claim of interpretability becomes less compelling.

**Questions:**

N/A

---

> ### Author Response · Authors · 2024-11-18
> **Comment on weakness #1: Supervised methods "guarantee" interpretability while unsupervised methods do not**
>
> >While I commend the authors for developing an unsupervised method to identify interpretable directions, the approach faces a key challenge. Supervised methods generally ensure that these directions align closely with actual concepts, providing a clearer sense of interpretability. But unsupervised methods can not guarantee that.
>
> - **Reply**: We understand the possible confusion of the reviewer. In some way, indeed yes. But from another viewpoint this is like stating that unsupervised semantic segmentation should not be called "semantic segmentation" because there is no guarantee that it can reliably detect semantic classes aligned with human intuition. Moreover, in post-hoc explainability, a direction which corresponds to the concept "car", as learned by a supervised method, with the linear classifier achieving 0.2 F1Score, might be considerd low and not interpretable. This means that "Supervised methods generally ensure that these directions align closely with actual concepts, providing a clearer sense of interpretability. " is accurate only under the assumption that the features of the considered concept are linearly separable, which in general may or may not be true (for instance consider concepts appearing in out-of-domain images). Interpretability not only depends on the method that finds the directions, but also on the network's weights themselves (i.e. the ability of the network to have the concepts linearly disentangled). The proposed approach is bottom-up (i.e. it identifies concepts from the scope of the model) like other unsupervised works in the literature (such as "Disentangled explanations of neural network predictions by finding relevant subspaces", Chormai et al, "Invertible concept-based explanations for cnn models with non-negative concept activation vectors" Zhang et al, or "Sparse subspace clustering for concept discovery (SSCCD)", Vielhaben et al.).Yet, unlike the previously mentioned unsupervised works which not all of them have a linear model for concept discovery, the proposed method considers a linear model for concepts and instead of aligning the directions with human intuition (through annotations), it ensures that the underlying assignment of representations to concepts is sparse, which, as discussed in Doumanoglou et al 2023, is an essential property of interpretability. We believe that, although there is no guaratee that the proposed method suggests concepts that are aligned with human intuition, its main strength is that it is bottom-up, which is essential for model debugging: i.e. it identifies concepts influential (that is another word for relevant) to the network's predictions which is a property extremelly useful for model debugging and correction, especially if one wants to identify peculiar or weird counfounding features that the network uses for its predictions.

---

> ### Author Response · Authors · 2024-11-18
> **Comment on weakness #2: Cosine similarity with ground truth concept directions**
>
> >The paper lacks experiments that measure cosine similarity with "ground truth concept vectors" on real datasets, such as Places365. This raises doubts about the authenticity of the learned vectors, as evidenced by the significant performance gap between unsupervised and supervised methods in Table 4. Putting together, if the learned "concept vector" doesn’t reliably represent the true concept, the claim of interpretability becomes less compelling.
>
> - **Reply**:  We respect the viewpoint of the reviewer, as indeed we agree that such a comparison with ground truth would make sense. Yet, unfortunately, we do not know of any method that is able to extract ground truth concept vectors on real dataset like Places365, and this is why we evaluated our method in a synthetic experiment. This problem has been a whole area of research while a detailed discussion was provided in Pahde 2024. The best estimator of prior work that we know of is pattern-CAVs (Pahde 2024) - and we compare against Pattern-CAVs regarding influence in the experiment section. However, as we showed in the synthetic experiment, even pattern-CAVs do not align with the ground-truth directions in a controlled experiment under the extended data model that we introduced in Section 2. In constrast, our work makes a contribution and advances over the data model of Pahde by considering multiple concepts instead of only the binary setting that was studied in the prior work and also solves the controlled experiment of Section 4.1, Indeed, the performance of the learned classifiers in pixel classification and segmentation tasks are subpar compared to a supervised approach, but this should not be a surprise when working without labels. Whether the gap is significant or not, we think it depends on what we are comparing against. Compared to supervised, it could be. But we also significantly improve upon the unsupervised prior work of Doumanoglou 2023;2024 (appendix A.8). At this point we argue that the method is aiming to provide explanations from the viewpoint of the model instead of the viewpoint of a human in order to reveal its strategy in making predictions (which can be accomplished by complementing our work with existing literature, as we have already discussed in the answer to Reviewer #2: HouE). For instance, the experiment for Resnet18 reveals that the directions learned with best signal estimator of prior work (pattern-CAVs which uses annotations for its estimation) do not influence model's predictions more than the signal vectors learned with the proposed method. Thus, in a more relaxed sense, one might argue that the proposed method does not find directions that align the best with human intuition, yet, the proposed method finds directions that are more influential to the model's predictions than directions of human intuition while being "as interpretable as possible" by enforcing sparsity of the explanations. Again, we do understand the concern of the reviewer regarding the alignment of discovered concepts with human intuition, but we also think that this can also be concept definition issue. We would like to also point the reviewer in our comments to Reviewer #3: 7aZ9 regarding the "Comment on weakness #3: Unclear if Recovered signals are interpretable". and [this](https://openreview.net/forum?id=yeEWZ8qvlS&noteId=fbO7izC558) comment to Reviewer #1:

---

### Official Review · Reviewer_7aZ9 · 2024-10-31

**Soundness:** 2
**Presentation:** 2
**Contribution:** 2
**Rating:** 5
**Confidence:** 3

**Summary:**

This paper proposes to automatically learn the set of concept vectors underlying a given model representation in an unsupervised manner. Specifically, given a model representation and an unsupervised image dataset, the method aims to extract a set of "signal" vectors that explain the representation, and are semantically meaningful. Experimental results show that (1) in synthetic settings where the ground truth signals are known, the proposed method recovers the ground truth; and (2) when applied for an image classification setting, the extract signal vectors score high on numerical interpretability metrics.

**Strengths:**

+ The topic of extracting a set of contributing semantic concepts underlying a representation is an important topic, that can potentially help efforts in debugging and controlling model behaviour.

+ The proposed method is somewhat empirically successful, in that it is able to recover ground truth signals in synthetic settings; and recover signal vectors that score high on numerical interpretability metrics.

**Weaknesses:**

This paper is **hard to read**. At times, the issue is missing details, and at other times, it makes lots of references to text in the appendix and/or other papers. For instance, the method defines six heuristic loss functions, however it is unclear how they are combined and how the regularization parameters are set. The exact definitions and motivations for these are sometimes buried in the appendix. In addition, several terms defined in individual papers (Doumanoglou et al. (2023), Pfau et al. (2020), and Pahde et al. (2024)) are used without defining or explaining them. In particular, the contributions in these listed individual papers (S^1;S^2 interpretability metrics, RCAV, and Pattern CAV) are unexplained and used without context in the paper. I recommend that the authors rewrite these sections by clearly explaining the method, including being explicit about the exact loss functions used and the optimization problem being solved. In addition, it would help if terminology from other papers is first clearly defined before being used.

**Missing Comparison with PCA / ICA / Dictionary learning**: One of the goals of this paper is to recover signal and distractor vectors that combine additively to generate the underlying inputs. However, this is precisely the setting in which classical methods such as PCA / ICA / dictionary learning are applicable (depending on the exact underlying assumptions re: orthogonality, etc), and the missing discussion and comparison with these methods in the context of this paper is a significant omission. An experiment comparing the proposed method with such classical baselines would help build the case for the proposed approach. In addition, it would be helpful to point out what is conceptually missing in such classical approaches that necessitates the present one.

It is **unclear whether recovered signals are interpretable**, in the sense that an unambiguous assignment exists from the signal vector to a semantic concept. In the interpretability literature, including network dissection (Bau et al., 2017), it has been discussed that not all nodes are interpretable in the sense that an assignment to a semantic concept is not always possible. If the authors can present visualizations of the set of recovered signal vectors and discuss what makes them interpretable, this would help claim the interpretability benefits of this approach.

**Minor**: The paper claims that its method overcomes a common weakness of concept-based methods, i.e., the requirement of annotated concept datasets (line 88). However, even the proposed method uses the Broyden concept dataset to annotate concept labels (line 412), and without this, it is unclear how the method recovers interpretable signals.

**Questions:**

- The "M" term used in the maximum margin loss does not appear in any other loss equations (except perhaps in equation 5, where it seems to be a different "M"). Can you please clarify how the margin loss "M" is calculated?

- Line 451 mentions "Significant Direction Count (SDC) and Significant Class-Direction Pairs (SCDP). SDC represents the number of learned signal vectors that significantly influence at least one of the model’s classes, while SCDP counts the total number of class-direction pairs in which the learned signal vector significantly affects the class." -> what is "significantly influenced" here?

- Conceptual clarification: Is the assumed data generation process in equation 1, identifiable? For example, if s_1 and s_2 are two signal directions, then is s_1 + s_2 also a signal direction?

- Conceptual clarification: given a candidate direction "v", what is a simple test to determine whether it is a signal or a distractor direction?

---

> ### Author Response · Authors · 2024-11-18
> **Comment on weaknesses #1: Regarding the difficulties in comprehension**
>
> >This paper is hard to read. At times, the issue is missing details, and at other times, it makes lots of references to text in the appendix and/or other papers. For instance, the method defines six heuristic loss functions, however it is unclear how they are combined and how the regularization parameters are set. The exact definitions and motivations for these are sometimes buried in the appendix. In addition, several terms defined in individual papers (Doumanoglou et al. (2023), Pfau et al. (2020), and Pahde et al. (2024)) are used without defining or explaining them. In particular, the contributions in these listed individual papers (S^1;S^2 interpretability metrics, RCAV, and Pattern CAV) are unexplained and used without context in the paper. I recommend that the authors rewrite these sections by clearly explaining the method, including being explicit about the exact loss functions used and the optimization problem being solved. In addition, it would help if terminology from other papers is first clearly defined before being used.
>
> - **Reply**: We admit that this is true. For instance $\mathcal{S}^1$ and $\mathcal{S}^2$ are provided in the appendix section A.7 with a reference in the main text while in other cases we directly cite the respective paper without much explanation. Indeed, this work builds and extends upon ideas from Doumanoglou, and Kindermans while being closely related to works like Zhou, Pahde, Pfau, Kim. We find it difficult to balance the space requirements between explaining background concepts from 6 papers while also presenting our contributions and thoroughly evaluating them. This is why we made a good use of the appendix (the total paper length is 32 pages, 22 pages more than the page limit). Based on the comments of all the reviewers, we offer to revise the main text in order to include more details from background papers. However, this cannot be done without significantly shortnening other parts of the paper including the evaluation section. We are open to suggestions on which parts of the main text may be ommited from the final version if the reviewer has any concrete propositions.

---

> ### Author Response · Authors · 2024-11-18
> **Comment on weakness #2: Comparison with classical decomposition/factorization methods PCA/ICA/Dictionary Learning**
>
> >Missing Comparison with PCA / ICA / Dictionary learning: One of the goals of this paper is to recover signal and distractor vectors that combine additively to generate the underlying inputs. However, this is precisely the setting in which classical methods such as PCA / ICA / dictionary learning are applicable (depending on the exact underlying assumptions re: orthogonality, etc), and the missing discussion and comparison with these methods in the context of this paper is a significant omission. An experiment comparing the proposed method with such classical baselines would help build the case for the proposed approach. In addition, it would be helpful to point out what is conceptually missing in such classical approaches that necessitates the present one.
>
> - **Reply**: We appreciate and thank the reviewer for this constructive comment. Indeed, the reviewer points towards a path of additional motivation for the current work. This path was not considered in the paper mostly due to space limitations. However, we offer to discuss comparison with classical approaches in the appendix of the revised version. Below we list some theoretical arguments on why these classical approaches might fail to recover the true signal directions within the data and why we expect them to also fail in our experiment on synthetic data:
>     1) PCA would reconstruct representations based on orthogonal components. For the potential of this to work there should be an additional orthogonality assumption across all signals and distractors. This is a rather a strict assumption which we do not have empirical evidence to be true. Even if we could make that assumption, then a second stronger assumption should be made about the latent space: concepts are encoded in directions of decreasing variance, for which, again to our knowledge, there is not such evidence. PCA is guaranteed to fail to solve our toy-experiment on synthetic data since the signal directions in that example are not orthogonal (The reviewer may also refer to Table 9 in A.6 for cosine similarities between signals and distractors in that experiment).
>     2) If we consider our proposed data model under the lens of ICA, and if we ignore the pre-processing whitening tranform for the shake of discussion, we can say that ICA assumes independence for all $\alpha_{p,i}$ and $\beta_{p,f}$. By assumption, in our data model, a distractor component ($\beta_{p,f}$) is indeed considered independent of signal components or other distractors. Yet, the signal components $a_{p,i}$ are not. More specifically the $\text{Cov}[a_{p,i},a_{p,j}]$ is expected to be negative since the presence of concept $i$ in $x_p$ implies non-presence of all the other concepts $j$. Since two independent variables have covariance zero, and we identified that $\text{Cov}[a_{p,i},a_{p,j}]$ is $< 0$, we can rather easily suspect that the solution to our problem does not lie in ICA's solution space. We also validated our claim that ICA's suggestion for the signals of the toy synthetic example is worse in terms of cosine similarity compared to the almost perfect solution suggested by our method.
>     3) Dictionary Learning with sparsity constraints is also fundamentally different than our approach in the following way: in Dictionary Learning, sparsity constraints are enforced in the units of latent variables (i.e. in the units of feature space). Yet, our proposed data model implies sparsity in the "semantic space", i.e. the space of concepts defined as:
>     $$
>     \mathbf{c}_\mathbf{p} =  \sigma(\mathbf{W\mathbf{x_p}-b}) \in [0,1]^{I}
>     $$
>     where $\mathbf{W}$ summarizes the filter directions of the concept detectors and $\mathbf{b}$ summarizes the concept detectors' biases. The latter assumption is a more accurate assumption to make than enforcing sparsity in the units of feature space. Provided we do have access to $\mathbf{W}$ we recover the concepts' signal directions using (4). In other words, in a Sparse dictionary representation it is implied that the representation has a zero coefficient for some of the concepts and distractors, which is a stronger assumption than ours.

---

> ### Author Response · Authors · 2024-11-18
> **Comment on weakness #3: Unclear if Recovered signals are interpretable**
>
> >It is unclear whether recovered signals are interpretable, in the sense that an unambiguous assignment exists from the signal vector to a semantic concept. In the interpretability literature, including network dissection (Bau et al., 2017), it has been discussed that not all nodes are interpretable in the sense that an assignment to a semantic concept is not always possible. If the authors can present visualizations of the set of recovered signal vectors and discuss what makes them interpretable, this would help claim the interpretability benefits of this approach.
>
> - **Reply**: We understand the confusion of the reviewer, especially as visualizations are not provided in the main text. Answering this question is the reason for the experiment we did, for instance, in Table 4. We use this quantified manner of testing interpretability because it is less susceptible to personal interpretation than visualizations. In other words, we might actually say that "how much interpretable a direction is?" can be (in some way) quantified and we present this evaluation in Table 4, where **per-pixel** classification accuracy of the learned classifiers is reported based on the labels assigned to the linear classifiers from NetDissect. The learned classifiers have high precision (0.81 on average) indicating that most of the times the directions are highly interpretable. Indeed, some classifiers may have a low classification score based on their assigned label and this might be a concept definition issue. For instance, imagine a classifier that accurately detects the concept "blue AND NOT water AND NOT sky". Clearly, to identify this concept one needs to have annotations for such a concept which is rather uncommon or the labeling method considering logical combinations of the individual concepts "blue", "water" and "sky" (such as  Mu J, Andreas J. Compositional Explanations of Neurons. In: Larochelle H, Ranzato M, Hadsell R, Balcan MF, Lin H, eds. Advances in Neural Information Processing Systems. Vol 33. Curran Associates, Inc.; 2020). Although the learned directions are not guaranteed to align with any "easily identifiable" human concept, the learned classifiers fulfill a sparsity property that is essential for interpretability as discussed in detail in Doumanoglou et al 2023. This property is motivated by the fact that if the semantic meaning of an image patch belongs to concept A, then this patch should most probably not be a representative of other possible concepts such as B,C,D, .. Z. Thus in the "semantic space" this patch has a sparse representation. The transformation from the standard feature space to the semantic space is accomplished via the application of the learned linear classifiers on the representation of the patch. The classifiers learned by the proposed method (and the ones from Doumanoglou et al) fulfill this property. Qualitative results for the segmentation ability of the learned classifiers are provided in appendix section A.10.

---

> ### Author Response · Authors · 2024-11-18
> **Comment on weakness #4: How is broden used to identify concepts ?**
>
> >Minor: The paper claims that its method overcomes a common weakness of concept-based methods, i.e., the requirement of annotated concept datasets (line 88). However, even the proposed method uses the Broyden concept dataset to annotate concept labels (line 412), and without this, it is unclear how the method recovers interpretable signals.
>
>   - **Reply**: We think it is important to address this issue, since it seems to originate from an important misunderstanding, possibly caused by the fact that we did not explicitly mention the details and instead we refered to Doumanoglou 2023 in line 411. Even though we use the Broden Dataset to discover the learned directions, we only use the images from this dataset and not the dataset's annotations. This is only for direction learning. However, in a subsequent step, and in order to quantitatively evaluate the method, we need to assign concept label names to each one of the learned classifiers. This is where Broden (with labels) and NetDissect are used. For more details on how Broden and NetDissect is used in this method we kindly ask the reviewer to refer to the reply comment on Reviewer #2: HouE.

---

> ### Author Response · Authors · 2024-11-18
> **Answers to the questions**
>
> >The "M" term used in the maximum margin loss does not appear in any other loss equations (except perhaps in equation 5, where it seems to be a different "M"). Can you please clarify how the margin loss "M" is calculated?
>
>   - **Reply**: We think the reviewer refers to $M_i$ of the maximum margin loss and equation 6 for the other "M" instead of equation 5. Indeed the two "M"s are different. The definition of $M_i$ can be found in the Section 2 line 133. In short, it is a learnable parameter which quantifies the margin of the linear classifiers. Largers values of $M_i$ indicate better separation of concepts in the latent space, from the perspective of the model. If the reviewer meant something else, please let us know.
>
> >Line 451 mentions "Significant Direction Count (SDC) and Significant Class-Direction Pairs (SCDP). SDC represents the number of learned signal vectors that significantly influence at least one of the model’s classes, while SCDP counts the total number of class-direction pairs in which the learned signal vector significantly affects the class." -> what is "significantly influenced" here?
>
>   - **Reply**: The term "significant direction", is taken from the terminology of RCAV. RCAV suggests a model-to-concept sensitivity test and to measure if this sensitivity is significant, a hypothesis significance test is performed based on a randomized experiment where the sensitivity of the network on a concept vector (here our signal vector) is compared against the sensitivity of the network against a random vector. Here "significantly influences" means that the signal vector passes RCAV's significance test with a p-value of 0.05 and Bonferroni correction.
>
> >Conceptual clarification: Is the assumed data generation process in equation 1, identifiable? For example, if $s_1$ and $s_2$ are two signal directions, then is $s_1 + s_2$ also a signal direction?
>
>   - **Reply**: In the data model of equation 1 its important to identify $s_1$ and $s_2$ separately since by assumption they constitute the actual directions of concept 1 and 2. The sum of $s_1+s_2$ would imply that the pixel (or patch) belongs to two concepts, both $s_1$ and $s_2$. Again this is somewhat vague. For instance imagine the concepts "yellow" and "car". If we want to identify primitive concepts, we would identify the direction of "yellow" ($s_1$) and "car" ($s_2$) separately. However, if the dataset contains only "yellow cars" then separating the two concepts might also be impossible, since we need to see for instance "blue cars" in order to identify the difference
>
> >Conceptual clarification: given a candidate direction "v", what is a simple test to determine whether it is a signal or a distractor direction?
>
>   - **Reply**: Nice question! We apologize  for not making it clear in the text. This is partially explained in detail in Kindermans et al. In short, in our data model, a distractor direction does not have predictive power over the concept compared to the signal direction. In essence, distractors are noisy components that are present in the representation $\mathbf{x}\_p$. As we mention in Section 2, we could say that for a signal direction (i.e. the direction of a concept) the components $a\_{p,i}$ for all $p$ containing the concept $i$ have strong values, for instance > 2. In constrast, $a\_{p,i}$ for $p$ NOT belonging to the concept should have weak values, in our example < 2. Thus the signal direction $i$ has predictive power over the concept since by extracting the component $a\_{p,i}$ (the value of the signal) from $\mathbf{x}_{p}$ and comparing it to $2$ we can classify the pixel representation to whether it belongs to concept $i$ or not. In constrast, in a distractor direction $f$, $\beta\_{p,f}$ follows a gaussian distribution across **all** $p$ whether they belong to the concept or not, and we cannot easily find a threshold against which we can compare $b\_{p,f}$ to identify if the pixels belong to concept $i$. To state it a bit more formal, a signal direction is informative regarding the presence of a concept while a distractor is not.

---

> > ### Comment · Reviewer_7aZ9 · 2024-11-26
> >
> > I thank the authors for their rebuttal.
> >
> > Unfortunately, most of my main concerns remain:
> >
> > 1. The paper is hard to read, and the rebuttal indicates that it may be non-trivial to re-write this in a self-contained manner in 10 pages.
> >
> > 2. It is understandable that simple methods like PCA / ICA / dictionary learning may not precisely fit the requirements of this paper, but it is still instructive to compare against them and present quantitative metrics to motivate the proposed method.
> >
> > 3. The answers to the conceptual questions are unclear. For the question about identifiability, note that unique recovery of signal vectors is only possible if the underlying generative process (eqn 1) is identifiable. Note that this is not the case for PCA, where principal components are only identifiable up to a rotation, however ICA is identifiable. The potential lack of identifiability invalidates the claim that "true" signal directions can be recovered.
> >
> > 4. Regarding how to determine whether a candidate direction "v" is a signal or a distractor, the author response states, "a signal direction is informative regarding the presence of a concept while a distractor is not". However, it is unclear how to determine whether a direction is informative regarding the presence of a concept, especially as the response to another question states that "... the learned directions are not guaranteed to align with any "easily identifiable" human concept ...".
> >
> > Given these, I am inclined to keep my score.

---

> > > ### Author Response · Authors · 2024-12-01
> > >
> > > >Regarding how to determine whether a candidate direction "v" is a signal or a distractor, the author response states, "a signal direction is informative regarding the presence of a concept while a distractor is not". However, it is unclear how to determine whether a direction is informative regarding the presence of a concept, especially as the response to another question states that "... the learned directions are not guaranteed to align with any "easily identifiable" human concept ...".
> > >
> > > We would like to kindly ask the reviewer to refer to Section 2.2 of the revised manuscript. In the signal-distractor data model, the information about a concept, is encoded in the amplitude of the signal component in the data. When we do have access to ground-truth directions, we are able to test whether one of them is signal or distractor, by constructing a classifier based on a regression filter, which extracts the value of the directional component in the data, and measure its ability to distinguish samples with the concept from samples without. In absence of ground-truth, we are only able to estimate the signal direction based on assumptions about the data. Such signal estimators are the Pattern-CAVs of prior work and the Signal Vectors that we propose here.
> > >
> > > Even though concepts may not align with human intuition, they may be defined by a set of positive and negative examples. This does not interfere with the rest of the approach.

---

### Official Review · Reviewer_HouE · 2024-11-03

**Soundness:** 2
**Presentation:** 2
**Contribution:** 2
**Rating:** 3
**Confidence:** 4

**Summary:**

This paper proposes a method to learn interpretable and influential directions in a trained model’s latent space, where these directions are modeled as linear “signal” classifiers. The authors propose a method called Uncertainty Region Alignment that aligns the subspaces where the model is uncertain and the subspaces where the learned “signal” classifiers are also uncertain, with the claim that this enhances the interpretability and influence of the learned directions. They present results from a small synthetic setting and from a slightly larger, more realistic setting on a single model and dataset (Resnet18, Places265), where they benchmark against IBD and find that their method produces better concept detectors.

**Strengths:**

* The problem of finding concepts or latent directions that are highly influential, which I take to mean causal, is an interesting and very active research area at the moment. Being able to use interpretability tools to exert influence over a model would be of high utility for future research and real-world applications.

**Weaknesses:**

- As it stands, it was difficult for me to understand the contribution of this work with respect to prior literature in this field. Further discussion of how this paper relates to the wide body of recent literature on generating concept-based explanations, including follow-up works to ACE/ICE/IBD such as CRAFT [1], work on Concept Bottleneck Model literature [2, 3, 4, 5], and recent work on sparse autoencoders and dictionary learning, would help clarify the gap in existing literature that this paper is trying to fill.
- Furthermore, it seems like this paper is a close follow-up to the papers by Doumanoglou et al. (2023; 2024) cited in the paper, in that the majority of the method is the same with some minor adjustments. Can you provide further discussion of how this method and its contributions differs from the methods in those two papers?
- The writing and presentation of this paper are not very clear to me - I struggled to understand the types of explanations given by the proposed method, and how those explanations could be used to gain a  “deeper understanding of the model’s strategy, fostering trust, and enabling model correction and improvement” as stated by the authors in the abstract. Clarifying the intent of the proposed method and how the experiments validate its practical applications with respect to the stated goals from the abstract and introduction would significantly improve the paper.
- Given that this paper is an explainability/interpretability paper, I believe it could strongly benefit from providing some example explanations yielded by the paper, and a more in-depth example or case study of how the explanations can be applied to the aforementioned end goals of model correction.

[1] Fel, Thomas, et al. "Craft: Concept recursive activation factorization for explainability." *Proceedings of the IEEE/CVF Conference on Computer Vision and Pattern Recognition*. 2023.

[2] Yuksekgonul, Mert, Maggie Wang, and James Zou. "Post-hoc concept bottleneck models." *arXiv preprint arXiv:2205.15480* (2022).

[3] Bhalla, Usha, et al. "Interpreting clip with sparse linear concept embeddings (splice)." *arXiv preprint arXiv:2402.10376* (2024).

[4] Oikarinen, Tuomas, et al. "Label-free concept bottleneck models." *arXiv preprint arXiv:2304.06129* (2023).

[5] Gandelsman, Yossi, Alexei A. Efros, and Jacob Steinhardt. "Interpreting CLIP's Image Representation via Text-Based Decomposition." *arXiv preprint arXiv:2310.05916* (2023).

**Questions:**

- Are the learned directions always labeled by NetDissect? I would appreciate some clarity on how NetDissect is being applied to the proposed method and its necessity in generating explanations. If so, why is the proposed method preferable to simply using NetDissect on its own?

- Please see above weaknesses.

---

> ### Author Response · Authors · 2024-11-18
> **Comments on the summary and strengths**
>
> >This paper proposes a method to learn interpretable and influential directions in a trained model’s latent space, where these directions are modeled as linear “signal” classifiers. The authors propose a method called Uncertainty Region Alignment that aligns the subspaces where the model is uncertain and the subspaces where the learned “signal” classifiers are also uncertain, with the claim that this enhances the interpretability and influence of the learned directions. They present results from a small synthetic setting and from a slightly larger, more realistic setting on a single model and dataset (Resnet18, Places265), where they benchmark against IBD and find that their method produces better concept detectors.
>
> - **Reply**: Regarding the following comment of the reviewer: "They present results from a small synthetic setting and from a slightly larger, more realistic setting on a single model and dataset (Resnet18, Places265), where they benchmark against IBD and find that their method produces better concept detectors." we think there is a misunderstanding. In Section 4.2 we provide experimental results for Resnet18 trained on Places365 (not Places265 which is another -smaller- dataset). Regarding the interpretability part, we compare the performance of the classifiers learned with the proposed method against the classification performance of classifiers learned in a supervised way, following the protocol introduced by IBD. In the text we commented on how the proposed approach compares to IBD but never claimed that the proposed method learned more interpretable classifiers than IBD. After all, since IBD is a supervised approach, in a broad sense, it roughly constitutes an upper-bound on "how much" interpretable a set of classifiers learned in an unsupervised way may be, at least for the concept set that is present in the concept dataset. In the text, we also did not use the term "signal classifiers" and we are not sure if the understanding of the reviewer is aligned with the mearning we would like to convey.
>
> >The problem of finding concepts or latent directions that are highly influential, which I take to mean causal, is an interesting and very active research area at the moment. Being able to use interpretability tools to exert influence over a model would be of high utility for future research and real-world applications.
>
> - **Reply**:  We think that there may be some precariousness regarding the term "causal" and that's why we would refrain from using this term in our context. Instead, we would like to link the term "influential" to the term "sensitivity" which is taken from related work (Kim etal, Pfau et al). In simple terms, a (signal) direction is "significantly influential" if the model's predictions are sensitive to this direction and this sensitivity passes a statistical significance test against the sensitivity of the model in random directions. We undestand that this might not be well explained in the text. Please also refer to our general comment to all the reviewers and our subsequent comments below.

---

> ### Author Response · Authors · 2024-11-18
> **Comments on weaknesses**
>
> >As it stands, it was difficult for me to understand the contribution of this work with respect to prior literature in this field. Further discussion of how this paper relates to the wide body of recent literature on generating concept-based explanations, including follow-up works to ACE/ICE/IBD such as CRAFT [1], work on Concept Bottleneck Model literature [2, 3, 4, 5], and recent work on sparse autoencoders and dictionary learning, would help clarify the gap in existing literature that this paper is trying to fill.
>
> - We understand the confusion and the fact that this might be not very clearly explained in the text. Yet, some discussion regarding some of these works has been written in A.1. In particular we describe how our work is different compared to ACE/ICE and IBD and also that our work can constitute a building-block to Post-Hoc Concept Bottleneck Model literature (this applies to [2]). First [2] is post-hoc while [4] is inherently-interpretable. In [2], the concept-subspace is learned with supervision (in a way similar to IBD or TCAV). In that part, our method could offer an unsupervised alternative with the limitation of not directly providing labels for the learned concepts (in which case someone would need to obtain labels through Network Dissection or a similar method or even asking a Vision Language model to generate a textual description for the images of each concept). Most importantly though, our work targets debugging and is bottom-up, that is, the concepts that we learn are from the perspective of the model with the aim to identify what concepts are important to the model in order to make predictions. A strategy like [2], speculates on what concepts **might be/or even should be** relevant to the prediction of class X and aims to create a concept-bottleneck based on that. However, such a strategy would not be very effective in identifying the reliance of the network to seemingly irrelevant features within the data, such as image watermarks or other counfounding correlated features. The same line of thought applies when comparing against most other post-hoc concept-bottleneck methods. In [4] the authors combine GPT-3 and CLIP to create the concept set and learn the concept-bottleneck. Our method is not directly comparable to them since their target is different than ours. They aim in constructing an inherently interpretable model while our method aims to explain, under the scope of debugging, an already trained network. For [1,3,5] we are going to comment in the following days. Regarding a sparse autoencoder scheme, we would argue that using a deeper model than a simple linear layer for modeling concepts does not enjoy the benefits of model debugging and correction that already exists in the literature. For Dictionary Learning, please refer to our answer for Reviewer #3: 7aZ9

---

> ### Author Response · Authors · 2024-11-18
> **Comments on weaknesses #2**
>
> >Furthermore, it seems like this paper is a close follow-up to the papers by Doumanoglou et al. (2023; 2024) cited in the paper, in that the majority of the method is the same with some minor adjustments. Can you provide further discussion of how this method and its contributions differs from the methods in those two papers?
>
> - **Reply**: In our opinion, the biggest leap forward in this work is three-fold, and in the following order: a) the extension of the "signal-distractor" data-model in a multi-label (multi-concept) setting, beyong the original binary-label formulation of prior work (Kindermans, Pahde). b) The introduction of signal vectors as estimators of the true concepts' signal directions and c) the Uncertainty Region Alignment Loss which is an advancement over the "CNN Classifier Loss" of Doumanoglou et al 2024. Regarding empirical evidence, in a synthetic experiment, we have shown that signal vectors faithfuly recover the true signal direction within the data, under the newly introduced multi-concept data-model. The latter was not possible with the prior work of Doumanoglou 2023;2024, as both works disregard the "signal-distractor" theory. Empirically, we have also improved on the interpretability of the learned concept directions compared to both previous unsupervised works (Doumanoglou et al 2023;2024). Although not in the main text, in A.8 we compare against Doumanoglou's work in interpretability terms and provide evidence that our proposed $\mathcal{L}^{uur}$ improves on interpretability metrics upon the prior work by a significant margin of up to 78.78% in $\mathcal{S}^2$. In other words, in A.8 we showed that, our proposed unconstrained Uncertainty Region Alignment Loss can improve upon the findings of Doumanoglou et al, even if we keep the rest of their method intact (that is without any of our contributions of Sections 3.1 or 3.2 or $\mathcal{L}^{cur}$). We achieve additional improvements in interpretability, at least for the most difficult to optimize metric $\mathcal{S}^2$, as we lift the prior work's orthogonality and feature standardization constraints while at the same time introducing our losses from Section 3.1. For instance, compare the second column of Table 10 with the last row of Table 2. When additionally taking into account the theory behind signal vectors, in the main paper we also provided emprical evidence that $\mathcal{L}^{fso}$ and $\mathcal{L}^{cur}$ from sections 3.2 and 3.3 further improves on the quality of the discovered directions (Table3 , right, last row). To conclude, none of (a,b,c) was considered in the prior works of Doumanoglou 2023;2024. Considering this set of contributions made and the empirical evidence that we provided, we would argue that this paper goes beyond a minor improvement over prior work. To our knowledge this is the first unsupervised method to consider the "signal-distractor" theory, which is important when trying to answer questions about concept influence on the studied network's predictions.

---

> ### Author Response · Authors · 2024-11-18
> **Comments on weaknesses #3**
>
> >The writing and presentation of this paper are not very clear to me - I struggled to understand the types of explanations given by the proposed method, and how those explanations could be used to gain a “deeper understanding of the model’s strategy, fostering trust, and enabling model correction and improvement” as stated by the authors in the abstract. Clarifying the intent of the proposed method and how the experiments validate its practical applications with respect to the stated goals from the abstract and introduction would significantly improve the paper.
>
> >Given that this paper is an explainability/interpretability paper, I believe it could strongly benefit from providing some example explanations yielded by the paper, and a more in-depth example or case study of how the explanations can be applied to the aforementioned end goals of model correction.
>
>   - **Reply**: We understand the confusion and appologise for this lack of clarity. Let us try to amend: the sentence "deeper understanding of the model’s strategy, fostering trust, and enabling model correction and improvement” that comes from the abstract, is, in fact, related to any concept **direction** discovery method (and not concept discovery in general). In the first paragraph of the introduction we cite several papers where this is actually demonstrated but without commenting on the exact details of each method. To pick one approach that we also use in the experiments RCAV (Pfau et al 2020), which is a follow-up work of "Testing with Concept Activation Vectors (TCAV - Kim et al 2018)", the authors propose a way to quantify the sensitivity of an image classifier to a human concept of interest, when making predictions for a specific class (Global XAI). For instance, this method may address the question how much the concept "stripes" influence the prediction of class "zebra". In that method, first, the direction of concept "stripes" is estimated in the latent space of the network and subsequently representations of images depicting zebras are manipulated towards the "stripe" direction and the impact on the prediction probability of class zebra is measured. An average score is computed across a set of images depicting zebras in order to quantify the previously mentioned research question. In the original paper this direction (which is called a Concept Activation Vector - CAV) was proposed to be equal to the normal of the hyperplane that separates representations of images with the concept from representations of images without. In practice a linear SVM was used to access this normal. In this method, a key component for success is to reliably identify the direction of the concept in the deep network's latent space since assessing concept sensitivity relies on a trustworthy concept direction. This is one example where directions can help for "deeper understanding of the model’s strategy (and) fostering trust".
>
>   - Regarding "model correction and improvement", we may refer to (Pahde et al 2023 and Anders et al 2022). Let's take a concrete real simple example: ImageNet. Anders et al has shown that, for instance, some images of the ImageNet class "Garbage Truck" contain a watermark in the bottom left corner of the image which is exploited by VGG-16 for making predictions of this class (i.e. the network predicts the class based on the presence of the watermark in the image). In their simplest approach (Projective Class Artifact Compensation P-ClArC), which does not require any form of training or fine-tuning of the subject network, the direction of the watermark in the feature space is identified using a CAV (i.e a concept direction where the concept is images with the watermark itself). Then, the network's code is "patched" to suppress feature components in the direction of the watermark CAV. In other words, at inference time, when a feature is identified to correspond to a watermark (the linear classifier behind the CAV can answer this question) or even always, the feature itself is manipulated to suppress its component in the direction of the watermark CAV. In that way, the subsequent layer of the network does not "see" a strong component in the watermark direction and thus it falls back to classifying the image based on the rules (e.g. other concepts) it has learned besides the watermark (these rules would have been learned based on images of garbage trucks that do not contain the watermark). This improves the generalization of the network by using more reliable features than watermarks.

---

> > ### Author Response · Authors · 2024-11-18
> > **Overall**
> >
> > - Overall, this sentence from the abstract answers the question "why direction learning is an important problem?". Since in the title we mention "interpretable and influential directions" we evaluate our approach for interpretability and influence and concrete example explanations are provided in the appendix, including visualizations of the segmentation performance of the learned concept detectors. Model correction is a possible area in which our method could be useful, but applying our method to model correction as a part of this work faces significant difficulties regarding the space requirements of the manuscript.

---

> ### Author Response · Authors · 2024-11-18
> **Answer to question regarding NetDissect**
>
> >Are the learned directions always labeled by NetDissect? I would appreciate some clarity on how NetDissect is being applied to the proposed method and its necessity in generating explanations. If so, why is the proposed method preferable to simply using NetDissect on its own?
>
> - **Reply**: We appologise for the confusion. Indeed, in the main text, details on this one are not described and the reader is asked to refer the cited paper of Doumanoglou 2023. Let us try to clarify. The directions are learned in an unsupervised way and do not rely in whatsoever on NetDissect. The necessity of NetDissect stems from the fact that in order to quantitatively assess and compare the method to the supervised approach we need to have a label for each direction learned with the proposed method. (Think of it like hungarian matching in unsupervised segmentation). In the absence of labels during direction learning, this is not possible. We don't know which concept name is associated with every learned direction. Thus, we need to measure the suitability of each classifier in detecting every possible concept in the dataset in order to identify which concept it detects better. This is what NetDissect is used for. We chose to label the directions with NetDissect and the Broden concept dataset, but any other labeling method or concept dataset could be used (e.g. Mu J, Andreas J. Compositional Explanations of Neurons. In: Larochelle H, Ranzato M, Hadsell R, Balcan MF, Lin H, eds. Advances in Neural Information Processing Systems. Vol 33. Curran Associates, Inc.; 2020) - although we would advise that the dataset to label the directions would better be the dataset that was used to learn the directions, as we did in our experiments. In its original form, NetDissect is able to label the **natural feature space basis**. Our learned directions can be seen as a feature space transformation, ie. a basis change, like in Doumanoglou et al 2023, with the difference that in our approach the basis is not orthogonal. Again, the necessity of NetDissect stems from the fact that we need to **quantitatively assess** the performance of the learned classifiers and we cannot compute scores like IoU,Precision,Recall,F1 if we do not have both a labeled dataset and a label for the binary classifier that we are testing. NetDissect provides an answer to the question "which concept does classifier X detect" ? As already mentioned in Doumanoglou, the choice of NetDissect to label the classifiers, compared to other alternatives - such as Mu at al - , may affect the quantitative evaluation. There is no perfect solution since the proposed approach is unsupervised and identifies concepts from the perspective of the model. (For instance, a classifier that actually identifies watermarks can be mislabeled by NetDissect since Broden does not contain any concept related to watermarks. Or concepts like "blue but not water" that were considered in Mu et al, are missed by NetDissect although blue and water exist in the Broden dataset but logical combinations between concepts in the dataset are not considered by NetDissect). To answer the question "why is the proposed method preferable to simply using NetDissect on its own?", becomes more aparent if we take a step back and see what NetDissect does. NetDissect is a tool to assign concept labels to the directions of **any** feature space basis. Only that, in their original paper, they used the method in order to assign labels to the **natural** feature space basis. The directions learned with our method form a different feature space basis and we use NetDissect to assign concept labels to each one of the learned directions. Thus, the two approaches (ours and NetDissect) work in tandem and should not be considered competing. NetDissect is a tool to label the directions of a basis, and our method is a way to find a (more intepretable) basis (than the natural).

---

> > ### Comment · Reviewer_HouE · 2024-11-24
> >
> > Thank you to the authors for their detailed response. Similar to what reviewer 7yop noted, I belive your main message sometimes gets lost in what are very lengthy responses. I might suggest in the future that the authors strive to be more precise and succinct in their responses.
> >
> > * Regarding your "Comments on weaknesses #2," I want to note that the idea of a signal-distractor decomposition in a multi-label setting is not novel, see for example [1, 2] which explore this framework for attribution methods. I am also unconvinced of the novelty of "signal vectors as estimators of the true concepts' signal directions," as in practice learned features or directions are of course almost always approximations in post hoc settings, and this is acknowledged by many feature learning methods. Please let me know if I have misunderstood what you meant to say here. While the proposed loss is certainly novel w.r.t. Doumanoglou 2023;2024, I cannot be convinced that it truly improved interpretability without either experiments on downstream tasks requiring interpretability or human studies evaluating the said interpretability.
> >
> > * Regarding "Comments on weaknesses #3," while you have shown that prior methods can be used for either auditing or editing models, you have not shown that your methods can be used to reliably do so. In the provided ImageNet Garbage Truck/watermark example, does your method correctly learn the watermark concept direction and are you able to edit it out of the model? Please let me know if I missed your experiments showing so.
> >
> > * While I agree that there are space constraints in the paper, I think the writing of this paper could be condensed significantly, providing both greater clarity and making space for experiments that showcase the true utility of this method over prior quite similar methods. I acknowledge that the authors have noted they will improve the writing, but I believe the changes that need to be made go beyond what is feasible in the remaining rebuttal time, and that this paper should likely be improved upon and resubmitted to another venue. In particular, the two weaknesses I have noted in this comment are quite significant, and I would likely not increase my score until they are adequately resolved.
> >
> > [1] Srinivas, Suraj, Sebastian Bordt, and Himabindu Lakkaraju. "Which models have perceptually-aligned gradients? an explanation via off-manifold robustness." Advances in neural information processing systems 36 (2023): 21172-21195.
> > [2] Bhalla, Usha, Suraj Srinivas, and Himabindu Lakkaraju. "Discriminative feature attributions: bridging post hoc explainability and inherent interpretability." Advances in Neural Information Processing Systems 36 (2024).

---

> ### Author Response · Authors · 2024-11-27
>
> We thank the reviewer for her comment. Regarding the pointers to related work that discuss the signal-distractor model, although we did not have the time to go through them thoroughly, we got the impression that they consider a signal-distractor model in the input space. Again, the reviewer may correct us if we are wrong, but in our case, we consider a signal-distractor model for the encoding of concepts in the feature space which is different than the perspective of [1] and [2].
>
> Indeed in Post-Hoc explainability there is an inherent difficulty to define what is ground truth. Maybe the phrase "signal vectors are estimators of the true concept signal direction" that we used in the previous version of the manuscript was too bold. In the revised version we tried to avoid this bold statement and instead refer to them just as "signal estimators" that are able to recover the ground-truth in controlled settings (see introduction section of the revised manuscript). We are also not sure what the reviewer is asking for when referring to more experiments in downstream tasks or human studies evaluating the said interpretability. In this manuscript we take a quantitative approach to actually test how well each learned classifier is able to detect human known concepts (instead of measuring this subjectively through human studies) - please refer to Table 5 of the revised manuscript. The learned classifiers demonstrate high precision in identifying concepts aligned with human intuition. Concrete classifier performance is also reported in Section A.12.1. We also offer subjective evaluation to the readers of the manuscript in Section A.13 where we report example segmentations obtained with the learned classifiers. Quantitative interpretability comparison with Doumanoglou et al has also been added to the main text of the revised manuscript in Table 3 and should be judged in combination with what the proposed method additionally offers compared to their approach. In the revised version of the manuscript we mention the contributions of this work related to the previous work of Doumanoglou more clearly (e.g. in the ending of Section 1, Section 2.3 and beginning of Section 3), while more details are also provided in the appendix, Section A.11.
>
> Regarding the comment of the reviewer for model correction, honestly we think that, while this would significantly strengthen the position of our manuscript, a detailed study would not fit in the page limitations of this paper and it goes beyond its scope. The title of the paper mentions that we are proposing a method to learn interpretable and influential directions. Experiments of Section 4 aim to justify these claims. In the abstract, we aimed to motivate why direction learning is important, and we provided examples where directions play an important role, one of them being model correction. This is a motivation sentence and not a subject of our method. Despite that, because we understand the concerns of the reviewer and possibly other readers of the manuscript, in the appendix of this revision, we tried to provide an example with an experiment that demonstrates what the reviewer asked for, in a toy setting. We would like to kindly ask the reviewer to refer to section A.14 of the revised manuscript.

---

### Official Review · Reviewer_2ift · 2024-11-09

**Soundness:** 3
**Presentation:** 3
**Contribution:** 3
**Rating:** 6
**Confidence:** 2

**Summary:**

This paper introduces an unsupervised method for discovering interpretable and influential directions in the latent space of deep neural networks, specifically for image classifiers. The proposed approach leverages signal vectors and uncertainty region alignment to identify latent space directions that significantly influence model predictions while maintaining high interpretability. Unlike previous methods that require annotated concept datasets and predefined concepts, this method does not rely on prior knowledge and instead learns from the inherent structure of the feature space. The authors validate their approach on both synthetic data and real-world benchmarks, demonstrating that the discovered directions effectively fulfill critical debugging criteria and outperform supervised methods in certain aspects.

**Strengths:**

The paper presents a novel unsupervised framework for discovering latent space directions that are both interpretable and influential. This is an advancement over prior work that often requires annotated datasets or predefined concepts.

The method is theoretically grounded, extending previous models to a multi-label setting and introducing new loss functions such as the Uncertainty Region Alignment loss. The experimental results on synthetic and real-world data support the efficacy of the approach.

**Weaknesses:**

While the experiments are promising, the evaluation is limited to a synthetic dataset and a single real-world dataset (Places365 with ResNet18). A broader range of datasets and models would strengthen the claims and demonstrate the generalizability of the method.

The paper could provide a more thorough comparison with existing methods, including additional baselines in both interpretability and influence metrics. This would help in positioning the proposed method within the existing literature more effectively.

**Questions:**

Is the proposed method applicable to other types of data beyond images, such as text or time series data? If so, what modifications, if any, would be necessary?

How does the method scale with larger models and datasets? Are there computational bottlenecks, and if so, how can they be mitigated?

---

> ### Author Response · Authors · 2024-11-18
> **Regarding experiments and baselines**
>
> >While the experiments are promising, the evaluation is limited to a synthetic dataset and a single real-world dataset (Places365 with ResNet18). A broader range of datasets and models would strengthen the claims and demonstrate the generalizability of the method.
>
> >The paper could provide a more thorough comparison with existing methods, including additional baselines in both interpretability and influence metrics. This would help in positioning the proposed method within the existing literature more effectively.
>
>   - **Reply**: In appendix, section A.8, we provide additional experiments regarding the interpretability of the directions while comparing with the unsupervised work of Doumanoglou 2023;2024. In particular, we compare our (unconstrained) "Uncertainty Region Alignment Loss" compared to their "CNN Classifier Loss", while keeping the rest of the method intact to theirs, that is without signal vectors and with orthogonal interpretable directions and feature space standardization. In that experiment, we consider both ResNet18 trained on Places365 and also ResNet50 trained on Moments In Time (MiT) (a larger network trained on a much larger dataset). Table 10 showcases that our proposed loss demonstrates upto +78.78% improvement in the $S^2$ interpretability metric for Resnet50/MiT. This example, at least up to one extend, demonstrates the efficacy of of our method to another -larger- architecture and dataset while also comparing with an additional baseline for unsupervised direction learning (which indeed is missing from the main text). In the main text, we also compare for interpretability against a supervised baseline which is a typical baseline in finding interpretable directions. We are not aware of any significantly different way to obtain an interpretable direction besides standad linear models such as logistic regression (such as IBD), lasso regression or SVMs. One fundamentaly different approach (which mostly would fit as a signal direction estimation method) could be through feature matrix decomposition (such as PCA,ICA,Dictionary Learning or Non-Negative Matrix Factorization). For the latter, please consider our reply to Reviewer #3: 7aZ9. We offer to revise the experimental section to include comparison with the previous unsupervised work of Doumanoglou in the main text, instead of including it in the appendix. In that case, we would find helpful, if the reviewer would like to suggest which other part of the experimental section may be omitted from the main text (and possibly moved to the appendix) in order to fit the space restrictions of the manuscript.
>   - Regarding influence comparison, things are getting more complicated, in the sense that even a random direction may have the potential to be influential, yet whether the direction corresponds to the true direction of a concept in the latent space is questionable. When trying to explain the latent space, concept signal directions are latent variables that we aim to estimate while we do not know the precise ground-truth. To our knowledge, the state-of-the-art in signal direction estimation (which considers a binary label data model) is the one of Pattern-CAVs (Pahde 2024). Yet, as we demonstrated in the experiment with synthetic data, even this method may fail to capture the true concept signal directions when applied to the multi-concept data model that we introduced in Section 2. In their work, Pahde at al, have demonstrated that their method is the best estimator of a concept's signal under the binary-label datamodel, being significantly more reliable compared to numerous common alternatives, such as "filter-CAVs" learned with logistic regression, LASSO regression or SVMs. This motives to compare against Patter-CAVs completly overlooking other alternatives that we know they are not accurate concept signal estimators. Even though, pattern-CAVs did not estimate correctly the signal direction in our synthetic experiment, we did compare for influence against them in the experimental section and we found that, overall, the directions suggested by pattern-CAVs do not influence model predictions more than the learned signal vectors, which makes our method competitive regarding finding influential directions. Additionaly, our method works significantly better than pattern-CAVs in estimating the true concept signal directions in the synthetic benchmark with known ground truth.

---

> ### Author Response · Authors · 2024-11-18
> **Answers the questions**
>
> >Questions:
>
> >Is the proposed method applicable to other types of data beyond images, such as text or time series data? If so, what modifications, if any, would be necessary?
>
> - **Reply**:
>     We think to answer this question we need to verify whether the assumptions of the method holds for text or time-series data. The main assumption that is made in the proposed approach is that the number of semantic labels that can be attributed to an image patch is small, compared to the number of possible semantic labels that could be attributed to it. For instance a patch may have the semantic label (concept) "tree" among a plethora of other possible semantic labels such as "car,sky,road,person,dog,cat,...". As soon as a spatial element in the intermediate representation corresponds to a single image patch (as in CNNs or vision transformers) the assumptions of the method are not violated. While the image domain naturally fits the assumptions made in this paper, we think that applying the method to the text domain and time-series architectures (besides transformers) like RNNs or LSTMs  needs to carefully reconsider the assumptions and possibly adopt the method in an a more appropriate way. Yet, we don't have a concrete idea on how this could be addressed.
>
>
> >How does the method scale with larger models and datasets? Are there computational bottlenecks, and if so, how can they be mitigated?
>
> - **Reply**: The proposed method discovers directions in the latent space of a **pre-defined layer of a frozen deep model** (i.e. the method is post-hoc). First, the method has linear complexity compared to dataset size provided that the studied layer of the deep model is the same. Second, if the deep model remains the same but the studied layer is different than the reference layer, the computatinal complexity of the method is analogous to the ratio between the size of an image representation in the new layer over the size of the image representation in the reference layer. For instance, in a standard Vision Transformer (ViTs) where the embedding dimension and the number of tokens of an image are constant across layers, the complexity of the method is equal across different layers. In another example, in the case of a CNN where the last layer has dimensionality 7x7x512 and the second-last layer has 14x14x512, the complexity of the method when applied to the second-last layer is (14x14x512)/(7x7x512) = 4 times more the complexity of applying the method in the last layer. As a mitigation measure to reduce the complexity of the method, the features of the studied layer are first recorded through one forward pass across the dataset (i.e. one epoch), to trade off storage against computation time. When storage is fast (e.g. SSD hard drives) this may save a lot of computation time which would be spent in re-computing the image features (which are constant) while learning the directions. However, in cases when an intermediate layer's representation has large dimensionality (either in its spatial dimentions or the feature embedding dimension) and for a larger dataset, the cost of saving the features in the hard drive may be high (for instance in a ViT architecture with spatial feature dimensionality of 28x28 and embedding dimension 768, as it is common in state-of-the-art ViTs) Provided that we recorded the intermediate features, in the next step we prune the studied network by keeping only the layers from the layer that follows the layer of study and up, and use this pruned model to compute $\mathcal{L}^{uur}$ and $\mathcal{L}^{cur}$ (this is the $f^+$ part of the network that we mention in Section 3.2.1).

---

> ### Author Response · Authors · 2024-11-27
>
> >While the experiments are promising, the evaluation is limited to a synthetic dataset and a single real-world dataset (Places365 with ResNet18). A broader range of datasets and models would strengthen the claims and demonstrate the generalizability of the method
>
> In the revised version of the manuscript, we moved the comparison with previous unsupervised work, from the appendix to the main text. In particular to Section 4.2 and Table 3. This introduces one more model (Resnet50) and one more dataset (Moments in Time) as a real-world comparison baseline for our method.

---

### Author Response · Authors · 2024-11-18
**Comment to all the reviewers**

We want to cordially thank all the reviewers for their hard efforts to make constructive suggestions regarding the improvement of our, admittedly, difficult-to-read manuscript. Unfortunately, based on the reviewer feedback, we have the honest impression that the critical details, that would make our contributions more appreciated, were not sufficiently comprehended by the reviewers. Indeed, as one of the reviewer mentioned, in many cases throughout the text, we refer the reader either to the appendix or to another ciatation. We totally understand that this pattern can make the review process difficult, espessialy due to the limited time available for the review process. We are willing to take responsibility for this and work towards a better average and even expert reader's comprehension. However, in the direction of remaining practical, in the limited space of 10 pages, we find it very challenging to enrhich the content of the main paper with more clarifications on contributions or related work while, at the same time, adding and commenting on more experimental results, as some of the reviewers have suggested. Moreover, we hypothesize that not all of the reviewers have considered our ~20-page appendix (twice the length of the main paper) especially as, in some cases, the answers to some (admittedly not all) of their questions are, at least to some extend, there.

While we recognise the value of experimental results, to move things forward, we think that it is more important to prioritize on improving the comprehension of our theoretical contributions (which is the main point of this paper) instead of enriching the experimental results section with additional text which would imply shortening the details of our method which is already of, some-what, limited comprehensibility. We are going to try our best, but we still think that for some details it will be inevitable to make use of the appendix.

We are open to concrete suggestions from all the reviewers regarding which parts of the main text they feel could be ommited (or deferred to the appendix) in the favor of more explanations regarding the core-concepts of related work. Our first impression, is to ommit  Section 3.4 and parts related to the ablation studies in Section 4.2, but we would really appreciate a feedback from the reviewers.

Once again, we want to express our cordial appreciation to the efforts of all the reviewers. Honestly, we consider your feedback extremelly valuable for the improvement of our manuscript. Below you may find our comments on your specific concerns.

---

### Author Response · Authors · 2024-11-27
**Comments on the revised version**

Dear reviewers,

We would like to sincerely thank you once again for your thoughtful and constructive feedback, which significantly helped us in improving our manuscript. In this revised version, we have made substantial efforts to address your concerns and suggestions to the best of our ability. The key improvements of the revised version are structural. In the revised version, we shortened the introduction, we revised the background section to discuss only on prior work, and clearly distinguish and describe our contributions in Section 3, while the experiments section on deep-image classifier has been enriched.

In more detail, the changes we made in this revision are:

- The introduction section has been re-written. It now more clearly mentions the contributions of our work compared to the prior work that we are based on.
- A graphical abstract figure has been added to the second page of the manuscript (Fig. 1), which aims to explain the core concepts used throughout the paper. It also presents the overall pipeline of the method, from learning the directions, to interpretability and influence evaluation.
- The background section has been written from scratch and is considerably longer than the previous version. However, it now more clearly mentions all the important concepts needed in order for a reader to understand our contributions. The new section solely describes prior work. In particular, in Section 2.2 we explain Signals/Distractors/Filters/Pattern-CAVS, in Section 2.3 we shortly present the important concepts and loss terms from Doumanoglou et al, in Section 2.4 we discuss on the use of Network Dissection and in Section 2.5 we discuss on the application of RCAV (with an additional more detailed discussion in Section A.8).
- The Method section has also been considerably revised, especially in terms of structure. All parts of Section 3 are important contributions that we make in this work, clearly separated from contributions of prior work that are now discussed only in the Background section.
- Section 3.1.3 (Maximum Margin Loss) of the original manuscript is now discribed in Section 2.3 and also discussed in the appendix section A.5.
- Section 3.2.1 (Baseline Influential Direction Loss) from the original version of the manuscript has been moved to the appendix section A.7. This loss term served as an alternative baseline against which we compare uncertainty region alignment loss and is not part of the proposal that we make in this manuscript since we show that Uncertainty Region Alignment is more effective in concept direction learning. The experiments that regard this loss have been moved from Section 4.2 to the appendix of the revised version (Table 7).
- Section 3.2.2 (Filter-Signal Orthogonality Loss) of the original manuscript is now discussed in Section 3.3: "Signal Vectors as Concept Signal Estimators".
- Section 3.4 (Direction Learning Process) from the original version of the manuscript has been moved to the appendix section A.9.
- The introductory paragraph of Section 3 of the revised version includes additional description regarding the motivation behind each contribution that we make.
- Section 3.2 of the revised version now clearly mentions what is the new data model that we propose in this work and how it is different from prior work.
- Section 3.3 of the revised version now more clearly mentions what signal vectors are and how they are learned.
- Section 3.4 of the revised version, now further elaborates on motivation regarding the Uncertainty Region Alignment loss.

---

> ### Author Response · Authors · 2024-11-27
>
> - The section regarding the experiment on Synthetic Data (Section 4.1) has been considerably shortened in the revised version defering details recarding RCAV evaluation to the appendix Section A.10. The previous Table 1 has been moved to Table 8 in A.10 and Fig. 4 has been moved to Fig.6  in the same section. Also the previous Fig. 3 has been updated with a more coinsise version (while the previous version is now again included in the appendix as Fig. 5). The discussion against the moved figures has also been shifted to the appendix.
> - Section 4.2: the real-world experiment has also been significantly revised. We moved the details regarding RCAV's concept sensitivity testing (the label permutation test) to the appendix section A.12. We also moved the ablation regarding the number of concept detectors $I$ to the appendix in the same section and the previous Table 3 has been moved to Table 7. In the revised version, Table 1 has been enriched with one more row (a row taken from the last row of old Table 3 - right) and the new Table 2 is a shorter version of the old Table 3, excluding the rows regarding the Baseline Influential Direction Loss that whose description, as we mentioned previously, we have also moved to the appendix.
> - Section 4.2 has also been enriched with the following elements: a) Description of Interpretability Losses from prior work b) a short description on Broden dataset c) Comparison against previous unsupervised works + Table 3 (this also contributes an additional baseline against which we compare against: Resnet50 trained on Moments in Time) d) Fig.4 which demonstrates an explanation provided with directions learned with our method when combined with Network Dissection and RCAV.
> - According to the request of one of the reviewers, we added discussion compared to classical matrix decomposition approaches in Section A.2.
> - According to the request of one of the reviewers, in the revised version of the manuscript we added an example case of applying our method on model-correction in a toy setting. The experiment demonstrates that our method was able to identify a bias of the model and by using the learned concept directions we were able to edit the model, improving its generalization performance. The reviewers are kindly asked to refer to section A.14.
>
> Finally, since this manuscript has received a lot of critisism regarding the interpretabilty of the directions, apart from the quantitative results of Table 5 (which demonstrate high precision in identifying concepts aligned with human intuition), we would like to kindly ask all the reviewers to skim through section A.13 to see example segmentation results obtained with our learned classifiers. Classification metrics regarding specific concepts are also provided in the appendix Section A.12.1. Regarding how our method may be used to provide explanations, again we would like to kindly ask the reviewers to refer to example explanations provided in Section A12.2 where we demonstrate how the learned directions can be combined with the global explainability method RCAV.

---

### Meta-Review · Area_Chair_vRv2 · 2024-12-16

**Metareview:**

This paper proposes an unsupervised method to discover semantically meaningful and actionable directions in the latent space of neural networks for debugging and alignment. The main criticism, mentioned by several reviewers is that the paper is difficult to read and lack standard baselines such as PCA / ICA / dictionary learning. While the authors claim these should not work well, this would be one more reason to actually try. Overall, the positioning with respect to related methods was called out several times and, while the authors improved the submission during the rebuttal, the paper should go through another round of reviews.

**Additional Comments On Reviewer Discussion:**

The reviewers were engaged with the authors with active discussions. The authors however did not address explicit requests from reviewers (e.g., concerning baselines and writing clarity).

---

### Decision · Program_Chairs · 2025-01-22

Reject